# Numerical issues in modeling thermally and hydraulically driven ice stream surge cycling

Kevin Hank[1], Lev Tarasov[1], and Elisa Mantelli[2,3,4]

[1]Department of Physics and Physical Oceanography, Memorial University of Newfoundland, St. John's, NL, A1B 3X7,Canada

[2]Department of Earth and Environmental Sciences , Ludwig-Maximillians-Universitaet Munich, Theresienstr. 41, 80333 Munich, Germany

[3]Alfred Wegener Institute for Polar and Marine Research, Am Alten Hafen 26, 27568 Bremerhaven, Germany

[4]Institute for Marine and Antarctic Studies, University of Tasmania, 20 Castray Esplanade, Battery Point TAS 7004, Australia

[*]khank@mun.ca

**Correspondence:** Kevin Hank (khank@mun.ca)

**Abstract.** Modeling ice sheet instabilities is a numerical challenge of potentially high real-world relevance. Yet, differentiating between the impacts of model physics, numerical implementation choices, and numerical errors is not straightforward. Here we use an idealized North American geometry and climate representation (similar to the HEINO experiments, Calov et al., 2010) to examine the numerical sensitivity of ice stream surge cycling in ice flow models. Through sensitivity tests, we iden-

5   tify some numerical requirements for a more robust model configuration for such contexts. To partly address model-specific dependencies, we use both the Glacial Systems Model (GSM) and Parallel Ice Sheet Model (PISM). We show that modeled surge characteristics are resolution-dependent though converging (decreasing differences between resolutions) at higher horizontal grid resolutions. Discrepancies between high and coarse horizontal grid resolutions can be reduced by incorporating sliding at sub-freezing temperatures. The inclusion of a bed thermal model markedly reduces the ice volume lost during surges

10  in both the GSM and PISM, as the substrate heat storage capacity dampens the change in basal temperature during surges. The inclusion of basal hydrology, as well as a non-flat topography, leads to an increase in the ice volume lost during surges in both models. Therefore, we conclude that these latter three components are essential to maximizing physical fidelity in ice stream surge cycle modeling.

## 1 Introduction

### 1.1 Motivation and background

The use of Ice Sheet Models has grown at least an order of magnitude over the last two decades. The relevance of such modeling studies to the actual physical system can be unclear without careful consideration and testing of numerical aspects and

implementations. This is especially true when modeling the highly non-linear ice sheet surge instability, which has significant implications not only for the ice sheet itself but also for the climate. In fact, it is often difficult to assess whether model results are physically significant (effects of physical system processes), a consequence of model-specific numerical choices, or a combination of both. This is especially important in the case of abrupt changes. Whether ice sheet instabilities observed in numerical simulations are the result of physical instabilities of the underlying continuum models or spurious effects of the discretization and numerical implementation of said models has long been debated (e.g., Payne et al., 2000; Hindmarsh, 2009) and is a consequential matter. The present study is concerned with characterizing the impact of model physics, numerical choices, and numerical errors on ice stream surge cycling.

Binge-purge ice stream cycling was first introduced in the glaciological literature by MacAyeal (1993) as an explanation for Heinrich Events arising from the former Laurentide Ice Sheet (LIS) in the Hudson Bay/Hudson Strait region. The key idea is that the ice stream gradually grows to a threshold thickness (binge phase) driven by surface accumulation. Once the ice stream is thick enough to sufficiently isolate the ice stream base from the cold surface, heat from geothermal and deformation work sources can slowly bring the basal temperature to the pressure melting point. The bottom layer of the ice stream is no longer frozen to the bed and thus enables basal sliding. Localized warm-based ice streaming increases the ice stream surface gradient (steeper slope) at the warm/cold-base transition point, leading to an increase in driving stress. The resultant increase of deformation work can warm the surrounding ice close to or to the pressure melting point, thus enabling (sub-temperate) basal sliding (Fowler, 1986). When the melting point is reached, the presence of water at the ice sheet/bed interface (decreased basal friction, Fowler and Schiavi, 1998) as well as in a deformable sediment layer (loosened sediment, Bueler and Brown, 2009) can further increase sliding velocities. Instead of the slow deformation flow (ice creep), the ice stream now flows rapidly (purge phase). As a consequence of the high ice velocities, the ice stream thins and cold ice is advected from either upstream or the lateral boundaries of the ice stream. Cold ice advection in combination with changing heat source contributions (from both deformation work and basal sliding) and lowering of the pressure melting point as ice thins eventually leads to refreezing of the ice/bed interface. The first localized frozen patch of ice acts as a *pinning point*, supporting some of the driving stress and decreasing the velocities and heat production in the adjacent ice. This marks the end of the surge, thus enabling the ice stream to enter the next binge phase. Whether hydraulically or thermally driven, these activation (purge) and stagnation (binge) phases can alternate in a quasi-periodic fashion (e.g., Robel et al., 2013) - this is what we refer to as 'ice stream surge cycling' in the remainder of this paper.

As a result of the involved physics and expected behaviors, modeling of ice stream surge cycling is challenging. The challenges entail, among others, rapid surge onset, high ice velocities, and non-linear (thermo-viscous, hydraulic, and thermo-frictional) feedbacks. In addition to the physical complexity, further challenges arise in the numerical modeling of ice stream surge cycling, whether in terms of model choices (e.g., choice of mechanical model, thermal modeling of the substrate, accounting for sub-glacial hydrology) and/or in terms of their numerical implementation (e.g., grid and time step size, convergence under grid refinement, etc.).

Our focus here is on the challenges arising from numerical modeling, both those related to the modeling choices and those related to the implementation. Numerical challenges have received limited attention in studies examining ice sheet surging.

The few studies to date that do examine numerical aspects of surge cycling suggest strong sensitivities in model response to implementation choices such as grid size (e.g., Calov et al., 2010; Roberts et al., 2016; Ziemen et al., 2019). However, the effects of different approximations of the Stokes equations have been previously addressed (e.g., Brinkerhoff and Johnson, 2015), and are therefore not discussed here.

The discretization and related numerical implementation choices (e.g., grid resolution, grid orientation, and time step size) have been shown to affect numerical results. As far as the choice of grid is concerned, Ziemen et al. (2019), for example, find a constantly active ice stream at $40 \, \mathrm{km}$ grid resolution and oscillatory behavior at $20 \, \mathrm{km}$ grid resolution. They argue that this higher grid resolution is necessary to resolve the Hudson Strait properly. However, only a few studies examine the effect of different grid resolutions on surge behavior (e.g., Payne and Dongelmans, 1997; Greve et al., 2006; Van Pelt and Oerlemans, 2012; Brinkerhoff and Johnson, 2015; Roberts et al., 2016) and an in-depth numerical analysis of Hudson Strait ice stream surge cycling (to whatever idealized form) is absent. In terms of grid rotation, Greve et al. (2006) and Takahama (2006) show only a minor effect of grid rotation on the general features of the oscillations.

The effect of time stepping on modeled surge cycling appears to be weak or entirely absent. Greve and MacAyeal (1996) examined the impact of different time steps on ice stream surge oscillations in a coupled dynamic/thermodynamic flowline model. They report similar dynamic behavior across different time steps, but both test runs crashed. Later studies using a three-dimensional version of the same but further developed model find shorter periods and a slight decrease in surge amplitude but otherwise reasonable convergence as the time step decreases (Greve et al., 2006; Takahama, 2006). Yet, none of the more recent studies on ice stream surge cycling includes experiments with different time steps.

An additional level of complexity in the modeling of ice sheet surge cycling arises from the fact that small perturbations of the initial or boundary conditions can significantly vary the surge characteristics (Souček and Martinec, 2011; Mantelli et al., 2016). For example, Souček and Martinec (2011) show that low levels of surface temperature noise can lead to chaotic behavior in the periodicity of ice stream oscillations, with mean periods varying by $\pm 2 \, \mathrm{kyr}$ ($\sim 20 \, \%$ of characteristic period of the oscillations, Fig. 8 in Souček and Martinec (2011)). Moreover, Souček and Martinec (2011) find differences in form, period, and amplitude of oscillations when using two different numerical implementations for calculating the basal temperature for thermal activation of basal sliding. However, whether this observed sensitivity arises from physical grounds (e.g., as in Mantelli et al., 2016) or is a spurious numerical effect, the numerical error remains unclear. Souček and Martinec (2011) thus rightfully conclude that *"... the implementation of surge-type physics in large-scale ice-sheet models is rather problematic since the information about the physical instability may be lost in the numerics"*.

## 1.2 Study overview

With this as a starting point, in this study we seek to disentangle the effects of numerical choices (both in terms of model components and in terms of their implementation) on ice sheet surges. We will do so through a numerical modeling study.

In terms of ice flow models, we primarily use the 3D glacial systems model (GSM, Tarasov et al., 2023). However, to mitigate the possibility that our conclusions are biased by specific numerical/modeling choices within the GSM, we repeat experiments that do not require implementation of novel physics with the widely used Parallel Ice Sheet Model (PISM, Bueler and Brown,

2009; Winkelmann et al., 2011). As the two model setups and physics are somewhat different (see Table 2 for details), we do not intend to compare model results directly. Instead, our aim is to increase confidence in model results by showing that the same conclusions can be drawn from two different models.

In order to partly address potential non-linear dependencies of surge cycling on model parameters, we run each of our numerical experiments with a high variance ensemble of 5 GSM parameter vectors (each comprising 8 model input parameters) and 9 PISM parameter vectors (each comprising 6 model input parameters).

In terms of different numerical choices, the impact on model results is usually determined by calculating the model error to the exact analytical solution. However, the theory behind the surge instability is not fully developed (no analytical solution exists), especially in the context of a spatially extended 3D system, thus precluding systematic benchmarking of numerical models.

To overcome this issue and provide at least a minimum estimate of the numerical model error, we first determine 'Minimum Numerical Error Estimates' (MNEEs). This is a new metric that aims to minimally resolve whether a change in surge characteristics due to changes in the model configuration is significant (see Sec. 2.3 for details).

Equipped with these tools, we set out to tackle the research questions detailed in Sec. 1.3, which we denote with labels $Q_1 - Q_{11}$. The remainder of the paper is then structured as follows: we start by describing our models and experimental setups in Sec. 2. We then present detailed results that allow us to answer our research questions in Sec. 3, with a concise summary provided in Sec. 4. The results are organized into the following main themes: 1) key surge characteristics of the reference setup (Sec. 3.1), 2) MNEEs (Sec. 3.2), 3) sensitivity experiments with and without a significant (with respect to the MNEEs) effect on the results (Sec. 3.3), and 4) convergence study (Sec. 3.4).

## 1.3 Key research questions

In this subsection, we detail the key research questions that we address through numerical experiments. Following the above-described structure in the description of the results, the research questions are divided into three sub-categories: minimum numerical error estimates (MNEEs), sensitivity experiments, and convergence study.

**Minimum numerical error estimates**

$Q_1$ *What is the threshold of MNEEs in the two models (Sec. 3.2)?*

Surge cycling is sensitive to numerical aspects (e.g., numerical solver error). Since we can not determine the model error, we provide a minimum estimate of the numerical error in the models (see Sec. 2.3 for details). These MNEEs set a minimum threshold for discerning whether the model response to a change in model configuration is significant.

**Sensitivity experiments**

Here we aim to determine the significance of different model configurations on the surge characteristics. We are particularly interested in model configurations affecting the basal temperature and thus the surge behavior. Therefore, we first discuss the

change in surge characteristics due to a bed thermal model ($Q_2$) and modeling choices affecting the basal temperature at the grid cell interface where the ice velocities are calculated ($Q_3$ and $Q_4$), including the basal sliding thermal activation criterion ($Q_5$). Previous studies examining the effects of ice stream behavior are often based on an idealized basal topography and sediment distribution and do not consider sub-glacial hydrology (e.g., Calov et al., 2010; Brinkerhoff and Johnson, 2015). Therefore, we determine the change in surge characteristics due to these aspects in $Q_6$, $Q_7$, $Q_8$, and $Q_9$, respectively. Since thermally and hydraulically driven ice stream surges are not exclusive, we also investigate the differences between the two mechanisms when used as the primary smoothing mechanism at the warm/cold-based transition zone ($Q_{10}$).

$Q_2$ *Is the inclusion of a bed thermal model a controlling factor for surge activity (Sec. 3.3.1)?*

Except for PISM, all models in the HEINO experiments did not include a bed thermal model (Calov et al., 2010). PISM is one of the few models that did not show oscillatory behavior in the HEINO experiments (except for experiment T1 (10 K colder minimum surface temperature, Calov et al. (2010))). We explore the role of the additional heat storage on surge activity by deactivating a 1 km deep bed thermal model in the GSM and PISM.

$Q_3$ *Do different approaches to determining the grid cell interface basal temperature significantly affect surge behavior, and if yes, which one should be implemented (Sec. 3.3.2)?*

Ambiguity arises when determining the basal temperature at the grid cell interface. On a staggered grid (commonly Arakawa C grid, Arakawa and Lamb, 1977), the velocities are calculated at the grid cell interfaces, whereas basal temperatures are situated in the grid cell center. Therefore, the basal temperature at the grid cell interface needed for the thermal activation of basal sliding needs to be determined as a function of the basal temperatures at the adjacent grid cell centers. Here we examine surge sensitivity to different interpolation schemes (see Sec. 3.3.2).

$Q_4$ *How much of the ice flow should be blocked by upstream or downstream cold-based ice, or equivalently, what weight should be given to the adjacent minimum basal temperature (Sec. S8.1)?*

At relatively coarse horizontal grid resolutions (e.g., 25 km), the basal temperatures at the adjacent grid cell centers are of physical relevance. For example, a cold-based grid cell in the downstream direction should block at least part of the ice flow across a 25 km long warm-based interface (Eq. (S1)). Here we examine surge sensitivity to a change in the weight of the adjacent (grid cell center) minimum basal temperature when calculating the grid cell interface temperature.

$Q_5$ *How different are the model results for different basal temperature ramps and what ramp should be used (Sec. 3.3.3)?*

Another issue that is often ignored is the basal sliding thermal activation criterion. Based on the results of Souček and Martinec (2011), the basal temperature is a critical factor in the onset and termination of (surging) ice streams. Mantelli et al. (2019) show that an abrupt onset of sliding at the transition from a cold-based ice sheet to an ice sheet bed at the pressure melting point causes refreezing on the warm-based side and, therefore, cannot exist. Observational and experimental evidence for sub-temperate sliding further supports a smooth transition from cold-based no-sliding conditions to fully warm-based sliding, with sliding velocities increasing as the basal temperature approaches the pressure melting point (Barnes et al., 1971; Shreve, 1984; Echelmeyer and Zhongxiang, 1987; Cuffey et al., 1999; McCarthy et al., 2017).

An additional argument for sub-temperate sliding can be made on numerical grounds for coarse horizontal grid resolutions. It is unlikely that an entire grid cell reaches the pressure melting point within one time step (e.g., 25x25 km in 1 yr). As such, the activation of basal sliding should start at grid-cell basal temperatures below the pressure-melting point and ramp up as the pressure-melting point is approached. As the horizontal grid resolution becomes finer, the range of sub-grid temperatures in a grid cell decreases (e.g., Figs. 10, S26, and S27). Consequently, the thermal activation ramp should be sharper (smaller transition zone) for higher horizontal grid resolutions, but the exact width of the ramp at the highest horizontal grid resolutions is unknown.

Experimental work (e.g., Barnes et al., 1971; McCarthy et al., 2017) supports the notion of sub-temperate sliding within a narrow range of temperatures below the pressure melting point ($< 5°C$). A wide temperature ramp (e.g., $T_{ramp} = 1°C$, see Eq. (9)) enables an earlier sliding onset (for increasing basal temperature), spatially extended sliding, and a prolonged sliding duration (for decreasing basal temperature).

While on theoretical ground, the sensitivity of sliding speeds on temperature is expected to grow with proximity to the pressure melting point, the appropriate functional form of the temperature ramp is not well constrained nor is the sensitivity to the functional form well documented. Herein, we use basal temperature gradients in high-resolution runs and approximations of the sub-grid warm-based connectivity between the faces of, e.g., a 25 km grid cell (there should be no ice streaming across the grid cell if a frozen sub-grid area disconnects warm-based patches) to constrain an apriori functional form of the basal temperature ramp. We then use upscaling and resolution-scaling experiments to constrain the dependency of the ramp on horizontal grid resolution.

$Q_6$ *Does the abrupt transition between a soft and hard bed significantly affect surge characteristics (Sec. 3.3.4)?*

An abrupt transition from hard bedrock to soft sediment can lead to additional localized shear heating caused by the difference in basal resistance and therefore sliding velocities at that transition. We explore the impact of the bed-type transition on surge characteristics by incorporating a smooth transition from $0 \%$ sediment cover (hard bedrock) to $100 \%$ (soft) sediment cover effectively changing the basal sliding coefficient $C$ in Eq. (6b).

$Q_7$ *How does a non-flat topography affect the surge behavior (Sec. 3.3.4)?*

Given the topographic lateral bounds of Hudson Strait, we examine the effects of a non-flat topography on the surge characteristics.

$Q_8$ *What is the effect of a simplified basal hydrology on surge characteristics in the GSM (Sec. 3.3.5)?*

The implementation of a fully-coupled basal hydrology model changes the basal drag and, therefore, has the potential to affect the surge characteristics. A basal hydrology model coupled to an effective-pressure dependent sliding law, or a Coulomb-plastic bed (as in PISM), introduces a positive feedback such that larger sliding speeds increase frictional heating, and thus meltwater availability which further weakens the bed and leads to even faster sliding. Different basal hydrology process representations have been proposed in the literature (e.g., a 0D (Gandy et al., 2019), poroelastic (Flowers et al., 2003), or linked cavity hydrology model (Werder et al., 2013)), and in-depth comparison is currently

 under review (Drew and Tarasov, 2022). Here we compare GSM surge statistics with and without a fully coupled 0D hydrology model.

$Q_9$ *How significant are the details of the basal hydrology model on surge characteristics in PISM (Sec. S8.2)?*

PISM surge characteristics are compared for local and mass-conserving horizontal transport hydrology models.

$Q_{10}$ *What are the differences (if any) in surge characteristics between local basal hydrology and a basal temperature ramp as the primary smoothing mechanism at the warm/cold-based transition zone (Sec. S8.3)?*

While both sub-glacial hydrology and a basal temperature ramp provide a means for a smooth increase in sliding velocities, these processes operate in slightly different temperature regimes. The basal temperature ramp enables sub-temperate sliding and the maximum velocities occur once the pressure melting point is reached. In contrast, a local basal hydrology model increases sliding velocities once the basal temperature reaches the pressure melting point (basal melting), and basal ice velocities further ramp up with decreasing effective pressure (ice overburden pressure minus basal water pressure). Note that sub-glacial hydrology is not an alternative for a basal temperature ramp. The ramp is still needed to prevent refreezing even when a description of sub-glacial hydrology is included (Mantelli et al., 2019).

**Convergence study**

$Q_{11}$ *Do model results converge (decreasing differences when increasing horizontal grid resolution and decreasing maximum time step, Sec. 3.4)?*

Significant surge pattern differences can occur when computationally more feasible (coarser) horizontal grid resolutions are used (e.g., $\sim 200$-fold increase in the GSM run time when increasing the horizontal grid resolution from 25 km to 3.125 km). Incorporating the findings of the above experiments, we study numerical convergence with respect to horizontal grid resolution and time step for surge cycling. By convergence, we mean decreasing differences between simulations when increasing the resolution or decreasing the time step. True model verification can only come from comparison with continuum model results, which are, however, only available to a very limited degree and do not encompass the process complexity considered here, so we limit ourselves to a formal study of numerical convergence.

## 2 Methods

### 2.1 GSM

#### 2.1.1 GSM model description

The 3D thermo-mechanically coupled glacial systems model (GSM) has developed over many years (e.g., Tarasov and Peltier, 1997; Tarasov et al., 2012; Bahadory and Tarasov, 2018). It includes an energy-conserving finite volume ice and bed thermo-dynamics solver. The current hybrid shallow shelf shallow ice physics is based on a slight variant of the ice dynamical core of

Pollard and DeConto (2012). As is standard for thermo-mechanically coupled glaciological ice sheet models, the GSM has a default explicit time step coupling between the thermodynamics and ice dynamics but also includes an optional implicit coupling scheme (c.f. Sec. 3.2.2). Ice dynamical time stepping is subject to CFL (Courant–Friedrichs–Lewy) constraint (Courant et al., 1928) with further reductions upon ice dynamical solver convergence failure. The source code of the model version used in this manuscript can be found in the supplementary material (Tarasov et al., 2023).

The GSM is run with an idealized down-scaled North American geometry (Fig. 1, modified after the ISMIP-HEINO setup (Calov and Greve, 2006)) and simplified climate representation. The surface temperature forcing in the GSM is given by

$$T_{\mathrm{surf}} = \mathrm{rTsurf} + \mathrm{lapsr} \cdot H + T_{\mathrm{asym}}, \tag{1}$$

where rTsurf and lapsr are input parameters for the domain-wide surface temperature constant and atmospheric lapse rate, respectively (Table 1), $H$ the ice sheet thickness, and $T_{\mathrm{asym}}$ the asymmetric (in time) temperature forcing (maximum difference of $10°$C, orange line in Fig. S1) calculated according to

$$T_{\mathrm{asym}} = \left| \left( \frac{t}{200 \text{ kyr}} \cdot 3 + 2 \right) - 1 \right| \cdot 5°\text{C}, \tag{2}$$

where $t$ is the model time ranging from $-200$ kyr to $0$ kyr (instead of $0$ kyr to $200$ kyr). The asymmetric temperature forcing enables the analysis of the timing of cycling onset and termination under different physical and numerical conditions (a comparison of ice stream ice volume evolution under constant and asymmetric temperature forcing is shown in Fig. S2 for one parameter vector).

The surface mass balance forcing is then determined by

$$M_{\mathrm{tot}} = M_{\mathrm{acc}} - M_{\mathrm{melt}}, \tag{3}$$

where $M_{\mathrm{acc}}$ and $M_{\mathrm{melt}}$ are the surface accumulation and melt, respectively. The surface accumulation is defined by

$$M_{\mathrm{acc}} = \mathrm{precRef} \cdot \exp\left( \mathrm{hpre} \cdot T_{\mathrm{surf}} \right), \tag{4}$$

where precRef and hpre are the precipitation coefficient input parameters. Surface melt is calculated according to a Positive Degree Day (PDD) approach:

$$M_{\mathrm{melt}} = \mathrm{rPDDmelt} \cdot \max\left( 0.0, \mathrm{POSdays} \cdot (T_{\mathrm{surf}} + 10.0°\text{C}) \right), \tag{5}$$

where rPDDmelt is the input parameter for melt per PDD and the PDD constant POSdays is set to $100$ days yr$^{-1}$. Note that we set $T_{\mathrm{surf}} = 0.1°$C and $M_{\mathrm{tot}} = -100$ m yr$^{-1}$ for ocean grid cells, and $T_{\mathrm{surf}} = 0.1°$C and $M_{\mathrm{tot}} = -200$ m yr$^{-1}$ at the boundaries of the model domain.

The GSM is initialized from ice-free conditions and the first 20 kyr of each model run are considered as spin-up interval, which is not included in the analysis of the surge characteristics. Note that this is a very conservative spin-up interval. Most runs reach their mean pseudo-Hudson Strait ice volume after $\sim 5$ kyr (e.g., Fig. 11). The coarsest horizontal grid resolution is

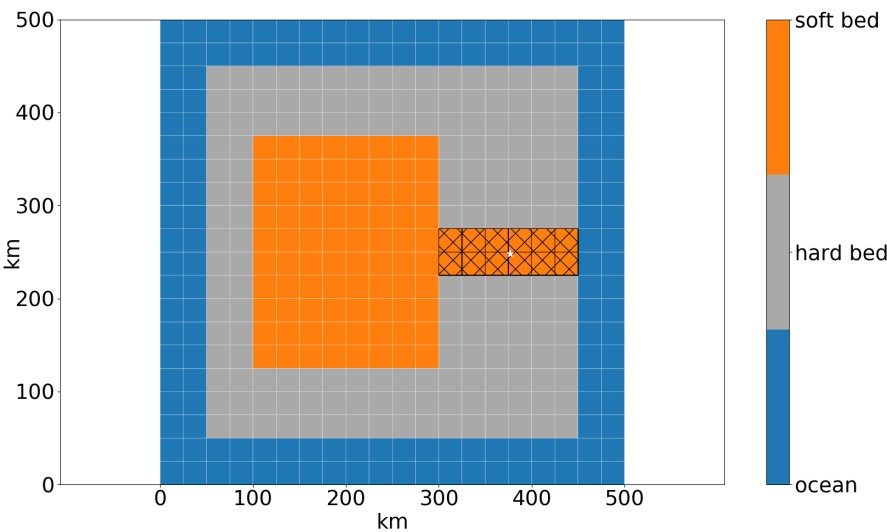

**Figure 1.** Modified ISMIP-HEINO geometry (Calov and Greve, 2006). The model domain is reduced to 500x500 km to enable horizontal grid resolutions up to 3.125 km. The shown grid resolution is 25x25 km. The basal topography is flat and the hatched area marks the soft-bedded pseudo-Hudson Strait. The white star indicates the location of the grid cell shown in Fig. 8+S21.

25x25 km and is progressively refined (halved) to 3.125x3.125 km. This gives a total of 4 different horizontal grid resolutions. The maximum time step is 1 yr (automatically decreased as needed to meet CFL constraint or when convergence fails).

While Mantelli et al. (2019) conclude that Stokes mechanics are needed to arrive at a mathematically well-posed model, running numerical experiments with a thermo-mechanically coupled Stokes model is to date unfeasible for a glacial cycle context. Previous ice stream surge modeling studies are often based on zeroth-order, thin-film approximations of the Stokes problem, like the shallow-ice Approximation (SIA, e.g., 8 out of 9 models in the ISMIP HEINO experiments (Calov et al., 2010)). While resolving vertical shear, which is the dominant mode of motion in slow flowing regions, SIA-based models
neglect longitudinal stress gradients and horizontal shear, which are known to be important for fast ice streams (Hindmarsh, 2009) and are instead captured by the zeroth-order shallow-shelf approximation (SSA).

    To partially offset the limitations of the zeroth-order approximations, the GSM uses hybrid SIA/SSA ice dynamics. This heuristic combination links the two sets of equations by including a shear softening term in the calculation of the effective viscosity in the respective other set (SIA internal shear in the SSA viscosity calculation and SSA vertical-mean longitudinal
stretching in the SIA viscosity calculation) (Pollard and DeConto, 2007, 2012). Additionally, horizontal shear and longitudinal stress gradient terms from the SSA equations reduce the driving stress in the SIA equations (Pollard and DeConto, 2007, 2012). A third coupling option adds the distinction between the depth-averaged internal-shear and basal velocity to the SSA basal stress term (Pollard and DeConto, 2007, 2012). This coupling term, however, tends to weaken numerical convergence without having much impact on ice sheet history and was therefore not used for the experiments in this paper. The hybrid SIA/SSA

ice dynamics are activated for grid cells with a SIA velocity exceeding $30 \text{ m yr}^{-1}$. Changing these activation velocities has no significant effect on the surge characteristics. Activating the SSA everywhere leads to more, shorter, and weaker surges because no threshold velocity needs to be overcome to initiate basal sliding (Sec. S1.2). Note that we set an upper limit of $40 \text{ km yr}^{-1}$ for the SSA velocity to ensure that sliding velocities stay within a physically reasonable range.

We set the GSM with a $1$ km deep ($17$ non-linearly-spaced levels) bed thermal model. A basal temperature ramp is used to
ensure a smooth transition between cold-based regions of no sliding and temperate sliding, account for observational evidence of sub-temperate sliding, and more accurately represent the sub-grid warm-based ice fraction in a grid cell and therefore more accurately represent sliding onset for coarse grid resolutions ($Q_5$ in Sec. 1.3). However, the shape of such a basal temperature ramp is not well constrained. In the GSM, the basal temperature ramp is incorporated into a Weertman-type power law

$$\boldsymbol{u_b} = C_b |\boldsymbol{\tau_b}|^{n_b - 1} \boldsymbol{\tau_b} \tag{6a}$$

as a dependence of the basal sliding coefficient $C_b$ on the estimated warm-based fraction of a grid cell (indirectly accounting
for sub-temperate sliding) $F_{warm}$ (Eq. (8))

$$\boldsymbol{C_b} = (1 - F_{warm}) C_{froz} + F_{warm} \boldsymbol{C}, \tag{6b}$$

where $\boldsymbol{u_b}$ is the basal sliding velocity, $\boldsymbol{\tau_b}$ the basal stress, $n_b$ the bed power strength (Table 1), and $\boldsymbol{C}$ the fully warm-based sliding coefficient (depends on the bed properties, see also Fig. S4). $C_{froz}$ is the fully cold-based sliding coefficient for numerical regularization:

$$C_{froz} = 2 \cdot 10^{-3} \text{ m yr}^{-1} \left(5 \cdot 10^{-6} \text{ Pa}^{-1}\right)^{n_b}. \tag{7}$$

$F_{warm}$ is calculated according to

$$F_{warm} = \max\left[0, \min\left(1, \frac{T_{bp,I} + T_{ramp}}{T_{ramp}}\right)\right]^{T_{exp}}, \tag{8}$$

where $T_{bp,I}$ is the grid cell interface basal temperature relative to the pressure melting point, negative $T_{ramp}$ the temperature below which the entire grid cell is cold-based, and $T_{exp}$ the exponent used for the ramp. The values used in previous GSM
modeling studies ($T_{ramp} = 1.0\,°\text{C}$ and $T_{exp} = 28$ (e.g., Bahadory and Tarasov, 2018)) were based on horizontal basal temperature gradients around the basal sliding activation zone with consideration of the sub-grid warm-based connectivity between grid cell interfaces (as basal sliding requires a connected sub-grid warm-based path). Different values for $T_{ramp}$ and $T_{exp}$ are explored within this paper. $T_{ramp}$ can be chosen as either constant or depending on the horizontal grid resolution (res, equal extent in x- and y-direction):

$$T_{ramp} = P_{T_{ramp}} \cdot \frac{\text{res}}{50 \text{ km}}\,°\text{C} \tag{9}$$

This choice of resolution dependence leads to a sharper temperature ramp for higher horizontal grid resolutions. The parameter $P_{T_{ramp}}$ is used to conduct experiments with different temperature ramps at the same horizontal grid resolution (Sec. 3.3.3). The

temperature ramps for all 4 horizontal grid resolutions and $P_{T_{ramp}} = 1$ (default value) are shown in Fig. 2. For comparison, a temperature ramp similar to the one suggested by Fowler (1986) and later Mantelli et al. (2019)

$$F_{warm} = \exp\left(\frac{T_{bp,I}}{\delta}\right) \quad \text{for} \quad T_{bp,I} \leq 0 \tag{10}$$

is shown for $\delta = 0.01$, where $\delta$ is a parameter controlling the width of the transition zone. Based on experiments conducted by Barnes et al. (1971), Mantelli et al. (2019) expect $\delta$ to be small.

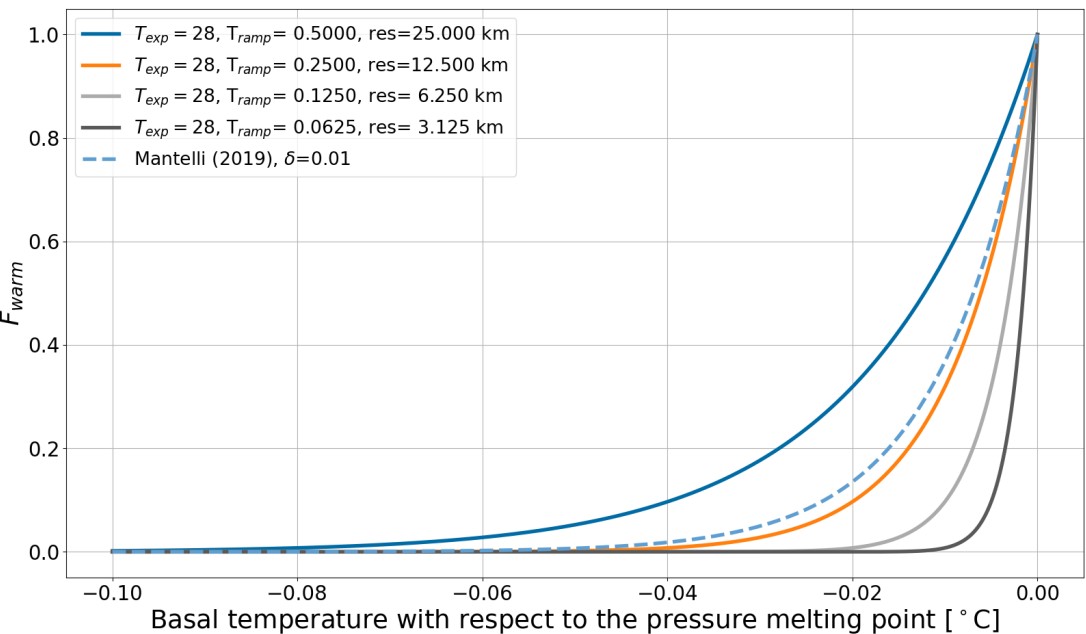

**Figure 2.** Temperature ramps for different values of $T_{ramp}$ which depend on the horizontal grid resolution. A temperature ramp similar to the one suggested by Mantelli et al. (2019) (Eq. (10)) is shown for $\delta = 0.01$.

### 2.1.2 GSM ensemble input parameter vectors

Each GSM experiment is run with an ensemble based on 5 input parameter vectors. The current idealized setup encompasses
a maximum of 8 input parameters (Table 1) per parameter vector. The 5 parameter vectors used in this study are hand-picked from an exploratory ensemble (Fig. S3). The criteria for these 5 parameter vectors was the highest subset variance in surge characteristics and soft bed sliding law exponent. Note that the soft and hard bed sliding law exponents in this study are equal ($n_b$ in Table 1). Due to the significantly increased model run time, exponents larger than 3 are not considered here. To isolate interactions, the GSM reference setup used in this paper does not incorporate basal hydrology and glacial isostatic adjustment

(GIA). Processes associated with basal hydrology, such as lubrication of the bed and decoupling of the ice sheet from the bed, are likely to have a major effect on surge patterns. To determine the impact of these effects, we run the GSM with local basal hydrology enabled (Eq. (20) to (22), Sec. 3.3.5) and examine resolution scaling (Sec. S9.2). However, experiments done with and without basal hydrology lead to qualitatively similar results (e.g., same conclusions from upscaling experiments in Sec. 3.3.3). We therefore omit sub-glacial hydrology coupling for the main analysis.

| Category | Parameter | Description | Range | Unit |
|---|---|---|---|---|
| **Ensemble parameter - ISM** | $C_{\mathrm{rmu}}$ | soft bed sliding coefficient | 0.3 - 1 | |
| | $C_{\mathrm{fslid}}$ | hard bed sliding coefficient | 0 - 3 | |
| | lapsr | atmospheric lapse rate | $-5$ - $-10$ | $^\circ\mathrm{C\,km^{-1}}$ |
| | PDDmelt | melt per Positive Degree Day (PDD) | 0.005 - 0.012 | $\mathrm{m\,PDD^{-1}\,(^\circ C)^{-1}}$ |
| | hpre | precipitation coefficient | 0.02 - 0.2 | $(^\circ\mathrm{C})^{-1}$ |
| | PrecRef | precipitation coefficient | 1 - 3 | $\mathrm{m\,yr^{-1}}$ |
| | rTsurf | domain wide surface temperature constant | $-9$ - $-15$ | $^\circ\mathrm{C}$ |
| | $n_b$ | soft and hard bed sliding law exponent, bed power strength | 1 - 3 | |
| **Hydrology parameters** | $h_{\mathrm{wb,Crit}}$ | effective bed roughness scale (Eq. (20)) | 0.01 - 1 | m |
| | rBedDrainRate | constant bed drainage rate | 0.001 - 0.01 | $\mathrm{m\,yr^{-1}}$ |
| | $N_{\mathrm{eff,Fact}}$ | effective pressure factor (Eq. (22)) | $2 \cdot 10^4$ - $2 \cdot 10^5$ | Pa |
| **Additional parameters** | $P_{Tramp}$ | basal temperature ramp scaling factor (Eq. (9)) | 0.125 - 16 (**1**) | |
| | $T_{ramp}$ | basal temperature (with respect to the pressure melting point) at which sub-temperate sliding becomes important (Eq. (8), (9)) | 0.03125 - 1 (**0.0625**) | $^\circ\mathrm{C}$ |
| | $T_{exp}$ | basal temperature ramp exponent (Eq. (8)) | 5 - 56 (**28**) | |
| | $W_{Tb,\min}$ | weight of adjacent minimum basal temperature for basal sliding temperature ramp (Eq. (S1)) | 0.0 - 1.0 (**0.5**) | |

**Table 1.** Model parameters are listed with respect to their purpose/category. Ice Sheet Model - ISM. Hydrology parameters used when running the GSM with local basal hydrology. Additional (non-regular) input parameters that are usually set to a fixed value. The default values of the 3.125 km horizontal grid resolution reference setup are shown in the brackets for the additional parameters.

### 2.1.3 GSM model setups


The reference setup (Table 2) has a 3.125 km horizontal grid resolution and 1 year maximum time step. The bed topography is flat (at sea level) and an asymmetric temperature forcing is used (Fig. S1). For the sake of generality, we chose a flat topography for the reference setup, while the effect of a basal trough is investigated at a later stage (Sec. 3.3.4). Branching off this reference set-up, we carry out one-at-a-time sensitivity experiments to isolate numerical and process impacts. These experiments, in turn, examine the response to: 3 numerical aspects related to the MNEEs, 4 model aspects affecting the thermal onset of basal sliding,


a change in sediment cover, a non-flat topography, the addition of local basal hydrology, different horizontal grid resolutions [25 km, 12.5 km, 6.25 km], and different maximum time steps [0.5 year, 0.25 year]. The 3 numerical aspects are stricter numerical convergence criteria, the addition of surface temperature noise ($\pm0.1°$C and $\pm0.5°$C), and an approximate implicit time-step coupling between the thermodynamics and ice dynamics. The 4 thermal model aspects are switching to thin (20 m)

bed thermal model, different approaches to determining the basal temperature at the grid cell interface, different weights of the adjacent minimum basal temperature for the basal sliding temperature ramp ($W_{Tb,\min}$), and different basal temperature ramps ($T_{ramp}$ and $T_{exp}$) for thermal activation of basal sliding. See Table 1 for details on parameter ranges.

## 2.2 PISM

### 2.2.1 PISM model description

In contrast to the GSM, the Parallel Ice Sheet Model (PISM) is not specifically developed for glacial cycle ensemble modeling. Therefore, the two models use distinct sets of numerical optimizations for computational speed. To minimize the model dependency of our analysis, experiments are also carried out with v2.0.2 of the PISM. Note that these experiments are not intended to be a direct comparison of the two models, but rather to show that the same conclusions can be achieved with different models (despite their differences in model setups, physics and numerics).

Similar to the GSM, PISM is a 3D thermodynamically-coupled ice sheet model and the SSA is used as a 'sliding law' once the sliding velocity exceeds $100$ m yr$^{-1}$. For further details on the model itself, refer to Bueler and Brown (2009); Winkelmann et al. (2011). The details on the default PISM setup, together with the default GSM values, are listed in Table 2. Given the higher computational cost of the PISM experiments, the relatively high sensitivity of PISM to the number of parallelized cores for these experiments (Table 6), and run time limitations of the computational cluster, the reference setup is run at $25$ km horizontal

grid resolution.

For stability reasons, the PISM adaptive time stepping ratio (used in the explicit scheme for the mass balance equation) was reduced to 0.01 when using small till friction angles (Constantine Khrulev, personal communication).

The default sliding law in PISM is a purely-plastic (Coulomb) model where

$$|\boldsymbol{\tau_b}| \leq \tau_c \quad \text{and} \quad \boldsymbol{\tau_b} = -\tau_c \frac{\boldsymbol{u}}{|\boldsymbol{u}|} \quad \text{if} \quad |\boldsymbol{u}| > 0. \tag{11}$$

Therefore, the basal shear stress $\boldsymbol{\tau_b}$ can never exceed the yield stress $\tau_c$, and basal sliding only occurs when $\boldsymbol{\tau_b}$ reaches $\tau_c$.

### 2.2.2 PISM ensemble input parameter vectors

The PISM configuration encompasses 6 model input parameters (Table 3). These parameters define the input fields for surface temperature, surface accumulation, and till friction angle. Similar to Calov and Greve (2006), the surface temperature at every grid cell is calculated as

$T_{surf} = T_{min} + S_t \cdot d^3,$ \hfill (12)

| Setup component | GSM | PISM |
|---|---|---|
| horizontal grid resolution | 3.125 km x 3.125 km | 25 km x 25 km |
| number of grid cells | 160 x 160 | 120 x 120 |
| model domain | 500 km x 500 km | 3000 km x 3000 km |
| vertical layers | 65 | 60 |
| run time | 200 kyr | 200 kyr |
| maximum time step | 1 yr | 1 yr |
| number of cores/processes | 1 | 8 |
| ice dynamics | hybrid SIA/SSA | hybrid SIA/SSA (maximum SIA diffusivity of $1000 \, \mathrm{m^2 \, s^{-1}}$) |
| sliding law | Weertman-type power law (Eq. (6a)) | Coulomb friction law (Eq. (11)) |
| bed topography | flat (at sea level) | flat (at sea level) |
| bed thermal model | 1 km deep (17 non-linearly-spaced levels) | 1 km deep (20 equally-spaced levels) |
| basal hydrology | not included | local basal hydrology model based on an undrained plastic bed model (Tulaczyk et al., 2000a) |

**Table 2.** Comparison between the GSM and PISM reference setup.

where $S_t$ represents the horizontal surface temperature gradient, $d$ the distance from the domain center ($x_{center}$,$y_{center}$) in km, defined as:

$$d = \sqrt{(x - x_{center})^2 + (y - y_{center})^2} < R,\tag{13}$$

and $R$ denotes the radius and sets an upper limit for $d$. A comparable equation is used to calculate the surface mass balance
(accumulation/ablation) rate input field.

$$B_{surf} = B_{max} - S_b \cdot d^5,\tag{14}$$

where $S_b$ is the horizontal surface mass balance gradient. The input field for the till friction angle is defined by simple grid assignment and a somewhat smoothed transition between the soft and hard bed region. Input fields for one parameter vector are shown for surface temperature, surface accumulation, and till friction angle in Fig. S6, S7, and S8, respectively.

The 6 model ensemble parameters (Table 3) were selected via Latin Hypercube sampling. After sieving an ensemble of 100 runs for those that show oscillatory behavior, a 9-member high-variance (with respect to the surge characteristics) subset was extracted by visual identification (Fig. S10). Each PISM experiment is run with an ensemble based on these 9 input parameter vectors.

| Category | Parameter | Description | Range | Unit |
|---|---|---|---|---|
| **Ensemble parameters** | $soft$ | soft bed till friction angle | 0.5 - 12.0 | $\circ$ |
| | $hard$ | hard bed till friction angle | 15.0 - 30.0 | $\circ$ |
| | $B_{max}$ | maximum surface mass balance (accumulation/ablation) rate | 50 - 450 | $\text{kg m}^{-2}\,\text{yr}^{-1}$ |
| | $S_b$ | horizontal surface mass balance gradient | $(0.15 - 1.00)\cdot 10^{-11}$ | $\text{kg m}^{-2}\,\text{yr}^{-1}\,\text{km}^{-5}$ |
| | $T_{min}$ | minimum surface temperature | 220 - 245 | K |
| | $S_t$ | horizontal surface temperature gradient | $(0.10 - 1.0)\cdot 10^{-8}$ | $\text{K km}^{-3}$ |
| **Constant parameters** | $x_{center}$ | location of the domain center in x-direction | 1500 | km |
| | $y_{center}$ | location of the domain center in y-direction | 1500 | km |
| | $R$ | maximum radius of the domain | 1500 | km |

**Table 3.** Parameters used to generate the PISM input fields.

### 2.2.3 PISM bed properties

A PISM ensemble parameter restriction arose as experiments carried out with PISM only show oscillatory behavior for small yield stresses $\tau_c$. This can be achieved by either a small till friction angle $\Phi$ or low effective pressure on the till ($N_{\text{till}}$) (Bueler and Van Pelt, 2015):

$$\tau_c = c_0 + \tan\left(\Phi\right) N_{\text{till}}, \tag{15}$$

where $c_0 = 0$ Pa is the till cohesion (Tulaczyk et al., 2000b). $N_{\text{till}}$ is given by

$$N_{\text{till}} = N_0 \left(\frac{\delta_e P_0}{N_0}\right)^s 10^{\left(\frac{e_0}{C_c}\right)(1-s)}, \tag{16}$$

where $N_0 = 1$ kPa is the reference effective pressure, $e_0 = 0.69$ the void ratio at $N_0$, $C_c = 0.12$ the dimensionless coefficient of compressibility, $\delta_e$ the effective fraction of the overburden pressure, $P_0$ the ice overburden pressure, and $s$ the ratio $\frac{W_{\text{till}}}{W_{\text{till}}^{\max}}$ (Tulaczyk et al., 2000b; Bueler and Van Pelt, 2015). $W_{\text{till}}$ and $W_{\text{till}}^{\max} = 2$ m are the effective and maximum thickness of water in the till, respectively. The values listed here are the PISM defaults. $C_c$ is on the lower end of measured values (Tulaczyk et al.,

2000b) with significantly larger (up to 17) values reported (Sauer et al., 1993; Mitchell and Soga, 2005). $e_0$ can vary between 0.45 (Tulaczyk et al., 2000b) and approximately 4 (Fig. 10.2 in Mitchell and Soga, 2005). The default value of $\delta_e$ is based on Greenland and Antarctic model runs, but $\delta_e$ is generally considered as a tuning parameter to match observed surface velocities, which are not available in a paleo context (Andy Aschwanden, personal communication).

When only changing the till friction angle, oscillations do not occur unless $\Phi < 1°$ (Fig. S13). This is well below the

measured values of about $10$ to $40°$ (K.M. Cuffey and W.S.B. Paterson., 2010). However, similar oscillatory results are obtained for till friction angles between 5 and $10°$ when slightly adjusting the values of $C_c = 0.2$, $e_0 = 0.6$, and $\delta_e = 0.01$ to favor sliding

(compare Fig. S11 and S12). These values are all well within the ranges set by laboratory measurements. For convenience, we decide to vary only the till friction angle between $0.5$ and $1°$, for which PISM shows oscillatory behavior, and otherwise use the PISM default values.

These resulting very slippery beds enabled occasional maximum sliding velocities of up to $\sim 600 \text{ km yr}^{-1}$ in the simulations (Fig. S11, Sec. S2.3). For comparison, observed outlet glacier velocities at Jakobshavn Isbræ (Greenland) approach $20 \text{ km yr}^{-1}$ (Joughin et al., 2012, 2014). Similar to the GSM, we, therefore, set an upper limit of $40 \text{ km yr}^{-1}$ for the SSA velocity.

### 2.2.4   PISM model setups

As for the GSM, we carry out one-at-a-time sensitivity experiments branching off the PISM reference set-up (Table 2) for all 9
parameter vectors. These experiments, in turn, examine the response to: 2 numerical aspects related to the MNEEs, removing the bed thermal model, an abrupt sediment transition zone, a non-flat topography (Fig. S9), a mass-conserving horizontal transport model for basal hydrology (Bueler and Van Pelt, 2015), different horizontal grid resolutions (50 km, 12.5 km), and a different maximum time step (0.5 and 0.25 year). The 2 numerical aspects are different number of cores/processes ($n = 2, 4, 16, 32$), and addition of surface temperature noise ($\pm 0.1°$C and $\pm 0.5°$C).

## 2.3   Run analysis approach

For both models, we use the Python module *scipy* (version 1.5.2 on GSM cluster and 1.7.0 on PISM cluster, different versions due to the availability on computational clusters) and its built-in function *scipy.signal.find_peaks* on the ice volume output to determine the surge characteristics. The standard output time steps in the GSM and PISM are 0.1 and 1 kyr, respectively. Note that these time steps might not exactly capture the minimum ice volume but are generally a good compromise between storage
requirements and temporal resolution (e.g., Fig. S16 and S17). The Python analysis scripts are provided as supplementary material.

**Surge characteristics**

The quantities being analyzed are: the number of surges, the surge duration, the ice volume change during a surge, and the period between surges (Fig. 3). The surge time is defined as the time of minimum (pseudo-Hudson Strait) ice volume, and
the duration of a surge includes the surge itself as well as the time it takes the ice sheet to recover approximately half the ice volume lost during the surge (as per *scipy.signal.peak_widths*). The calculated ice volume change is the difference between the pre-surge and minimum (pseudo-Hudson Strait) ice volume in that particular surge (as per *scipy.signal.peak_prominences*). The period between surges is the time span between two subsequent occurrences of minimum (pseudo-Hudson Strait) ice volume (not defined for the very last event). The spin-up interval (first 20 kyr of every run) is not incorporated in the analysis,
and only surges with a (pseudo-Hudson Strait) ice volume change of more than $500 \text{ km}^3$ and $40^4 \text{ km}^3$ are considered in the GSM and PISM analyses, respecitvely ($\sim 5 \%$ of mean ice volume across all runs).

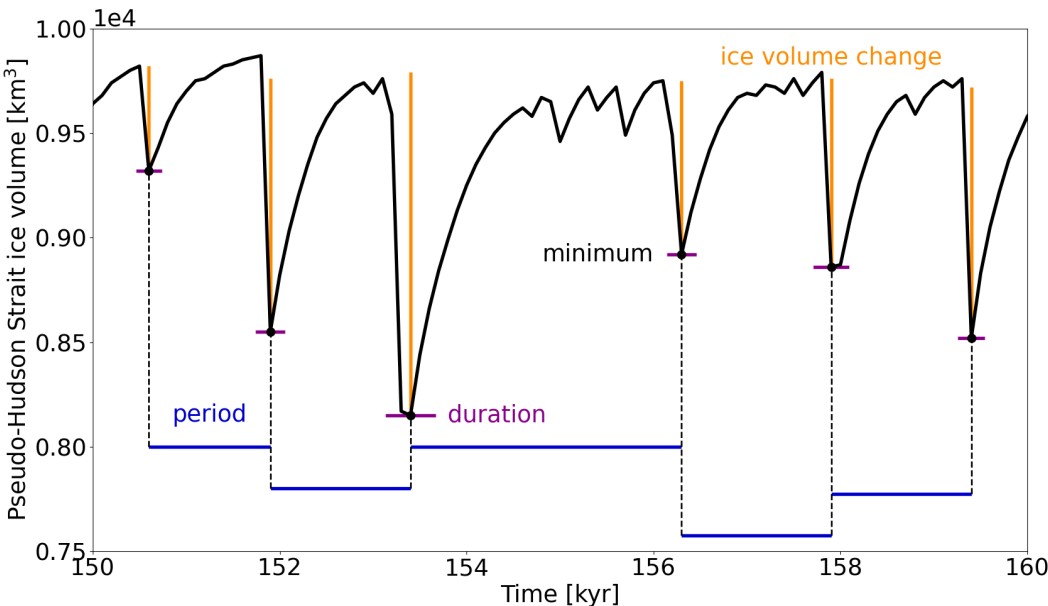

**Figure 3.** Pseudo-Hudson Strait ice volume of a GSM model run with visual illustration of the surge characteristics used to compare different model setups. The horizontal grid resolution is 3.125 km.

In addition to the surge characteristics, the Root Mean Square Error (RMSE) and mean bias are calculated as a percentage deviation from the reference (pseudo-Hudson Strait) ice volume time series for all setups (each parameter individually) and then averaged over the 5 parameter vectors (Eq. (S2) and (S3)). The full run time is considered (no spin-up interval).

**Percentage differences**

We compare different model setups by calculating the percentage difference between the reference setup and all other setups for every parameter vector individually and then average this difference over the 5 parameter vectors. Crashed runs are not considered and runs with less than 2 surges require special treatment (see Sec. S5 for further details on the analysis).

**Surge area**

In the GSM, the whole pseudo-Hudson Strait (Fig. 1) is ice-covered and at maximum ice volume at the beginning of a surge. Surges in the GSM, therefore, consistently appear as ice volume minima, which allows us to directly use the pseudo-Hudson Strait ice volume for the GSM results.

For PISM, a large fraction of the pseudo-Hudson Strait area is only ice-covered when a surge occurs (e.g., Fig. 5), leading to an inconsistency in the surge detection. This issue is addressed by including the ice volume over the eastern half of the

pseudo-Hudson Bay, the area most affected by the surge drained through the pseudo-Hudson Strait. See Sec. S2.4 for further details and a comparison between the two approaches.

**Minimum numerical error estimates**

We compute the new '**M**inimum **N**umerical **E**rror **E**stimates' (MNEEs) metric by examining the model response to changes in the model configuration that are not part of the physical system. The MNEEs are defined as the percentage differences in surge

characteristics when applying a stricter (than default) numerical convergence in the GSM and adjusting the matrix solver used in PISM (changing the number of processor cores used). They are then used as a threshold to determine if model sensitivities to changes in the model configuration that affect the physical system (e.g., the inclusion of a bed thermal model or sliding depence on effective pressure from basal hydrology) are above the level of background noise induced by iterative numerical solvers in the model. We refrain from drawing conclusions about the effects of a change in model configuration with physical

relevance when the model sensitivities in question are smaller than the MNEEs. In these cases, the actual physical response of the model might be hidden within the numerics.

While the MNEEs are useful to our purpose, we wish to emphasize that they can not replace proper model verification and validation and are missing uncertainties due to, e.g., different approximations of the Stokes equations and other physical processes not included in the models. Nonetheless, they provide a minimum estimate of the numerical model error, which is

still a significant improvement over ignoring this issue entirely.

## 3   Results

### 3.1   Key surge characteristics of the reference setup

#### 3.1.1   Surge onset, propagation, and termination

Before analyzing ensemble characteristics, it is crucial to understand how surges initiate, propagate and terminate. Surges in

the GSM originate at the pseudo-Hudson Strait mouth ($x = 450$ km, $y = 225$ to $275$ km) and propagate towards the center of the pseudo-Hudson Bay ($x = 200$ km, $y = 250$ km, Fig. 1 and 4). The surging onset is a complex interplay between heating at the ice sheet bed, basal temperature, and ice sheet velocity. The beginning of a surge is shown in an online video (video 01 of Hank (2023)) and Fig. 4. Just before the start of the surge, the entire South-North extent of pseudo-Hudson Strait grid cells close to the ocean is warm-based. At $t = 6.69$ kyr, the SIA velocities exceed $30$ m yr$^{-1}$ and the SSA is activated

(Sec. 2.1.1). The longitudinal stress gradient and horizontal shear terms provide additional heating (heating due to shelfy stream dynamics in video 01 of Hank (2023)). This leads to several small ice streams with relatively strong heating due to basal sliding ($\sim 10^7$ J m$^{-2}$ yr$^{-1}$) at $t = 6.70$ kyr in the video. This is an order of magnitude larger than heat production from deformation work. The additional heat fosters higher ice velocities, leading to even more heating, the extension of the warm-based area to the West, and therefore the upstream propagation of the small ice streams ($t = 6.71$ kyr). The narrow ice streams draw

in warm-based ice from the surrounding grid cells, increasing the velocities and heat production in the area between the ice

streams. This leads to a merger of the ice streams with now high velocities occurring over the full South-North extend of the pseudo-Hudson Strait ($t = 6.72$ kyr). The warm-based area rapidly extends towards the West due to the strong heating and high ice velocities, causing a pseudo-Hudson Strait surge.

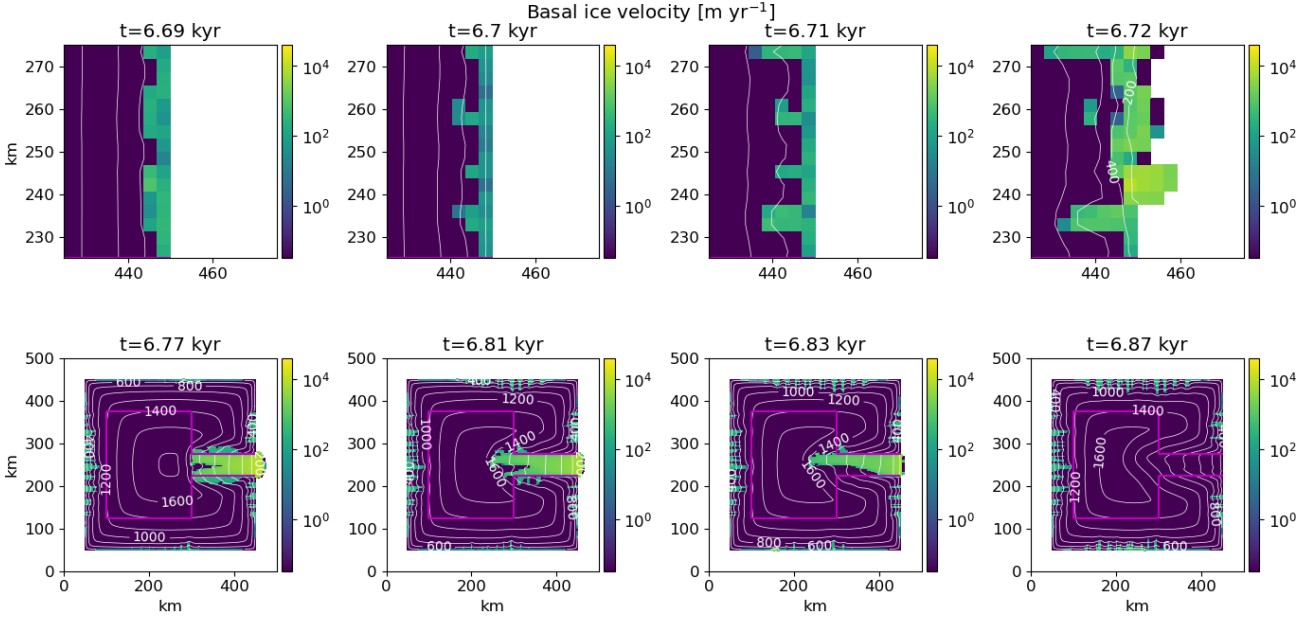

**Figure 4.** Basal ice velocity for parameter vector 1 at different time steps using the GSM. The horizontal grid resolution is 3.125 km and the maximum model time step is 1 yr. The contour lines show the ice sheet surface elevation in m. The magenta line outlines the soft-bedded pseudo-Hudson Bay and Hudson Strait. Note that the top and bottom rows show different areas of the domain, with the top zooming in on the surge onset area.

The surge propagates nearly symmetrically until the pseudo-Hudson Bay area is reached ($t = 6.77$ kyr in Fig. 4 and video
02 of Hank (2023)). After this point, the northern branch of the ice stream propagates more rapidly and extends further to the West than the southern branch. While the smaller southern branch starts to shrink at $t = 6.81$ kyr, the northern part propagates until $t = 6.83$ kyr. At this time, the southern branch has vanished almost completely due to a thinner ice sheet (than at the start of the surge) and the advection of cold ice into the surge area. After $t = 6.83$ kyr, the available heating is no longer sufficient to keep the ice sheet bed at the pressure melting point, and the northern part collapses as well. The surge ends after 150 yr (at
$t = 6.87$ kyr).

Since the GSM setup and climate forcing are symmetric about the horizontal axis in the middle of the pseudo-Hudson Strait ($y = 250$ km in Fig. 1), we interpret the induced asymmetry as a numerical induced bifurcation. We define the asymmetry as positive when the surge is stronger Northward (Fig. 4 and video 02 of Hank (2023)) or shifted Northward. The asymmetry

sign varies across the first surges (i.e., the surge least biased by previous asymmetries) of the 5 reference runs, ruling out any

persistent numerical bias.

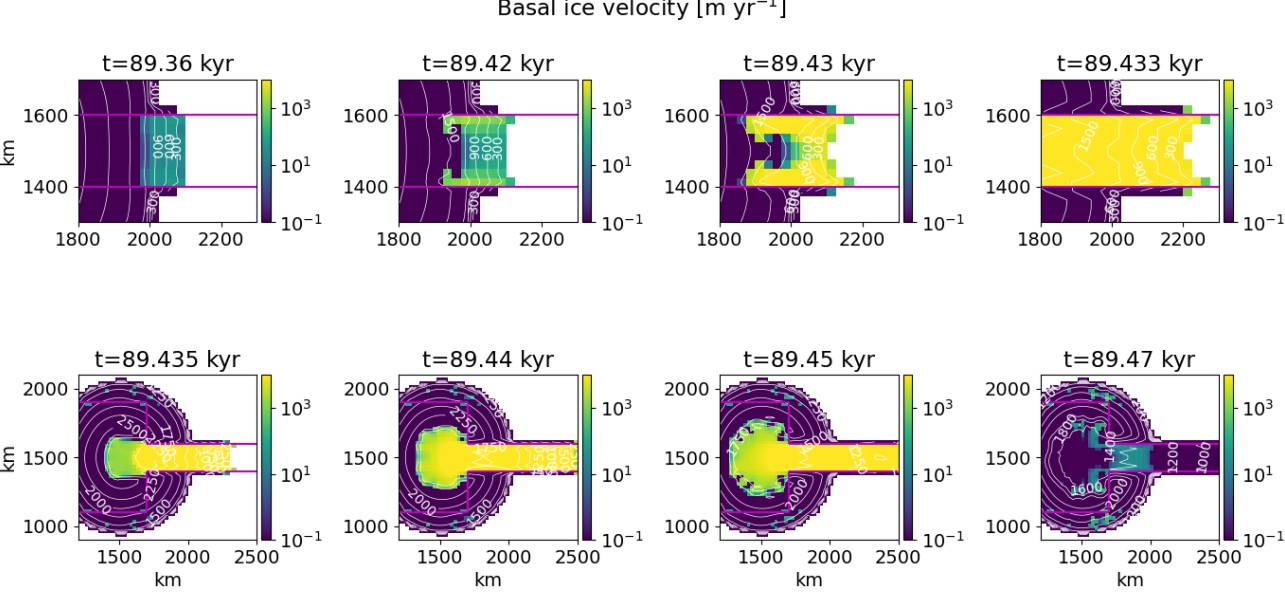

**Figure 5.** Basal ice velocity for parameter vector 8 at different time steps using the PISM. The horizontal grid resolution is 25 km and the maximum model time step is 1 yr. Otherwise as in Fig. 4.

Surges in the PISM originate at the ice sheet margin in the soft-bedded pseudo-Hudson Strait (exact position varies between runs) and propagate towards the center of the pseudo-Hudson Bay ($x = 1300$ km, $y = 1500$ km, Fig. S8 and 5). The ice near the margin is already flowing downstream before the start of the surge ($t = 89.36$ kyr). However, the basal temperature is below the pressure melting point, and the ice velocities are low ($< 100$ m yr$^{-1}$). As the ice sheet upstream of the margin thickens, the

warm-based area extends further downstream, particularly along the 100 % soft-bedded contour line (magenta line in Fig. 5). Once the warm-based area connects with the margin ($t = 89.42$ kyr), the ice velocities increase beyond $100$ m yr$^{-1}$, activating the SSA (Sec. 2.2.1). Similar to the surges in the GSM, the sliding velocities then increase rapidly, quickly extending the warm-based area ($t = 89.43$ kyr and $t = 89.433$ kyr). The surge propagates upstream into the pseudo-Hudson Bay and the ice is transported along the pseudo-Hudson Strait into regions with increasingly negative surface mass balance rates ($t = 89.435$ kyr

to $t = 89.45$ kyr, Fig. S7). The ice sheet thins, the basal temperature at the margin falls below the pressure melting point, blocking parts of the upstream ice stream, and the surge ceases at $t = 89.47$ kyr ($\sim 100$ yr surge duration). The ice volume in the surge-affected area continues to decrease for, on average, another 2.5 kyr due to the large amounts of ice in the negative surface mass balance regions. In contrast to the GSM, the PISM results remain symmetrical about $y = 1500$ km throughout the surge.

### 3.1.2 Surge characteristics of the GSM and PISM reference setup

Due to the differences in model setup, physics, and numerics (Table 2), the GSM and PISM reference setup yield different surge characteristics (Table 4). While resembling the inferred ice-rafted debris (IRD) interval duration as closely as possible is not a goal of this study, the modeled values are in agreement with the literature (200 to 2280 yr (Hemming, 2004)). The mean modeled GSM period is shorter than the observed period of, on average, 7 kyr (K.M. Cuffey and W.S.B. Paterson., 2010). However, exploratory GSM runs with a dimensionally accurate (not downscaled) model domain (but otherwise identical experimental setup) yielded periods within the range of geological inferences. The mean modeled PISM period is within limits set by the literature. The mean (pseudo-Hudson Strait) ice volume change in the GSM corresponds to 15 % of a 1.5 km thick ice sheet covering the downscaled pseudo-Hudson Strait area (150x50 km). In PISM, the mean ice volume change is 7.1 % of the mean (across reference setup runs) maximum ice volume in the eastern half of the pseudo-Hudson Bay and pseudo-Hudson Strait.

| Metric | GSM reference setup | PISM reference setup |
|---|---|---|
| number of surges | $180 \pm 100$ | $35 \pm 25$ |
| mean period | $1.1 \pm 0.5$ kyr | $10 \pm 10$ kyr |
| mean duration | $0.3 \pm 0.1$ kyr | $3 \pm 2$ kyr |
| mean pseudo-Hudson Strait ice volume change | $1.7 \pm 0.2 \cdot 10^3$ km$^3$ | $1.1 \pm 0.3 \cdot 10^5$ km$^3$ |

**Table 4.** Surge characteristics of the GSM ($T_{ramp} = 0.0625°$C, $T_{exp} = 28$ (black line in Fig. 2), $W_{Tb,min} = 0.5$, TpmTrans for the interface calculation, sharp transition between hard and soft bed) and PISM reference setup (Table 2). No runs crashed and all runs had more than 1 surge event. The first 20 kyr of each run are treated as a spin-up interval and are not considered in the above.

### 3.2 Minimum numerical error estimates

Differences in surge characteristics (compared to the reference setup) are considered significant when they exceed the MNEEs given in Table 5 and 6 for the GSM and PISM, respectively. However, this does not necessarily mean that smaller changes have no physical relevance but rather that their interpretation is difficult (if not impossible) because the physical response is hidden within the numerical sensitivities. Likely sources of the MNEEs are the iterative SSA solutions and floating point accuracy.

To determine a minimum significant threshold in the GSM, we re-run a set of GSM runs with 3.125 km horizontal grid resolution, imposing a stricter numerical convergence (decreasing final iteration thresholds). In a second experiment, we additionally increase the maximum iterations from 2 to 3 for the outer Picard loop (ice dynamics) and from 2 to 4 when solving the non-linear elliptic SSA equation. Note that the goal of these experiments is not necessarily to decrease the model error, especially since we do not know the exact solution and, therefore, can not determine the model error. Instead, we aim to show (by changing purely numerical aspects) what the minimum numerical errors are for each surge characteristic.

The largest differences between simulations occur for the mean period (7 %, Table 5) when using stricter convergence thresholds (no change in the maximum number of iterations). The standard deviations are on the same order of magnitude as the values themselves, indicating different responses across the 5 parameter vectors. Note that determining the MNEEs at 12.5 km instead of 3.125 km horizontal grid resolution yields similar results, except for the mean pseudo-Hudson Strait ice volume change (21 %, Table S2).

| Metric | reference setup | stricter numerical convergence [% difference] | stricter numerical convergence with increased maximum iterations [% difference] |
|---|---|---|---|
| number of surges | $180 \pm 100$ | $\mathbf{-4.1 \pm 4.9}$ | $-0.9 \pm 3.6$ |
| mean period | $1.1 \pm 0.5$ kyr | $\mathbf{7.0 \pm 10.6}$ | $4.7 \pm 10.6$ |
| mean duration | $0.3 \pm 0.1$ kyr | $2.5 \pm 3.2$ | $\mathbf{3.9 \pm 4.8}$ |
| mean pseudo-Hudson Strait ice volume change | $1.7 \pm 0.2 \cdot 10^3$ km$^3$ | $-1.1 \pm 3.1$ | $\mathbf{4.6 \pm 4.6}$ |

**Table 5.** Percentage differences of surge characteristics between GSM runs with regular and stricter numerical convergence and increased maximum iterations for the ice dynamics loops at 3.125 km horizontal grid resolution. The values represent the average of 5 parameter vectors. No runs crashed and all runs had more than 1 surge event. The first 20 kyr of each run are treated as a spin-up interval and are not considered in the above. The bold numbers mark the largest MNEE for each surge characteristic.

MNEEs in PISM are determined by comparing runs with different numbers of cores. Although most parameter vectors show similar results at the beginning of the runs, minor differences can slowly accumulate and lead to significant discrepancies in surge activity by the end of the run (Fig. S18). The largest differences occur for the number of surges (16 %) and mean ice volume change (16 %) for nCores= 32, but the standard deviations are large due to a more than $\sim 200$ % increase in both surge characteristics for parameter vector 6.

| Setup | number of surges | mean period | mean duration | mean ice volume change | nS1 |
|---|---|---|---|---|---|
| 25 km reference setup | $35 \pm 25$ | $10 \pm 10$ kyr | $3 \pm 2$ kyr | $1.1 \pm 0.3 \cdot 10^5$ km$^3$ | 0 |
| nCores= 2 | $-7.1 \pm 19.5$ | $6.8 \pm 36.2$ | $-0.4 \pm 9.5$ | $1.5 \pm 10.3$ | 0 |
| nCores= 4 | $-8.2 \pm 22.9$ | $-3.8 \pm 6.6$ | $2.8 \pm 18.3$ | $0.6 \pm 4.8$ | 1 |
| nCores= 16 | $-10.9 \pm 26.0$ | $\mathbf{-8.2 \pm 14.7}$ | $7.6 \pm 21.2$ | $-0.7 \pm 13.3$ | 1 |
| nCores= 32 | $\mathbf{16.0 \pm 56.2}$ | $6.9 \pm 48.5$ | $\mathbf{-8.0 \pm 17.4}$ | $\mathbf{16.3 \pm 35.1}$ | 0 |

**Table 6.** Percentage differences of surge characteristics (except first row) between the PISM reference setup and setups with different numbers of cores at 25 km horizontal grid resolution. The values represent the average of 9 parameter vectors. No runs crashed and all runs showed at least 1 event. Runs with just one surge (nS1) are ignored when calculating the change in mean period. The first 20 kyr of each run are treated as a spin-up interval and are not considered in the above. The bold numbers mark the largest MNEE for each surge characteristic.

The differences in surge characteristics between different numbers of cores can be minimized by decreasing the relative Picard tolerance in the calculation of the vertically-averaged effective viscosity ($10^{-4}$ to $10^{-7}$) and the relative tolerance for the Krylov linear solver used at each Picard iteration ($10^{-7}$ to $10^{-12}$, Table S5 and Fig. S19). However, this leads to an unreasonable increase in model run time ($\sim 300 \%$) that is not feasible for an ensemble-based approach (more than $50 \%$ of all runs did not finish within the time limit of the computational cluster). Intermediate decreases in the relative tolerances still lead to significant differences in surge characteristics while increasing the model run time and are, therefore, not used in the PISM reference setup.

### 3.2.1 Adding surface temperature noise

Low levels of surface temperature noise have previously been shown to cause chaotic behavior in the mean periods of oscillations (Souček and Martinec, 2011). Adding low levels of uniformly distributed surface temperature noise ($\pm 0.1°$C and $\pm 0.5°$C) to the climate forcing does not significantly affect the surge characteristics for the GSM (Table S3). For example, the effect of adding $\pm 0.5°$C surface temperature noise on the mean period is only $4 \%$ (compared to the $\sim 20 \%$ for $\pm 0.01°$C reported by Souček and Martinec (2011)). Adding the same levels of uniformly distributed surface temperature noise to PISM increases the mean duration by $12 \%$ (for $\pm 0.1°$C), but has no significant effect on the other surge characteristics (Table S6).

### 3.2.2 Implicit thermodynamics/ice dynamics coupling

In contrast to the commonly used explicit time step coupling between the thermodynamics and ice dynamics in glaciological ice sheet models, we test the impact of approximate implicit time step coupling via an iteration between the two calculations for each time step. The implicit coupling decreases the mean duration and pseudo-Hudson Strait ice volume change ($-13 \%$ and $-25 \%$, respectively). The number of surges and mean period show no significant change (Table S4). While the changes in mean duration and pseudo-Hudson Strait ice volume change are larger than the MNEEs, they do not justify an increase in run time of $\sim 265 \%$ and the implicit coupling is therefore omitted for the GSM reference setup.

## 3.3 Sensitivity experiments

Here we discuss differences in surge characteristics due to changes in the model setup. An overview of the results can be found in Fig. 6 and 7 for the GSM and PISM, respectively. The exact values of the percentage differences and information on crashed runs or runs without oscillations are provided in the supplement. We first examine the 4 model aspects affecting the thermal activation of basal sliding (Sec. 3.3.1 to 3.3.3), followed by the analysis of a smooth sediment transition zone, non-flat topography, and local basal hydrology (Sec 3.3.4 and 3.3.5). Experiments without significant differences in the surge characteristics are only briefly mentioned here (Sec. 3.3.6). A more in-depth discussion of these latter experiments is available in the supplement.

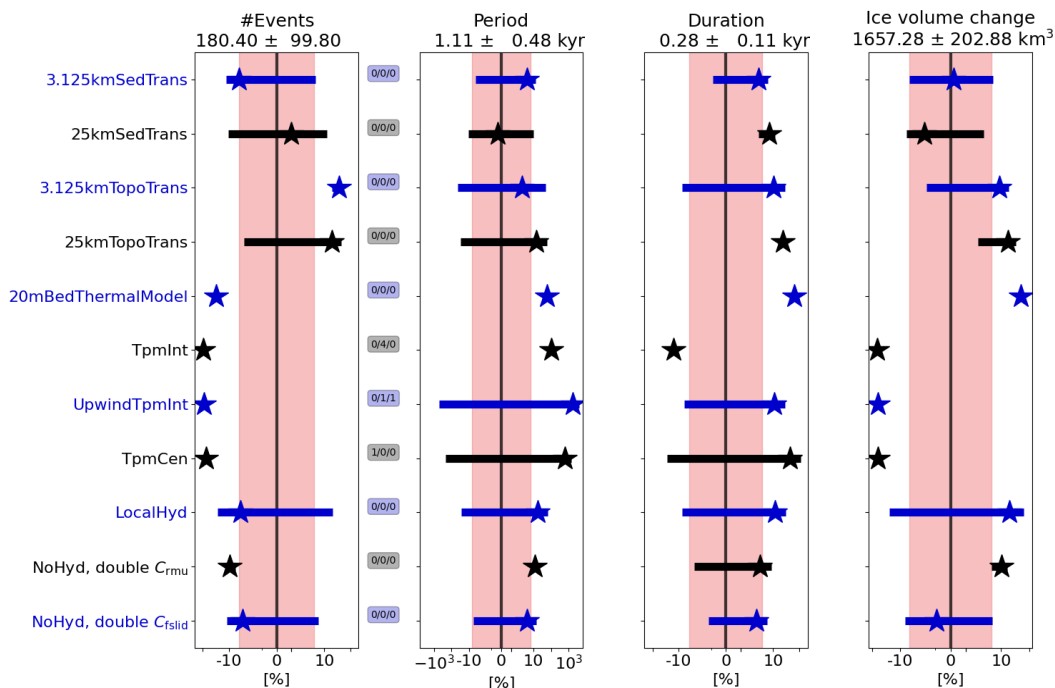

**Figure 6.** Percentage differences in surge characteristics compared to the GSM reference setup for model setups discussed in Sec. 3.3 (average of the 5 parameter vectors). The horizontal grid resolution is 3.125 km. The different colors were added for visual alignment of the individual model setups, the stars are the ensemble mean percentage differences, and the horizontal bars represent the ensemble standard deviations. The shaded pink regions mark the MNEEs (Table 5) and the black numbers in the title of each subplot represent the mean values of the reference setup. The 3 small numbers between the first two columns represent the number of crashed runs (nC), the number of runs without a surge (nS0), and the number of runs with only one surge (nS1), respectively. The first 20 kyr of each run are treated as a spin-up interval and are not considered in the above. The x-axis is logarithmic. Further details of each individual experiments are provided in the subsequent sections and the supplement. The model setups, from top to bottom, are: 3.125 km wide sediment transition zone (instead of an abrupt transition in the reference setup), 25 km wide sediment transition zone, 3.125 km wide sediment transition zone with pseudo-Hudson Bay/Hudson Strait topography (instead of a flat topography in the reference setup), 25 km sediment transition zone with pseudo-Hudson Bay/Hudson Strait topography, 20 m deep (1 layer) bed thermal model (instead of a 1 km deep bed thermal model (17 non-linearly-spaced layers) in the reference setup), 3 different approaches to calculate basal grid cell interface temperature (TpmInt, upwind TpmInt, TpmCen), local hydrology (instead of no hydrology), and doubling the values of the soft and hard bed sliding coefficients (as an attempt to represent basal hydrology without actually adding it).

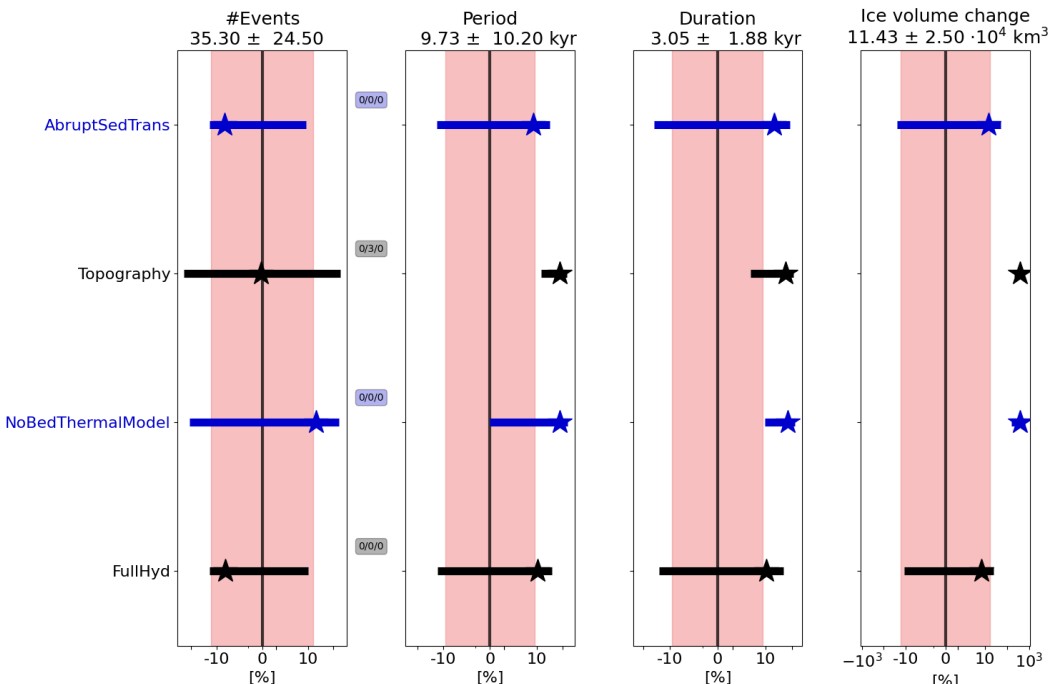

**Figure 7.** Percentage differences in surge characteristics compared to the PISM reference setup for model setups discussed in Sec. 3.3 (average of the 9 parameter vectors). Otherwise same as Fig. 6. The model setups, from top to bottom, are: abrupt sediment transition (instead of the transition shown in, e.g., Fig. S8), pseudo-Hudson Bay/Hudson Strait topography (instead of a flat topography in the reference setup, Fig. S9), no bed thermal model (instead of a 1 km deep bed thermal model (20 equally-spaced layers) in the reference setup), and a mass-conserving horizontal transport model for basal hydrology (instead of a local hydrology).

### 3.3.1 Bed thermal model

First, we examine the effects of a 1 km deep bed thermal model on the basal temperature and the surge characteristics in the GSM as well as PISM. Both models show significant differences when limiting the bed thermal model to one layer (GSM) or removing it entirely (PISM).

Advection of cold ice near the end of a surge rapidly decreases the basal ice temperature and, therefore, increases the temperature gradient between the basal ice and the bed. In GSM runs with the 1 km deep (17 non-linearly-spaced levels) bed thermal model (reference setup), this stronger gradient increases the heat flux from the bed into the ice and dampens the actual change in basal ice temperature. Similarly, a rapid increase in basal ice temperature due to higher basal ice velocities at the beginning of a surge reverses the existing temperature gradient at the base of the ice sheet, leading to a heat flux from the ice

into the bed. Consequently, less heat is available to warm the surrounding cold-based ice, counteracting the surge propagation (Fig. 8).

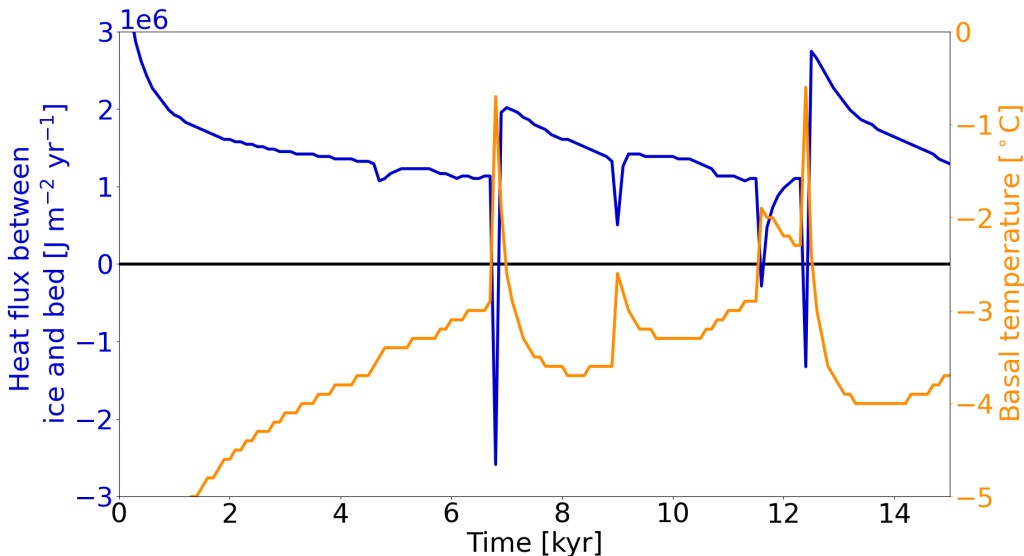

**Figure 8.** Heat flux at the base of the ice sheet (positive from bed into ice) and basal ice temperature for a grid cell in the center of the pseudo-Hudson Strait (grid cell center at $x = 376.5625$ km and $y = 248.4375$ km, white star in Fig. 1) and parameter vector 1 with the 1 km deep bed thermal model (17 non-linearly-spaced levels) using the GSM. The horizontal grid resolution is 3.125 km.

With only one bed thermal layer (20 m deep, removing most of the heat storage), the variance of the average basal temperature with respect to the pressure melting point in the pseudo-Hudson Strait increases (Fig. S20) and more heat is available to warm the surrounding ice (no or smaller heat flux into the bed, Fig. S21). The additional heat increases the mean pseudo-
550 Hudson Strait ice volume change and duration (50 % and 65 %, respectively, Fig. 6). Due to the larger changes in pseudo-Hudson Strait ice volume and average basal temperature with respect to the pressure melting point, the ice sheet requires more time to reach the pre-surge state when only one bed thermal layer is used. Therefore, the period increases (60 %) while the number of surges drops. These differences in surge characteristics exceed the MNEEs (Table 5). The stronger surges (larger pseudo-Hudson Strait ice volume change) lead to overall less ice volume in the pseudo-Hudson Strait (Table S7).
Running PISM without the 1 km deep (20 equally-spaced levels) bed thermal model yields similar behavior as the GSM, further underlining the impact of a bed thermal model. The mean period, mean duration, and mean ice volume change all increase (80 %, 70 %, and 396 %, respectively; Fig. 7). In contrast to the GSM characteristics, the number of surges increases for runs without a bed thermal model. However, the standard deviation is large and the change in the number of surges is somewhat misleading. The number of surges decreases for 6 out of 9 runs. Parameter vectors showing an increase in the
number of surges without a bed thermal model show very few surges (e.g., Fig. S22) or transition to a constantly active ice stream when the bed thermal model is included. As for the GSM, the stronger surges lead to an overall smaller ice sheet in the surge affected-area (Table S8).

### 3.3.2 Basal temperature at the grid cell interface

Another modeling choice that affects the thermal activation of basal sliding is the approach to determining the basal temperature at the grid cell interface. The most straightforward approach to determining the basal temperature with respect to the pressure melting point at the grid cell interface ($T_{bp,I}$) is to use the mean of the two adjacent basal **T**emperatures with respect to the **p**ressure **m**elting point at the grid cell **Cen**ters (TpmCen).

$$T_{bp,I} = 0.5 \cdot (T_{bp,L} + T_{bp,R}), \tag{17}$$

where $T_{bp,L}$ and $T_{bp,R}$ are the grid cell center basal temperatures with respect to the pressure melting point to the left and right of the interface, respectively. Similarly for upper and lower grid cells adjacent to a horizontally aligned interface. However, this approach does not explicitly account for ice thickness changes at the grid cell interface.

TpmInt, on the other hand, calculates the basal temperature at the **Int**erface ($T_I$) by averaging the adjacent grid cell center basal temperatures ($T_L$ and $T_R$, Eq. (18a)). $T_{bp,I}$ is then determined by using the interface ice sheet thickness (average of adjacent grid cell center ice thicknesses $H_L$ and $H_R$, Eq. (18b)).

$$T_I = 0.5 \cdot (T_L + T_R) \tag{18a}$$

$$T_{bp,I} = T_I + \beta_P \frac{H_L + H_R}{2}, \tag{18b}$$

where $\beta_P = 8.7 \cdot 10^{-4}\,°\mathrm{C\,m^{-1}}$ is the standard basal melting point depression coefficient. When TpmInt is used with the upwind scheme and the basal ice velocity exceeds 20 m yr$^{-1}$, Eq. 18a is replaced by $T_I = T_{up}$, where $T_{up}$ is the upstream adjacent grid cell center basal temperature.

The last approach (TpmTrans) attempts to represent heat transfer from sub-glacial hydrology and ice advection by accounting for extra warming above the pressure melting point given by

$$T_{\mathrm{add}} = M_b \cdot \frac{L_H}{c_H} \cdot \frac{1}{H_b} \cdot \Delta t, \tag{19a}$$

where $M_b$ is the basal mass balance in m yr$^{-1}$ (positive for melt), $L_H = 3.35 \cdot 10^5$ J kg$^{-1}$ the specific latent heat of fusion of water/ice, $c_H = 2097$ J kg$^{-1}$ K$^{-1}$ the heat capacity of ice at 273.03 K, $H_b$ the basal ice layer thickness in m, and $\Delta t$ the current model time step in yr. In an intermediate calculation step, the temporary basal temperature at the grid cell center $T_{Im,C}$ is calculated by accounting for the additional heating $T_{\mathrm{add}}$

$$T_{Im,C} = T_C + T_{\mathrm{add}}, \tag{19b}$$

where $T_C$ is the basal temperature at the grid cell center. The basal temperature with respect to the pressure melting point at each adjacent grid cell center $T_{bp,Im,C}$ is then calculated using the interface ice thickness.

$$T_{bp,Im,C} = T_{Im,C} + \beta_P \frac{H_L + H_R}{2} \tag{19c}$$

In the intermediate steps to calculate the interface temperature (Eq. (19b) and (19c)), $T_{Im,C}$ and $T_{bp,Im,C}$ are allowed to exceed the pressure melting point. This temporary higher basal temperature is an attempt to account for heat transported to the interface by ice advection and basal water.

$$\text{IF } T_{bp,Im,C} > 0°\text{C}: \quad T_{bp,Im,C} = \min\left(0.5°\text{C}, 0.5 \cdot T_{bp,Im,C}\right) \tag{19d}$$

Averaging the adjacent basal temperatures with respect to the pressure melting point at the grid cell center ($T_{bp,Im,L}$ and $T_{bp,Im,R}$) yields the final basal temperature with respect to the pressure melting point at the interface ($T_{bp,I}$).

$$T_{bp,I} = 0.5 \cdot \left(T_{bp,Im,L} + T_{bp,Im,R}\right) \tag{19e}$$

Note that neither the grid cell center nor the interface basal temperature may exceed the pressure melting point (only the basal temperature in the intermediate calculation steps).

The GSM reference setup (no hydrology) uses TpmTrans. The additional heat embodied in $T_{\text{add}}$ warms up the grid cell interface. Without the extra warming (TpmInt), 4 out 5 parameter vectors do not show any surges. For the only run that still has cyclic behavior (parameter vector 1), the number of surges decreases by 84 % (note that runs without surges are considered for the number of surges in Fig. 6). Using TpmInt with an upwind scheme leads to slightly more surges (difference of 7 % and, therefore, on the same order of magnitude as the MNEE (4 %, Table 5)). Sporadic surges now occur in all but one run, leading to a large increase in the mean period (1645 %, Fig. 6).

The most straightforward approach, TpmCen, leads to 75 % fewer surges, and an increase in mean period and mean duration (609 % and 43 %, respectively). The mean pseudo-Hudson Strait ice volume change decreases ($-61$ %). Note that the TpmInt, TpmInt uwpind, and TpmCen surge characteristics are difficult to compare due to the different number of runs considered (except for the number of surges, decrease of 97 % vs. 90 % vs. 75 %, respectively). Due to significantly fewer surges, the mean pseudo-Hudson Strait ice volume increases for runs with TpmInt, TpmInt uwpind, and TpmCen (Table S9).

### 3.3.3 Basal temperature ramps at different resolutions

Here we examine the effect of different basal temperature ramps (thermal activation criteria for basal sliding) at 3.125 km horizontal grid resolution and determine ramps for the coarse resolution runs that best match the 3.125 km model results (later used in Sec. 3.4.1). For coarse resolutions, changing the basal temperature ramp can lead to a shift from oscillatory to non-oscillatory behavior (compare 25 km runs in Fig. S23 and 12).

When running the GSM at 3.125 km horizontal grid resolution, surges are apparent for all tested basal temperature ramps. Due to an earlier sliding onset and easier surge propagation, increasing the width of the temperature ramp generally increases the mean pseudo-Hudson Strait ice volume change and duration (Fig. 9). The ice sheet takes longer to recover from the surge (longer regrowth phase), increasing the mean period and decreasing the average number of surges. The largest differences in surge characteristics occur for the widest ramp. Running the GSM without a basal temperature ramp leads to small but significant (according to Sec. 3.2) differences in the mean duration ($-7$ %).

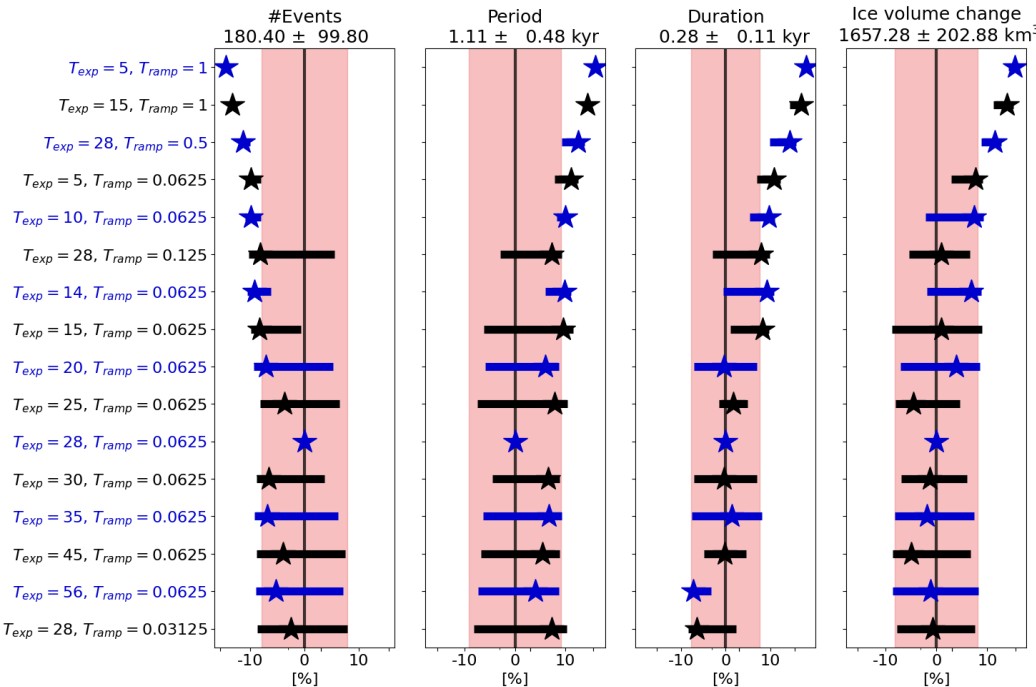

**Figure 9.** Percentage differences in surge characteristics compared to the GSM reference setup ($T_{ramp} = 0.0625$, $T_{exp} = 28$) for different basal temperature ramps at $3.125$ km horizontal grid resolution (average of the 5 parameter vectors). The ramps are sorted from widest (first row) to sharpest (last row, see Fig. S24 for a visualization of all ramps). Otherwise same as Fig. 6. No runs crashed and all runs had more than 1 surge event. The exact values are given in Table S10.

All ramps (wider and sharper than reference setup) show fewer surges and a longer mean period than the reference temperature ramp setup ($T_{ramp} = 0.0625$, $T_{exp} = 28$). However, except for the 5 widest temperature ramps, changes in both surge characteristics are smaller than the MNEEs with standard deviations on the same order of magnitude (Fig. 9). The mean duration and mean pseudo-Hudson Strait ice volume change show a consistent response (increase/decrease for both surge characteristics for a wider/sharper ramp) except for the four basal temperature ramps with the smallest difference to the reference setup (Fig. S24).

Except for the three widest ramps, the mean bias is less than one percent. The RMSE, on the other hand, is roughly $8$ %, indicating that the average pseudo-Hudson Strait ice volume is similar, but the timing of surges varies even for small differences in the width of the ramp (Table S10).

We compare the different temperature ramps at $25$ km, $12.5$ km and $6.25$ km horizontal grid resolution by calculating a single score for the mean and standard deviation of all surge characteristics (Sec. S7.3). The ramps yielding the smallest differences compared to the $3.125$ km reference setup are listed in Table S11 and shown in Fig. S25. These results may be different for a different reference setup (see Table S22 for a comparison of different reference setups with local basal hydrology).

At 25 km horizontal grid resolution, only 3 out of 12 basal temperature ramps remain after removing the ramps for which the sum of scores (score-mean + score-std, last column in Table S11) differs by more than 50 % from the minimum sum

of scores (bold numbers in last column in Table S11). The minimum scores for the mean and standard deviation occur for the same ramp ($T_{exp} = 5$, $T_{ramp} = 0.5$), clearly identifying it as the ramp that best resembles the 3.125 km horizontal grid resolution reference runs. For the two higher horizontal grid resolutions, the minimum mean and standard deviation scores arise for different temperature ramps, preventing the determination of a single best ramp.

We complement the above analysis by upscaling the 3.125 km reference runs. For example, a 25x25 km grid cell contains

635 a patch of 64 3.125x3.125 km grid cells. The scatter plot (e.g., Fig. 10) of the warm-based fraction (basal temperature with respect to the pressure melting point at 0 °C) and the mean basal temperature with respect to the pressure melting point of the patch can be used to estimate the parameters $T_{ramp}$ and $T_{exp}$ of the basal temperature ramp (Eq. (8)). However, this does not account for the connectivity between the faces of, e.g., a 25 km grid cell. Without a continuous warm-based channel from one grid cell interface to another, there should be effectively no basal sliding across the grid cell, even when the average basal

temperature is close to the pressure melting point. Consequently, this estimate for the basal temperature ramp should be a lower bound to the points in the scatter plot. Furthermore, the upscaling results depend on the bed properties (soft sediment vs. hard bedrock) and the specific scenario (surge vs. quiescent phase). As such, the upscaling statistics only consider grid cells within the pseudo-Hudson Strait area during surges. Due to the limited storage capacity for the 10 yr output fields, only the first 10 kyr after the first surge are used for the upscaling experiments.

The upscaling results agree well with the score analysis at 25 km horizontal grid resolution. Both indicate that at this resolution, the ramp $T_{exp} = 5$, $T_{ramp} = 0.5$ (first row in Table S11, Fig. 10) gives results that best match those of the 3.125 km reference run. The two approaches yield a similar range of temperature ramps at 12.5 and 6.25 km horizontal grid resolution, but the upscaling experiments generally favor wider temperature ramps (Table S11 and Fig. S26 and S27). This is likely a consequence of the above-mentioned role of sub-grid warm-based connectivity not accounted for in the upscaling analysis.

When using the resolution-dependent ramp of Eq. (9), the upscaling experiments, therefore, provide a lower bound of $T_{exp} = 5$. Upscaling experiments with local basal hydrology lead to similar results.

### 3.3.4 Smooth sediment transition zone and non-flat topography

The effects of a smooth sediment transition zone (instead of an abrupt transition from hard bedrock (0 % sediment cover) to 100 % (soft) sediment cover) and a non-flat topography on surge characteristics are examined here. In the GSM, the

655 smooth transition zone alone does not significantly affect the surge characteristics. Additionally imposing a non-flat topography (Fig. S5) leads to more, longer, and stronger (larger mean pseudo-Hudson Strait ice volume change) surges (Fig. 6). The PISM experiments show fewer, longer and stronger surges for a non-flat topography (Fig. 7 and S9), but no significant effect for an abrupt sediment transition (instead of a more gradual transition, e.g., Fig. S8).

The abrupt transition from hard bedrock to soft sediment (pseudo-Hudson Bay and Hudson Strait) in the GSM reference

setup and the corresponding difference in basal sliding coefficient provide an additional heating source due to shearing between slow and fast-moving ice. This additional heat appears to foster the propagation of small surges along the transition zone (e.g.,

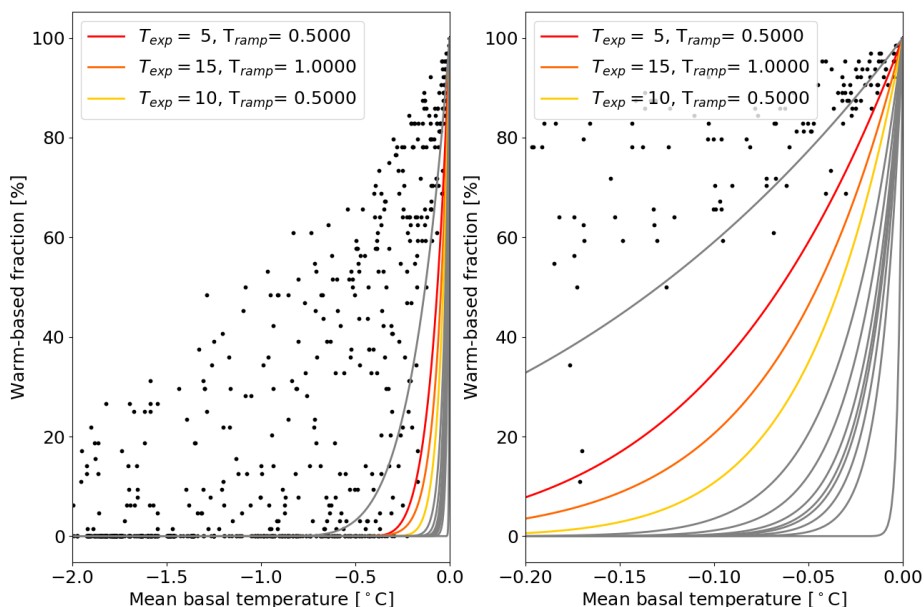

**Figure 10.** Warm-based fraction (basal temperature with respect to the pressure melting point at 0 °C) vs. mean basal temperature with respect to the pressure melting point when upscaling a 3.125 km run to 25 km horizontal grid resolution including all 5 parameter vectors using the GSM. For example, an upscaled 25 km patch (containing 64 3.125 km grid cells) with 32 3.125 km grid cells at the pressure melting point and 32 3.125 km grid cells at −1°C with respect to the pressure melting point has a warm-based fraction of 50 % and a mean basal temperature of −0.5°C. Only grid cells within the pseudo-Hudson Strait and time steps within the surges of the 10 kyr after the first surge are considered. The restriction to the 10 kyr after the first surge for these experiments is set by storage limitations due to the high temporal resolution of the model output fields (10 yr). The colored ramps correspond to the 25 km horizontal grid resolution basal temperature ramps in Table S11 and the gray lines show all other ramps that were tested at this resolution.

6 to 6.3 kyr in the upper row of video 03 of Hank (2023)). Incorporating a smooth transition zone (3.125 km or 25 km wide) affects the location of the small-scale surges (not considered in surge characteristics) but shows only minor differences for the major surges (< 7.5 % for all surge characteristics, Fig. 6). The mean bias for both widths is < 1 %, indicating only minor

differences in ice volume between an abrupt and smooth transition. However, the timing of surges varies for different transition zones (RMSE ≤ 8 %, Fig. 11). A wider transition zone (more sediment surrounding the pseudo-Hudson Strait and Hudson Bay) generally favors an earlier sliding onset (e.g., Fig. 11), but the details depend on the parameter vector in question.

In the GSM, imposing a non-flat basal topography has a more significant effect than the sediment transition zone. In general, the number of surges, mean duration, and mean pseudo-Hudson Strait ice volume change all increase compared to a flat

topography (Fig. 6). Note that Fig. 6 also shows an increase in the mean period, but this is somewhat misleading due to the now

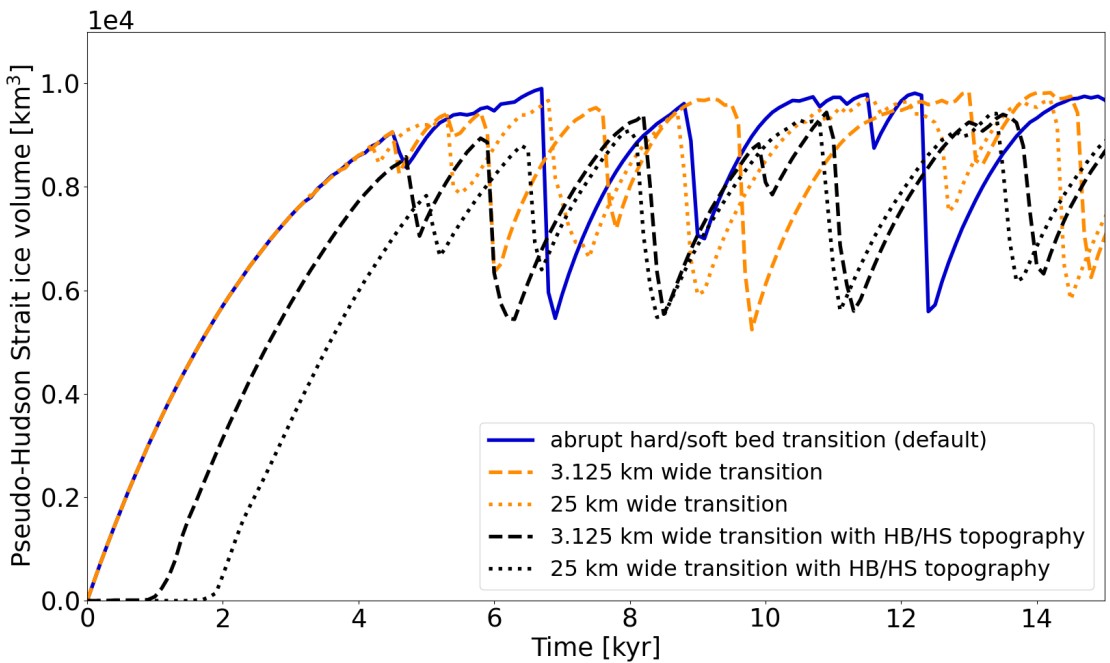

**Figure 11.** Pseudo-Hudson Strait ice volume for GSM parameter vector 1 and three different bed configurations. The horizontal grid resolution is 3.125 km. Note that the width of the topographical transition zone matches the width of the soft to hard bed transition zone. In experiments with a pseudo-Hudson Bay/Hudson Strait (HB/HS) topography, the pseudo-Hudson Strait topography is below sea level, increasing the time required for glaciation. A wider transition zone (larger area below sea level) leads to a later glaciation.

early surges for parameter vector 0 and the subsequent large increase in the mean period ($\sim 100$ %, no surges in the middle part of the run due to cold surface temperatures (Fig. S28)). All other parameter vectors show a decrease in the mean period for both widths of the transition zone. The mean bias indicates a decrease in ice volume of $\sim 6.5$ % for runs with a non-flat topography caused by the larger surges.

A wider transition zone (smaller slope) leads to fewer (difference of 16 %) but stronger surges (difference in mean pseudo-Hudson Strait ice volume change and mean duration of 9 % and 14 %, respectively, Fig. 6). A detailed comparison of an individual run is presented in Sec. S7.4.

The width of the transition zone ($-200$ m to sea level) affects the position and width of the major surges (tendency towards wider surges for a wider transition zone (video 04 of Hank (2023))). The pseudo-Hudson Strait topography also suppresses the
small surges otherwise observed in the vicinity of the pseudo-Hudson Strait.

Similar to the GSM results, the PISM percentage differences between a smooth (reference setup) and abrupt sediment transition show no significant effect, except a 22 % increase in surge duration (Fig. 7). While imposing a non-flat topography fosters surges in both models, the increase in mean ice volume change is much larger in PISM (390 %) than in the GSM

(maximum $\sim 17$ %), leading to a longer regrowth-phase (79 % increase in mean period) and overall less ice volume (mean bias $-30$ %, Table S13). The longer recovery times in the PISM outweigh the effect of earlier sliding onsets leading to more surges described above for the GSM. Therefore, the number of surges decreases in the PISM (while increasing in the GSM) when using a non-flat topography (Fig. 7).

Since the topography will vary from ice stream to ice stream, we stick to a flat topography for the remaining experiments.

### 3.3.5 Basal hydrology

The effects of adding a simple local basal hydrology model to the GSM are examined here. The local basal hydrology sets the basal water thickness by calculating the difference between the basal melt rate and a constant basal drainage rate (rBedDrain-Rate in Table 1). This sub-glacial hydrology provides a simple and computationally efficient way to capture changes in basal sliding velocities due to effective pressure variations (Drew and Tarasov, 2022, under review). However, it does not account for basal ice accumulation, englacial or supraglacial water input, or horizontal water transport.

The basal water thickness ($h_{wb}$) and an estimated effective bed roughness scale ($h_{wb,Crit}$ in Table 1) determine the effective pressure coefficient

$$N_{C,eff} = 1 - \min\left(\frac{h_{wb}}{h_{wb,Crit}}, 1.0\right)^{3.5} \tag{20}$$

The basal water thickness is limited to $h_{wb,Crit} = 10$ m and is set to $h_{wb} = 0$ m where the ice thickness is less than 10 m and where the temperature with respect to the pressure melting point is below $-0.1°$C. Experiments with $h_{wb,Crit} = 5$ m yield the same results, and removing all the water for $H < 1$ m, $H < 50$ m, and $T_{bp} < -0.5°$C does not significantly (according to Sec. 3.2) affect the model results. The effective pressure at the grid cell interface is then

$$N_{eff} = g\rho_{ice} \cdot 0.5\left(H_L N_{C,eff,L} + H_R N_{C,eff,R}\right), \tag{21}$$

where $g = 9.81$ m s$^{-2}$ is the acceleration due to gravity, $\rho_{ice} = 910$ kg m$^{-3}$ the ice density, $H$ the ice thickness and the subscripts $L$ and $R$ denote the adjacent grid cells to the left and right of the interface, respectively (similarly for upper and lower grid cells adjacent to a horizontally aligned interface). We enforce that $N_{eff}$ never falls below 10 kPa (denominator in Eq. (22), similar results for $N_{eff,min} = 5$ kPa). Finally, the effective pressure of each grid cell alters the basal sliding coefficient in the sliding law (Eq. (6a)) according to

$$C_b = C_b \cdot \min\left(10, \max\left(0.5, \frac{N_{eff,Fact}}{N_{eff} + 10^4 \text{ Pa}}\right)\right), \tag{22}$$

where $N_{eff,Fact}$ is the effective pressure factor (Table 1). The change of the basal sliding coefficient $C_b$ is, therefore, limited to $C_b \cdot 0.2$ to $C_b \cdot 10$. Allowing a larger change of $C_b \cdot 0.1$ to $C_b \cdot 20$ does not significantly (according to Sec. 3.2) change the model results.

When running the GSM with the local sub-glacial hydrology model, intermediate values are used for all 3 parameters (the effective bed roughness scale $h_{wb,Crit} = 0.1$ m (Eq. (20)), the constant bed drainage rate rBedDrainRate $\simeq 0.003$ m yr$^{-1}$, and the effective pressure factor $N_{eff,Fact} \simeq 63246$ Pa (Eq. (22))) for all 5 parameter vectors. However, different values were tested

for all 3 parameters (not shown). In general, a larger $N_{\mathrm{eff,Fact}}$ increases the basal sliding coefficient (Eq. (22)) and, therefore, leads to fewer but stronger surges. The results for $\mathrm{h_{wb,Crit}}$ and rBedDrainRate are not as straightforward to interpret. The model response varies for the 2 tested parameter vectors, and the changes are generally smaller than the MNEEs of Table 5.

    Adding the local basal hydrology model to the GSM increases the mean ice volume change and duration by 20 % and 12 %, respectively (Fig. 6, exceeding the MNEEs (Table 5)). The stronger surges are due to the reduction of effective pressure and,

thus, increased sliding (Eq. (22) and (6a)). The mean period increases (17 %) while the number of surges decreases ($-4$ %), but the standard deviations are large.

    Since the local hydrology model effectively increases the basal sliding coefficient, we test if this impact can be replicated simply by increasing the sliding coefficients (Table 1) in a GSM configuration without basal hydrology. Doubling the soft bed sliding coefficient leads to a similar model response but with a smaller increase in the mean period (12 % vs. 17 %)

and mean pseudo-Hudson Strait ice volume change (11 % vs. 20 %) than the local hydrology model. Increasing the hard bed sliding coefficient has no significant effect on the surge characteristics (pseudo-Hudson Bay and Hudson Strait are soft-bedded, Fig. 6). Simultaneously increasing the soft and hard bed sliding coefficient yields similar results to increasing the soft bed sliding coefficient alone (not shown).

### 3.3.6   Sensitivity experiments without a significant effect

The effect of an experiment is considered insignificant when the change in surge characteristics is smaller than the MNEEs (Sec. 3.2). This is the case for different weights of the adjacent minimum basal temperature when calculating the basal interface temperature ($Q_6$), for different implementations of the basal hydrology ($Q_9$), and when using basal hydrology instead of the basal temperature ramp as the primary smoothing mechanism ($Q_{10}$). The details of these experiments are presented in Sec. S8.1, S8.2, and S8.3, respectively. We want to emphasize that experiments without a significant effect can still have

physical relevance, but it is currently hidden within the numerical sensitivities.

### 3.4   Convergence study

In this section, we examine the horizontal grid resolution and time step dependence of the GSM and PISM model results. Model results are considered as converging when the differences in surge characteristics decrease with increasing horizontal grid resolutions and decreasing time steps. In general, both models show convergence, but the discrepancies between different

horizontal grid resolutions are significant.

### 3.4.1   GSM convergence study

Significant differences in surge characteristics occur when changing the horizontal grid resolution. These differences can be as large as a highly oscillatory behavior at 3.125 km and no oscillations at 25 km horizontal grid resolution (Fig. S23). Changing the basal temperature ramp can somewhat counteract this discrepancy by enabling basal sliding at lower basal temperatures

for coarser grid resolutions (Fig. 12 and video 05 of Hank (2023)). Further details on discrepancies between horizontal grid resolutions for individual parameter vectors are discussed in Sec. S9.1.

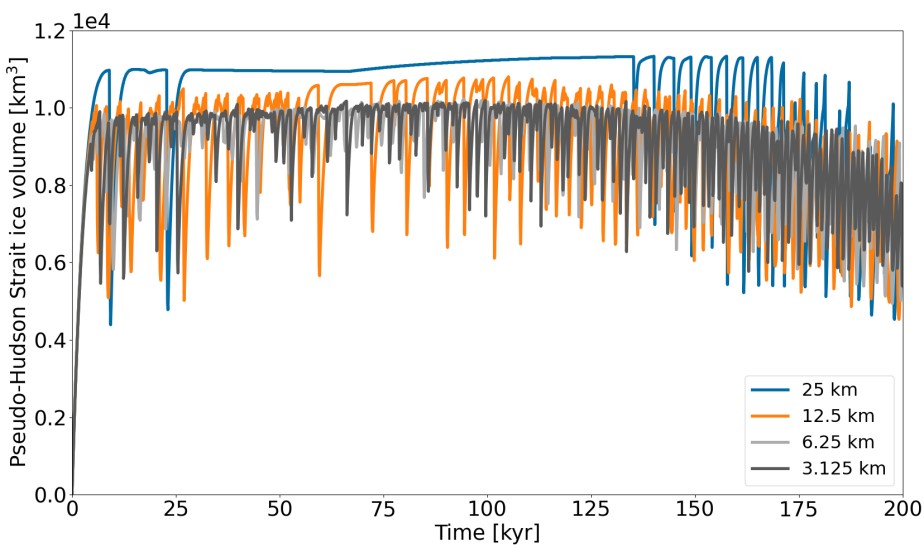

**Figure 12.** Pseudo-Hudson Strait ice volume for parameter vector 1 and different horizontal grid resolutions using the GSM. A resolution-dependent temperature (Eq. (9)) with $P_{T_{ramp}} = 1$ and $T_{exp} = 28$ is used for all horizontal grid resolutions (matching colors in Fig. 2).

We compare the differences in surge characteristics for different basal temperature ramps at each resolution. We examine: a constant ramp ($T_{ramp} = 0.0625$, $T_{exp} = 28$), a resolution-dependent temperature ramp ($T_{exp} = 28$, Fig. 2), and the ramp with the smallest differences in surge characteristics (bold mean score in Table S11). Note that the large differences in mean period
at 25 km resolution are caused by long time intervals without any oscillations in the coarse resolution runs (Table S18). 25 km, 12.5 km, and 6.25 km runs show progressively smaller differences for the constant and resolution-dependent ramp, indicating model convergence. Convergence of the GSM results with increasing grid resolutions is further supported by successively smaller pseudo-Hudson Strait ice volume RMSE and mean bias values (Table S19). RMSE and mean bias are smaller across all resolutions when using a resolution-dependent instead of a constant temperature ramp (except for the RMSE at 12.5 km
horizontal grid resolution).
All three basal temperature ramps lead to similar differences in surge characteristics at 6.25 km and 12.5 km horizontal grid resolution (Table S18). At 25 km resolution, the ramp with the minimum differences in surge characteristics significantly improves the agreement with the 3.125 km runs, with differences smaller than for any other ramp or resolution. This could either be a coincidence or indicate that despite thorough testing, the best ramp has not been found at 6.25 km and 12.5 km
horizontal grid resolution. Since other ramps at 25 km horizontal grid resolution show only slightly larger differences in surge characteristics (e.g., difference of 0.23 in the mean score, Table S11), it is unlikely that it is just a coincidence. However, the

sensitivity of the surge characteristics to grid refinement remains, no matter the choice of the temperature ramp, with differences significantly exceeding the MNEEs from Sec. 3.2.

Since including a sub-glacial hydrology model significantly affects the surge characteristics, we also examine the horizontal grid resolution scaling with a local basal hydrology model (Sec. S9.2). The results show overall smaller differences (relative to the 3.125 km reference simulations) in surge characteristics than without (Table S22 vs. S18). The analysis of the convergence study (with and without basal hydrology) and the upscaling experiments in Sec. 3.3.3, therefore, suggest a resolution-dependent temperature ramp with $T_{exp}$ between 5 and 10.

Experiments with different maximum time steps show only minor ($< 8$ %) differences compared to the reference runs for all surge characteristics (Table S18), RMSE and mean bias (Table S19). Overall, changes due to different maximum time steps are considerably smaller than for different horizontal grid resolutions and within the range of MNEEs (Table 5). Therefore, the implemented CFL condition is adequate for determining the ice dynamical time step, even though the condition is only sufficient for the solution of linear partial differential equations.

### 3.4.2 PISM convergence study

Similar to the results presented for the GSM, running PISM with different resolutions can lead to significant differences in surge behavior. However, the PISM surge characteristics do not show convergence for the three resolutions examined here (Table S25). Note that 4 out of 9 12.5 km runs did not finish within the time limit of the computational cluster and are considered as crashed runs (potentially skewing the statistics). Additionally, 1 12.5 km did not show any surges and was also excluded from the analysis. The differences in surge characteristics for different grid resolutions are, in general, larger than the MNEEs, but can be smaller (mean ice volume change of the 25 km runs). The ice volume RMSE and mean bias converge (but not the surge characteristics, see Table S26).

Similar to the results for different numbers of cores, small differences can slowly accumulate for different maximum time steps, leading to a different pattern at the end of the run (e.g., Fig. S31). As for the GSM, different horizontal grid resolutions have a larger effect on surge behavior than the maximum time step for all surge characteristics (Table S25).

### 4   Results Summary and Discussion

This section summarizes our modeling results in the context of the research questions outlined in Sec. 1.3 and previous modeling studies.

For the sensitivity experiments, changing the approach for determining the basal temperature at the grid cell interface shows the largest differences in surge characteristics compared to the reference setup. Including a bed thermal model has overall the second strongest effect. A non-flat basal topography and sub-glacial hydrology show moderate differences (Fig. 6 and 7). The effects of different basal temperature ramps depend on the width of the ramp but are generally moderate (Fig. 9). The weight of the adjacent minimum basal temperature, the details of the basal hydrology model, and the primary smoothing mechanism

at the warm/cold-based transition zone (sub-glacial hydrology vs. basal temperature ramp) do not have an effect above our reference MNEE threshold.

In general, changing the horizontal grid resolution shows moderate differences. However, depending on the resolution and the temperature ramp, the differences can be as large as for the most impactful sensitivity experiments (Table S19). Experiments with different maximum time steps show differences smaller than the reference MNEE threshold.

### Minimum numerical error estimates

$Q_1$ *What is the threshold of MNEEs in the two models?*

Since the model error to the exact analytical solution can not be determined in the context of this study, we use MNEEs to determine a minimum threshold for the significance of a change in the model configuration. When modeling ice stream surge cycling, numerical sensitivities are apparent in both the GSM and PISM. The differences in surge characteristics when applying stricter numerical convergence criteria in the GSM can be as large as 7 % (Table 5). Adjusting the matrix solver used in PISM (different number of cores) leads to differences in surge characteristics of up to 16 % (Table 6). Consequently, the model sensitivity to physical model aspects cannot be determined if the differences in surge characteristics are smaller than the MNEEs and are considered insignificant.

In contrast to the findings of Souček and Martinec (2011), adding low levels of surface temperature noise does not significantly affect the GSM and PISM results (Table S3 and S6). Potential reasons for the different model responses are the use of an Arakawa A grid (velocities and temperatures are calculated on the same node, Arakawa and Lamb, 1977) and the JOSH (JOint Shallow-ice/Higher-order model) ice sheet dynamics in Souček and Martinec (2011).

We expect other ice sheet models with a comparable experimental design and ice dynamics to show similar levels of MNEEs. To minimize the possibility of interpreting numerical errors as a physical response to a change in model setup, it is crucial to determine MNEEs (or a comparable metric).

### Sensitivity experiments with a significant effect

$Q_2$ *Is the inclusion of a bed thermal model a controlling factor for surge activity?*

Including a 1 km deep bed thermal model significantly (according to the MNEEs in Sec. 3.2) affects the surge characteristics in the GSM and PISM. The additional heat stored in the bed changes the thermal conditions at the ice-bed boundary, dampening the ice volume change during a surge (Fig. 6 and 7). Models with similar setups but without a bed thermal model likely overestimate the ice volume change during a surge (e.g., Calov et al., 2010; Brinkerhoff and Johnson, 2015). Therefore, the inclusion of a bed thermal model is a key aspect of modeling ice stream surge cycling.

$Q_3$ *Do different approaches for determining the grid cell interface basal temperature significantly affect surge behavior, and if yes, which one should be implemented?*

The choice of approach for determining the basal temperature at the grid cell interface significantly changes the surge characteristics. Without considering additional heat transfer to the grid cell interface (as an attempt to represent heat

contributions from sub-glacial hydrology and ice advection), 4 out of 5 runs do not show any surges, and the number of surges for the remaining run decreases by 84 % (TpmInt). The additional heat is, therefore, an essential component for modeling surges in the GSM. Using an upwind scheme for TpmInt has no significant effect (Fig. 6).

The additional heat transfer to the grid cell interface is comparable to spreading 50 % of the basal heating effect from sliding in a grid cell to the surrounding grid cells used in mPISM (latest version based on PISM v0.7.3) (e.g., Ziemen et al., 2014, 2019; Schannwell et al., 2023). This spreading of basal heating warms the grid cells adjacent to an ice stream and was necessary to model Heinrich Event-like surges (Florian Ziemen, personal communication). While no additional heat transfer was added to PISM v2.0.2 used within this study, the till friction angles had to be reduced to model surges.

$Q_5$ *How different are the model results for different basal temperature ramps and what ramp should be used?*

Similar to Souček and Martinec (2011), we find significant differences in the period and amplitude of surges at all tested resolutions when using different implementations for thermal activation of basal sliding (the basal temperature ramp). In the GSM, a wider temperature ramp enables sliding onset at lower temperatures, fostering surge propagation and leading to stronger surges. However, the choice of the most appropriate temperature ramp at the highest resolution tested (3.125 km, Fig. 9) is unclear and identifying a single best ramp (fit of coarse resolutions runs to 3.125 km runs) is challenging (Table S11). In general, a resolution-dependent ramp with $T_{exp}$ between 5 and 10 (Eq. (8) and (9)) yields the smallest differences between high and low resolution simulations. Even at the highest tested horizontal grid resolution (sharpest temperature ramp), running the GSM without a basal temperature ramp leads to significant (according to Sec. 3.2) differences in the mean duration, underlining the importance of the basal temperature ramp across all resolutions.

To account for observational and experimental evidence of sub-temperate sliding (Barnes et al., 1971; Shreve, 1984; Echelmeyer and Zhongxiang, 1987; Cuffey et al., 1999; McCarthy et al., 2017), avoid an abrupt onset of sliding at the warm/cold-based transition that causes refreezing on the warm-based side (Mantelli et al., 2019), and minimize resolution dependencies, a basal temperature ramp (or similar mechanism) should be implemented in all ice sheet models for contexts where surge onset/termination are important.

$Q_6$ *Does the abrupt transition between a soft and hard bed significantly affect surge characteristics?*

An abrupt transition between the soft (100 % sediment cover) and hard bed (0 % sediment cover) sliding law (as, e.g., used in the HEINO experiments Calov et al., 2010) can lead to additional localized shear heating due to differences in basal resistance. Incorporating a smooth transition zone with two different widths (3.125 km and 25 km) in the GSM does affect the location of proximal small-scale ice streams (video 03 of Hank (2023)). However, the abrupt transition is not the cause of the major surges (Fig. 6). PISM experiments with an abrupt transition (instead of the smooth transition used in the reference setup) show a slight increase in surge duration (22 %, Fig. 7) but otherwise no significant differences in the surge characteristics. Since the sediment cover can change within a few kilometers (e.g., Andrews and MacLean,

2003), we conclude that, despite the minor differences, an abrupt transition between soft and hard beds is a reasonable simplification, especially considering horizontal grid cell dimensions of 25 km or larger.

$Q_7$ *How does a non-flat topography affect the surge behavior?*

Adding a 200 m deep pseudo-Hudson Strait and Hudson Bay with a smooth transition zone and 500 m deep ocean to the GSM setup displaces the origin of surges slightly further inland. Due to both the resultant warmer basal temperature and depressed pressure melting point, the surges propagate faster, last longer, and evacuate more ice volume (Fig. 6). The topography slopes down towards the pseudo-Hudson Strait, increasing the ice inflow from the surroundings. The ice sheet recovers faster from the previous surge, decreasing the mean period.

Comparing the results for two different widths of the topographic transition zone indicates fewer but larger surges for a wider transition zone. Due to the gentle slope, the topography affects a larger area, increasing the width of the ice stream. More ice is available for evacuation, prolonging the surge and decreasing the pseudo-Hudson Strait ice volume at the end of the surge. The stronger surges for a wider transition zone increase the recovery time, leading to a smaller increase in the number of surges than for the narrow transition zone (difference of 16 %, Fig. 6).

Imposing a non-flat topography in the PISM also leads to longer and stronger surges (Fig. 7). However, the increase in mean ice volume change is much higher than in the GSM (390 % vs. ∼ 17 %), prolonging the regrowth phase by ∼ 80 % and reducing the number of surges.

In agreement with previous modeling studies (e.g., Winsborrow et al., 2010, and references within), the topography is a key aspect of ice stream modeling. When interested in a comparison with observational data or proxy reconstructions, a more realistic topography (in contrast to the idealized flat topography) should be used.

$Q_8$ *What is the effect of a simplified basal hydrology on surge characteristics?*

The local basal hydrology model (including the addition of effective pressure dependence into the sliding law) in the GSM increases the mean ice volume change, mean period, and mean duration while the number of surges slightly decreases (Fig. 6). Somewhat stronger surges are expected due to the reduction in effective pressure introduced by the sub-glacial water. Model runs without sub-glacial hydrology will therefore tend to underestimate the amplitude of surges (mean ice volume change and duration).

Increasing the soft bed sliding coefficient in model runs without basal hydrology has a smaller increase in the mean duration, mean period, and pseudo-Hudson Strait ice volume change than including the local basal hydrology model, but a stronger effect on the number of surges (Fig. 6). Therefore, simply changing the basal sliding coefficient cannot replace the basal hydrology model. The importance of sub-glacial hydrology has also been shown in several other studies examining the effects of ice sheet surges and ice streaming within a continuum model approach (e.g., Fowler and Johnson, 1995; Fowler and Schiavi, 1998; Benn et al., 2019).

**Sensitivity experiments without a significant effect**

$Q_4$  *How much of the ice flow should be blocked by upstream or downstream cold-based ice, or equivalently, what weight should be given to the adjacent minimum basal temperature?*

Changing the weight of the adjacent minimum basal temperature for the basal sliding temperature ramp in the GSM yields a maximum difference of $15\%$ (Table S15). These somewhat small effects on surge characteristics are likely due to the fact that most surges propagate upstream (from the ocean to the pseudo-Hudson Bay) and the adjacent minimum basal temperatures (almost exclusively located upstream) have little potential to affect (e.g., partly block) the ice flow.

$Q_9$  *How significant are the details of the basal hydrology model on surge characteristics in PISM?*

Incorporating a mass-conserving horizontal transport hydrology model does not significantly change the surge characteristics in PISM (Fig. 7), indicating that the computationally much cheaper local hydrology model is a reasonable simplification for this context. More nuanced results, depending on the surge characteristics examined, are observed for the GSM (Drew and Tarasov, 2022, under review).

$Q_{10}$  *What are the differences (if any) in surge characteristics between local basal hydrology and a basal temperature ramp as the primary smoothing mechanism at the warm/cold-based transition zone?*

Surge characteristics in runs with an active local basal hydrology and a sharp temperature ramp ($T_{ramp} = 0.001$, $T_{exp} = 28$, minimizing the smoothing effect of the basal temperature ramp, Table S17) show only minor differences compared to the GSM setup with a local hydrology and the reference temperature ramp ($T_{ramp} = 0.0625$, $T_{exp} = 28$, Fig. 6). Once included, the local basal hydrology is the primary smoothing mechanism. However, since the two smoothing mechanisms operate in different temperature regimes, a basal temperature ramp (representing sub-temperate sliding) cannot be replaced by a basal hydrology scheme (as in, e.g., Robel et al., 2013; Kyrke-Smith et al., 2014; Brinkerhoff and Johnson, 2015). The numerical sensitivities prevent further analysis.

**Convergence study**

$Q_{11}$  *Do model results converge (decreasing differences when increasing horizontal grid resolution and decreasing maximum time step)?*

Systematic grid refinement shows a converging overall ice volume (mean bias) in both models (Table S19+S23 and S26). However, surge characteristics converge for constant and resolution-dependent basal sliding activation ramps in the GSM (Table S18), but not in PISM (Table S25). This clearly illustrates that mean ice volume and, consequently, mean ice thickness, as presented, e.g., in Van Pelt and Oerlemans (2012), are insufficient metrics to determine whether cyclic model results exhibit a resolution dependency. The highest horizontal grid resolution used for PISM is 4 times coarser than the highest resolution in the GSM (12.5 km vs. 3.125 km), which might explain why PISM results do not converge. In the GSM, the agreement between coarse and high-resolution runs can be significantly improved when applying a resolution-dependent temperature ramp (Table S18 and Sec. S7.3).

Surge characteristics in both the GSM and PISM show a strong resolution dependence for all sensitivity tests (Table S18+S22 and S25). While other studies examining thermally induced ice streaming do not find a strong resolution dependence (Hindmarsh, 2009; Brinkerhoff and Johnson, 2015), these studies are not directly comparable. The different results are likely due to differences in the experimental design. For example, neither Hindmarsh (2009) nor Brinkerhoff and Johnson (2015) consider a bed thermal model. While Hindmarsh (2009) considers sub-temperate sliding, his model allows sliding far below the pressure melting point (order of $\delta = 1$ compared to $\delta = 0.01$ within this study, Eq. (10)) and focuses on steady ice streams, not ice stream surge cycling. Over 200 kyr, even minor differences at the beginning of a run can slowly accumulate and yield overall different surge characteristics (e.g., Fig. S30). Furthermore, Brinkerhoff and Johnson (2015) examine ice stream statistics over the whole domain and not a specific soft-bedded region. Additionally, both of these studies analyze just one parameter vector, and there are some parameter vectors for which, e.g., the GSM exhibits only a minor resolution dependence.

Even though the studies are not directly comparable, the results of Brinkerhoff and Johnson (2015) offer some insight relevant to this study. For example, they suggest membrane stresses are necessary for convergence under horizontal grid refinement. The hybrid SIA/SSA ice dynamics used in the GSM and PISM might be insufficiently 'higher-order' and lead to a stronger resolution dependence than the schemes used in Hindmarsh (2009); Brinkerhoff and Johnson (2015). However, GSM experiments with the SSA active everywhere show a resolution dependence comparable to the velocity-dependent SSA activation criteria (Table S24 and S18, respectively), indicating that the hybrid SIA/SSA ice dynamics are not the sole reason for the strong resolution dependence.

Although claiming that their model does not show a strong resolution dependency, Roberts et al. (2016) show differences in the surge timing, ice volume change, and period in their supplement. Greve et al. (2006) also conclude that different horizontal grid resolutions have no significant effect on the surge characteristics, but the two time series are difficult to compare (not shown in the same plot).

Decreasing the maximum time step leads to only minor ($< 6$ % except the $-15$ % difference in the number of surges for the 0.5 year maximum time step setup in PISM, Table S18 and S25) changes in surge characteristics for both PISM and the GSM. This is in agreement with the findings of earlier studies (Greve and MacAyeal, 1996; Greve et al., 2006; Takahama, 2006). However, individual parameter vectors might still show a different surge pattern (e.g., Fig. S31).

## 5 Conclusions

Within the limitations of hybrid SIA/SSA ice dynamics, we investigate the effect of ice sheet model numerics on surge characteristics often neglected in ice sheet modeling studies. Minimum numerical error estimates (MNEEs, differences in surge characteristics of up to 16 % when changing the settings of the numerical solver) are used to discern the significance of the process in question. For some experiments (e.g., the weight of the adjacent minimum basal temperature), the MNEEs are on the same order of magnitude or larger than the modeled differences in surge characteristics, hindering the analysis of the underlying physical process. Experiments showing only minor changes in surge characteristics (generally smaller than the MNEEs)

include: a mass-conserving horizontal transport hydrology model (instead of a local hydrology model), a smoothed transition between regions of soft sediment and hard bedrock (instead of an abrupt transition), and smaller (than 1 yr) maximum time steps in the CFL condition.

On the other hand, surge characteristics are sensitive to the basal sliding activation function and show a strong resolution dependency. Since both the GSM and PISM show a resolution dependency, it is likely that it also exists in other ice sheet models with similar approximations. Incorporating a resolution-dependent basal temperature ramp for basal sliding thermal activation reduces the resolution dependency in the GSM. Based on our results, we suggest that those interested in modeling ice stream cycling at horizontal grid resolutions $> 3$ km should use a resolution-dependent ramp with $T_{exp} = 10$ as a reference test configuration. However, we strongly recommend resolution testing to determine the configuration with the smallest resolution dependency. Additionally, our results indicate that modeling of ice stream surge instabilities that aims to reflect the physical behavior of actual ice streams should include a non-flat topography, a bed thermal model, and a basal hydrology model.

Basal temperature spokes, such as the ones modeled in the EISMINT-F and H experiments (Payne et al., 2000), are not apparent in the PISM experiments. The GSM runs show some warm-based areas at the margins interspersed by colder regions, but this is likely due to a steep surface slope leading to a large driving stress, high velocity, and then consequently, a basal temperature increase. Therefore, neither the PISM nor GSM instabilities discussed here are comparable to the EISMINT temperature spokes. The absence of basal temperature spokes is likely due to the inclusion of membrane stresses in the ice dynamics of both models (Bueler et al., 2007; Bueler and Brown, 2009).

The key takeaway of this study is the numerical sensitivity that must be considered when numerically modeling ice stream surge oscillations. Our analyses offer guidance in minimizing these sensitivities for research contexts that limit horizontal grid cell resolution to larger than about 3 km. Significant (albeit smaller) MNEEs to the choice of thermal activation ramp remain at our highest tested horizontal grid resolution (3.125 km). Analytical examination (where possible) and/or higher-resolution numerical modeling with higher-order glaciological models is needed to further verify that modeling approaches represent the actual physical system for this context.

*Code availability.* TEXT

*Data availability.* TEXT

*Code and data availability.* The GSM source code (v01.31.2023) and run instructions are available at https://doi.org/10.5281/zenodo.7668472 (Tarasov et al., 2023). Instructions on how to install and run PISM and the PISM source code (v2.0.2) can be acquired from the repository at https://zenodo.org/record/6001196. Further information on how to recreate this work's results, input files, parameter vectors, and the analysis scripts used to determine the surge characteristcs can be found at https://doi.org/10.5281/zenodo.7905404 (Hank, 2023).

*Sample availability.* TEXT

*Author contributions.* TEXT

K.H. and L.T. conceptualized the ideas behind this study. All authors were involved in designing the experimental setup of
the GSM. K.H. designed the experimental setup for PISM and performed the modeling analysis for both models under the
supervision of L.T. All authors contributed to the results, interpretation, and writing of the manuscript.

*Competing interests.* The authors have no competing interests.

*Disclaimer.* TEXT

*Acknowledgements.* The authors thank *Andy Aschwanden*, *Ed Bueler* and *Constantine Khrulev* for support with the Parallel Ice Sheet Model
(PISM). We thank *Ed Bueler* and *Daniel F Martin* for fruitful discussions about the bed thermal model and the numerical tolerances,
respectively. We also thank *Florian Ziemen* and *Clemens Schannwell* for insightful discussions on modeling Heinrich Event-like surges. This
research has been supported by an NSERC Discovery Grant (number RGPIN-2018-06658), the Canadian Foundation for Innovation, and
the German Federal Ministry of Education and Research (BMBF) as a Research for Sustainability initiative (FONA) through the PalMod
project. EM was supported by the European Union (ERC-2022-STG, grant number 101076793). Views and opinions expressed are however
those of the author(s) only and do not necessarily reflect those of the European Union or the European Research Council Executive Agency.
Neither the European Union nor the granting authority can be held responsible for them.

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
