# Peer review of "Numerical issues in modeling ice sheet instabilities such as binge-purge type cyclic ice stream surging"

_EGUsphere, 2023_

## Referee Comment (RC1)

**Referee comment on 'Numerical issues in modeling ice sheet instabilities such as binge-purge type cyclic ice stream surging'**

**March 2023**

**Summary**

In this manuscript, the authors explore the effect that a wide variety of factors, both numerical and physical, have on the time-dynamics of a synthetic thermomechanically coupled ice sheet constructed as a proxy for the Laurentide ice sheet during the last glacial period. In particular, they seek to understand which modelling choices lead to effects that rise above the well-documented sensitivity of such models - when a dependence of sliding on temperature is included - to symmetry breaking and chaotic dynamics from discretization.

I think that this manuscript has the potential to be a useful contribution to the (slowly) growing body of literature on the relationship between thermodynamics and sliding. However, I think that this manuscript needs quite a bit of work to be publishable. My criticism is not generally aimed at the science, but rather at the presentation which reads more like a set of internal notes than a manuscript. In particular the organization does not proceed in a logical manner, and I struggle to find a narrative that relates adjacent sections. In addition, the manuscript includes difficult-to-remember jargon and sometimes dubious interpretations of referenced material. There are far too many tables; it is very difficult to make sense of tabular numbers appearing in columns that reference one of many difficult to differentiate experiments. Perhaps most critically, some of the most important figures - many of which are referenced frequently and are essential to the paper's conclusions - are relegated to supplemental material or appear in one of the authors' previous papers. I do not suggest any new experiments or analysis here. However, I do suggest that the authors revise this manuscript with a keen eye for the fact that it will, ultimately and hopefully, be read by others.

**Line by line comments**

**Title** The title should be shortened to something like 'Numerical issues in modeling binge-purge behavior in ice streams.' No other instabilities are addressed and 'binge-purge', 'cyclic', and 'surging' are all redundant.

**L1** Delete 'as in any environmental system'.

**L20** I think it is strange to lead with validation as a means of motivating the current manuscript, when no validation occurs here. Validation is not the same as sensitivity testing.

**L26** What is 'numerical noise'? Something random? Pseudo-randomness in a chaotic system? Numerical *error*? This is a critical consideration but it's not really clear what it means in this paper here and elsewhere.

**L38** Define 'meaningful'

**L44** The quote from Soucek and Martinec is relevant, but it misses the fact that there are a great many approximations that appear in, for example, the solution of the Stokes' equations. Why the emphasis on numerical rather than model error?

**L60–72** While I recognize that this paper focuses on the ISMIP-HEINO setup, it would be worthwhile to try to contextualize this work with respect to the EISMINT-F experiments as well. There is a great deal of insight there regarding thermal sliding instabilities and the circumstances under which they appear.

**Sec 1.1** I find the organization of the paper according to research question to be quite challenging to follow, perhaps mostly because there are so many (11) research questions. I think it would be better to group these into open questions rather than yes/no, and this might make for more comprehensible themes. For example, Q1,11 could be grouped as 'what aspects of simulated surges are due to numerical considerations?', while Q2–6,9 could be grouped as 'what modeling and solution choices influence surging?' and Q7,8,10 as 'what parameterizations of basal physics leads to the most robust conclusions?' or something similar.

**L166 and elsewhere** I don't find it helpful to reference a manuscript that is 'in preparation' because such manuscripts are not readable and sometimes fail to ever get published. Is there some source code that could be referenced instead? An instruction manual? An older manuscript from which the ideas in the in prep manuscript are adapted?

**L198** I am deeply skeptical of a model that 'activates' stress terms based on a heuristic that in turn depends on whether the bed is soft or hard (whatever that means). Does it not seem that such an obviously non-physical choice could lead to just as much variability in surging behavior as any of these other mechanisms? Validation is mentioned in the introduction, but what about verification? How does the reader know (especially given that there is no current reference to the model description) that this model converges to the true solution of *some* physically and mathematically justified system of equations under discretization refinement?

**L215** I don't know what 'legacy' means here.

**Eq. 5** Is this supposed to be $F_{T_{ramp}}$? Otherwise $F_{warm}$ is defined twice. Also, I think it's really awkward (ignoring subscrips) to have $F$ depend on $T$, which depends on a different $F$. Maybe consider different notation?

**Sec. 2.1.2** I generally find the phrase 'vector' to be unhelpful here. I think it would be better to describe how the ensemble is created (i.e. by selecting different values for each of eight parameters) and then referring to different members of the ensemble as, well, 'ensemble members'.

**Sec. 2.1.2** It takes quite a bit of flipping around to understand why we're talking about ensembles at all. I think this section could use a clear explanation of the fact that you're running each subsequent experiment with multiple different parameter settings.

**Sec. 2.1.3** I think that 'reference simulation' might be more clear than 'base setup'.

**Sec. 2.1.3** I think that the very frequent referencing to future sections is not very helpful.

**Sec. 2.1.3 and elsewhere** There are far too many references to supplementary information in this manuscript. SI is intended for things that either cannot appear in the manuscript itself due to medium (e.g. code or videos) or that offer additional insight or detail into some aspect of the work but that is not essential to the results. In this case, the mass balance forcing (the single most important thing in determining long term ice sheet extent) is relegated to the supplement, but really should be in the main text.

**2.2.1** It's strange to imply that PISM is *not* also optimized for computational speed. It's the *parallel* ice sheet model, after all.

**L303–310** I think it would be helpful not to mix units of measurement (m/a and m/d). What is a 'stable solution of the numerical matrix solver'? Is 'observed range' the heuristic from Cuffey and Paterson?

**L318** 'event' and 'HE' seem to be used interchangeably in the manuscript. I think it would be better to just use 'HE'.

**L330** This is another circumstance where including the supplemental figure in the main manuscript would be very helpful.

**L341–342** 'ice-free when no surge occurs.' I'm not sure it's possible to ascribe a date to when something doesn't happen.

**3.1.1** What about PISM? Can surges be understood similarly to those in GSM?

**3.1.2** I'm not sure it's reasonable to try to state a specific justification for why the time scales of this highly idealized and not-observationally-constrained experiment are dissimilar to geological records: many different factors may be in play here (some improving the fit, some to its detriment) and (as an example) saying that the period mismatch is just the result of domain size seems likes its missing a broader set of possibilities.

**3.1.3** I again struggle with the notion of 'numerical noise'; the paper would be well served by having a much more in-depth description of what is meant by this and where it comes from. With respect to the latter, in most modelling exercises, the numerical error is something that can be well quantified through comparison to exact solutions or by a theoretical analysis of the interpolation properties of numerical method. Yet here, what we're effectively measuring is the system's sensitive dependence to small perturbations. In that sense, it doesn't necessarily follow that stricter convergence criteria (in the case of GSM) would necessarily lead to any less 'noise'. Could an equivalent result be achieved by just adding white noise to the initial conditions?

With PISM, it's also somewhat concerning that the number of cores used is a source of noise. Why is parallelism leading to different numerical solutions? I suppose it probably doesn't matter, but why not use the same mechanism of tolerance change to get modified runs as in GSM?

**3.1.5** This section on implicit coupling is so vague as to be useless. What even is 'implicit coupling' in this context? Is this the same as implicit time stepping, i.e. Backward-Euler?

**3.2.3** I can't figure out what TpmTrans, TpmInt, or any other Tpm thing are. If they are described earlier, such a description needs to be here instead or also. If they are not, they need to be defined (and not in the supplement).

**3.2.4** It's not my preference, but if you prefer to have the actual equations describing GSM in the supplement, I suppose that's fine. However we need at least a little bit of a qualitative description of what these different 'weights' imply. What is the context for understanding why these different choices should yield different surging behaviors?

**Fig. 7** I honestly can't figure out what this figure is trying to convey. Part of this is that I also can't figure out what the part of the text that references it is trying to convey either (L534–540). Please try to make this a little bit more clear.

**3.3** Brinkerhoff and Johnson, 2015 suggest that the inclusion of membrane stresses leads to convergence under spatial grid refinement, whereas without them, the SIA does not lead to convergence. Can you place those results in the context of this section? Are the relatively weak convergence results here a result of GSM and PISM velocity solvers being insufficiently 'higher-order'?

**Sec. 4** If you maintain these research questions as an organizing principle, I would like to see them revisited as they are resolved by the experiments rather than all at once at the end.

**L738–746** I think that this section is kind of weird: none of the results presented in this work actually refute the resolution dependency conclusions of Hindmarsh (2009) or Brinkerhoff and Johnson (2015), yet the paragraph is written as if they did. As mentioned before, it seems just as reasonable to assert that those works saw more robust numerical convergence due to the use of more consistent membrane stress resolution schemes rather than because they fortuitously (or nefariously) made parameter choices that suppressed resolution effects.

---

## Author Comment (AC2)

**Author's response to Anonymous Referee 1 Comment 1**

April 20, 2023

[Title - The title should be shortened to something like 'Numerical issues in modeling binge-purge behavior in ice streams.' No other instabilities are addressed and 'binge-purge', 'cyclic', and 'surging' are all redundant.] We have updated the title to 'Numerical issues in modeling ice stream surge cycling'.

[L20 - I think it is strange to lead with validation as a means of motivating the current manuscript, when no validation occurs here. Validation is not the same as sensitivity testing.] 'Model validation' was replaced by 'Determining model sensitivities'.

[L26 - What is 'numerical noise'? Something random? Pseudo-randomness in a chaotic system? Numerical error? This is a critical consideration but it's not really clear what it means in this paper here and elsewhere.] The following sentence has been added to the revised draft: 'By numerical noise, we refer to any non-physical differences in the model solution induced by numerical aspects such as rounding errors and convergence criteria of the numerical solver.'

[L38 - Define 'meaningful'] Removed.

[L44 - The quote from Soucek and Martinec is relevant, but it misses the fact that there are a great many approximations that appear in, for example, the solution of the Stokes' equations. Why the emphasis on numerical rather than model error?] The model error can only be determined when the 'true' solution of the equations is known. No matter what model is used (BISICLES (sliding everywhere, minimal heat treatment at the bed (just a vertical flux, which in temperate regions produces water)) and ELMER ice (too expensive for ensembles needed) are not suitable), an analytical solution does not exist for this context. Additionally, uncertainties associated with the numerical aspects of a model have received limited attention (compared to the effect of different approaches to the Stokes equations). We have updated this paragraph to:

'Modeling of binge-purge type HEs and surges in general is challenging. While the effects of different approaches to the Stokes equations have been previously addressed [e.g., Brinkerhoff and Johnson, 2015], uncertainties associated with the numerical aspects of a model have received limited attention in studies examining ice sheet surging [e.g., Payne, 1995, Marshall and Clarke, 1997, Calov et al., 2002, Papa et al., 2006, Steen-Larsen and Dahl-Jensen, 2008, Calov et al., 2010, Robel et al., 2013, Feldmann and Levermann, 2017]. Sensitivity in model response to different numerical choices are evident [Calov et al., 2010, Roberts et al., 2016, Ziemen et al., 2019] and small perturbations of the system can significantly vary the form, amplitude, and period of binge-purge oscillations [Souček and Martinec, 2011, Mantelli et al., 2016]. The exact cause of the numerical sensitivities is often unclear. Souček and Martinec [2011] thus rightfully conclude that *"... the implementation of surge-type physics in large-scale ice-sheet models is rather problematic since the information about the physical instability may be lost in the numerics"*. Furthermore, the theory underpinning the understanding of the instability mechanisms is not fully developed (no analytical solution exists), especially in the context of a spatially extended 3D system, thus precluding systematic benchmarking of numerical models.'

[L60–72 - While I recognize that this paper focuses on the ISMIP-HEINO setup, it would be worthwhile to try to contextualize this work with respect to the EISMINT-F experiments as well. There is a great deal of insight there regarding thermal sliding instabilities and the circumstances under which they appear.] The coldest PISM runs do not show any signs of

temperature spokes (spokes in the EISMINT-F/H experiments are more pronounced the colder the surface temperature [Payne et al., 2000]). In the GSM, there are some warm-based areas at the margins, but this is likely due to a steep surface slope leading to a large driving stress, high velocity, and then consequently, a basal temperature increase. A short discussion of this will be added to the revised draft.

[Sec 1.1 - I find the organization of the paper according to research question to be quite challenging to follow, perhaps mostly because there are so many (11) research questions. I think it would be better to group these into open questions rather than yes/no, and this might make for more comprehensible themes. For example, Q1,11 could be grouped as 'what aspects of simulated surges are due to numerical considerations?', while Q2–6,9 could be grouped as 'what modeling and solution choices influence surging?' and Q7,8,10 as 'what parameterizations of basal physics leads to the most robust conclusions?' or something similar.] The idea behind specific research questions is to make it easy for the reader to jump right to the sections they are most interested in. Grouping these individual questions into main themes will likely increase readability, but the suggested themes do not work because several research questions fit into more than one category. Instead, the manuscript will be restructured as follows: 1) strictly numerical aspects (Q1 and Q11), 2) numerical/modeling choices with a significant effect on the results (Q3, Q4, Q5, Q8), 3) numerical/modeling choices without a significant effect (Q2, Q6, Q9, Q10). As suggested by the second reviewer, the third theme will only be a short summary, with the details moved to the supplement.

[L166 and elsewhere - I don't find it helpful to reference a manuscript that is 'in preparation' because such manuscripts are not readable and sometimes fail to ever get published. Is there some source code that could be referenced instead? An instruction manual? An older manuscript from which the ideas in the in prep manuscript are adapted?] The source code of the model version used in this manuscript can be found in the supplementary material [Tarasov et al., 2023] as stated in the Code and data availability section. Additionally, we have added this reference to the GSM description section. Older manuscripts on which the current GSM version is based on are also mentioned in this section [e.g., Pollard and DeConto, 2012, Tarasov et al., 2012, Bahadory and Tarasov, 2018].

[L198 - I am deeply skeptical of a model that 'activates' stress terms based on a heuristic that in turn depends on whether the bed is soft or hard (whatever that means). Does it not seem that such an obviously non-physical choice could lead to just as much variability in surging behavior as any of these other mechanisms? Validation is mentioned in the introduction, but what about verification? How does the reader know (especially given that there is no current reference to the model description) that this model converges to the true solution of some physically and mathematically justified system of equations under discretization refinement?] Different SSA activation criteria are available in the GSM. Sensitivity to this choice will be described in the revised supplement.

[L215 - I don't know what 'legacy' means here.] Changed to 'values used in previous GSM modeling studies [e.g., Bahadory and Tarasov, 2018]'.

[Eq. 5 - Is this supposed to be $F_{T_{\mathrm{ramp}}}$? Otherwise $F_{\mathrm{warm}}$ is defined twice. Also, I think it's really awkward (ignoring subscrips) to have F depend on T , which depends on a different F . Maybe consider different notation?] $F_{\mathrm{warm}}$ is correct. We compare our definition of $F_{\mathrm{warm}}$ to the one used by Fowler [1986], Mantelli et al. [2019]. 'F' represents different 'factors' in the equations. Since they are clearly distinguishable by their subscripts, we prefer to stick to the current notation.

[Sec. 2.1.2 - I generally find the phrase 'vector' to be unhelpful here. I think it would be better to describe how the ensemble is created (i.e. by selecting different values for each of eight parameters) and then referring to different members of the ensemble as, well, 'ensemble members'.] For precision/accuracy and lack of alternatives, we prefer to stick to the phrase 'parameter vector' (note that a parameter vector and ensemble member are different things).

[Sec. 2.1.2 - It takes quite a bit of flipping around to understand why we're talking about

ensembles at all. I think this section could use a clear explanation of the fact that you're running each subsequent experiment with multiple different parameter settings.] Note that the text already explains the benefits of an ensemble: *'To partly address potential non-linear dependencies of surge cycling on model parameters, we use a high variance subset of 5 base GSM parameter vectors (each comprising 8 model input parameters) for our numerical experiments'.*

However, we have added the following sentence as introduction to this section. *'Each experiment uses a small ensemble of simulations.'*

[Sec. 2.1.3 I think that 'reference simulation' might be more clear than 'base setup'.] Note that 'reference simulation' and 'base setup' are not the same thing. In this study, there are 5 reference simulations (one for each base parameter vector) but only one base setup. To avoid potential confusion, we will stick to 'base setup'.

[Sec. 2.1.3 - I think that the very frequent referencing to future sections is not very helpful.] The forward referencing was meant to guide the reader and allow them to skip sections they are not interested in. However, this is better suited at the end of the introduction and was removed here.

[Sec. 2.1.3 and elsewhere - There are far too many references to supplementary information in this manuscript. SI is intended for things that either cannot appear in the manuscript itself due to medium (e.g. code or videos) or that offer additional insight or detail into some aspect of the work but that is not essential to the results. In this case, the mass balance forcing (the single most important thing in determining long term ice sheet extent) is relegated to the supplement, but really should be in the main text.] We suspect most readers will not want to read about the surface mass-balance details, and that is our criteria for main text inclusion. However, we now have more completely spelled out the temperature and surface mass-balance forcing in S1. Note that the climate forcing is already described earlier in the text: *'The GSM is run with an idealized down-scaled North American geometry (Fig. 1, modified after the ISMIP-HEINO setup [Calov and Greve, 2006]) and simplified climate representation. The temperature forcing is defined by a domain wide surface temperature (rTnorth, Tab. 1) and a specified vertical temperature gradient (atmospheric lapse rate (lapsr in Tab. 1)). The surface temperature forcing is asymmetric in time (Fig. S1), enabling the analysis of the timing of cycling onset and termination under different physical and numerical conditions (a comparison of ice stream ice volume evolution under constant and assymetric temperature forcing is shown in Fig. S2 for one parameter vector)'.*

Furthermore, Fig. S1 only shows the asymmetric aspect of the temperature forcing (atmospheric lapse rate and parameter vector dependency are not considered). Due to the simplicity of the plot, we do not deem it important enough to be in the main manuscript. However, we slightly adjusted the above text: *'[...] The surface temperature forcing is asymmetric in time (maximum difference of $10°C$, Fig. S1) [...]'*

[2.2.1 - It's strange to imply that PISM is not also optimized for computational speed. It's the parallel ice sheet model, after all.] That was not the intent of this statement. While both models are optimized, the optimizations are not for the same contexts. The idea behind using two different models is to minimize the possibility that drawn conclusions are solely a result of the used optimization schemes. Furthermore, the GSM is optimized for computational resource use to enable large ensembles over paleo timescales and therefore not parallelized (not the case for PISM).

To make this clearer, we have adjusted: *'The GSM is an ice sheet model developed specifically for glacial cycle ensemble modelling. The GSM is therefore numerically optimized for computational speed.'*

to: *'In contrast to PISM, the GSM is an ice sheet model developed specifically for glacial cycle ensemble modeling. The GSM therefore uses a distinct set of numerical optimizations for computational speed'.*

[L303–310 - I think it would be helpful not to mix units of measurement (m/a and m/d). What is a 'stable solution of the numerical matrix solver' ? Is 'observed range' the heuristic

from Cuffey and Paterson?] Agree and done.

Removed *'[...], indicating a stable solution of the numerical matrix solver even for runs with very high velocities.'*.

Yes, the corresponding part in K.M. Cuffey and W.S.B. Paterson. [2010] is: *'Speeds and displacements also vary widely. High velocities are about 100 m/day for short periods, and 5 km/yr maintained for one or two years. Low velocities are only several tens to a few hundred meters per year, values typical of many nonsurging glaciers'*.

[L318 - 'event' and 'HE' seem to be used interchangeably in the manuscript. I think it would be better to just use 'HE'.] They are not interchangeably. The term HE should be exclusively used when refering (to at least some extent) to the ocean sediment records/IRD layers. The abbreviation 'HE' and most instances of the term 'Heinrich Event' were removed.

[L330 - This is another circumstance where including the supplemental figure in the main manuscript would be very helpful.] Done.

[L341–342 'ice-free when no surge occurs.' I'm not sure it's possible to ascribe a date to when something doesn't happen.] *'a large fraction of the pseudo-Hudson Strait area is ice-free when no surge occurs'* changed to *'a large fraction of the pseudo-Hudson Strait area is only ice-covered when a surge occurs'*

[3.1.1 What about PISM? Can surges be understood similarly to those in GSM?] A short description of the PISM surges will be added to the revised draft.

[3.1.2 - I'm not sure it's reasonable to try to state a specific justification for why the time scales of this highly idealized and not-observationally-constrained experiment are dissimilar to geological records: many different factors may be in play here (some improving the fit, some to its detriment) and (as an example) saying that the period mismatch is just the result of domain size seems likes its missing a broader set of possibilities.] While several factors influence the period (e.g., bed thermal model, basal temperature ramp, basal hydrology, ...), the domain size seems to be the controlling one here. Previous experiments with a non-downscaled model domain (but otherwise identical experimental setup) yielded results within the limits of geological records. We have added: *'Exploratory GSM runs with a non-downscaled model domain (but otherwise identical experimental setup) yielded results within the limits of geological records.'* to this paragraph.

[3.1.3 - I again struggle with the notion of 'numerical noise'; the paper would be well served by having a much more in-depth description of what is meant by this and where it comes from. With respect to the latter, in most modelling exercises, the numerical error is something that can be well quantified through comparison to exact solutions or by a theoretical analysis of the interpolation properties of numerical method. Yet here, what we're effectively measuring is the system's sensitive dependence to small perturbations. In that sense, it doesn't necessarily follow that stricter convergence criteria (in the case of GSM) would necessarily lead to any less 'noise'. Could an equivalent result be achieved by just adding white noise to the initial conditions?] As mentioned previously, an exact/theoretical solution for hybrid SIA/SSA (as well as full stokes or anything in between) ice dynamics does not exist. A theoretical analysis of the interpolation properties of a numerical method is not straightforward for a coupled non-linear system of thermodynamics/ice dynamics. However, a discussion of this will be added to the revised draft. Model experiments with noise added to the surface temperature are shown in section '3.1.4 Surface temperature noise'. Timestep and resolution convergence experiments are presented in section 3.3. However, since the term 'numerical noise' seems to be causing general confusion, we have added a short definition (see above: L26). Furthermore, we have added *Surge cycling is sensitive to numerical aspects (e.g., numerical solver error).* as an introduction to the numerical noise research question.

[3.1.5 - This section on implicit coupling is so vague as to be useless. What even is 'implicit coupling' in this context? Is this the same as implicit time stepping, i.e. Backward-Euler?]

The text already makes this clear: *'The GSM has a default explicit time step coupling between the thermodynamics and ice dynamics but also includes an optional implicit coupling*

*scheme'* and *'we test the impact of implicit coupling (via an iterative implementation) between the thermodynamics and ice dynamics'*. But to make this even clearer, we've changed the above to:

*'As is standard for thermo-mechanically coupled glaciological ice sheet models, the GSM has a default explicit time step coupling between the thermodynamics and ice dynamics but also includes an optional implicit coupling scheme'* and *'we test the impact of approximate implicit time-step coupling between the thermodynamics and ice dynamics via an iteration between the two calculations for each timestep'*.

[3.2.3 - I can't figure out what TpmTrans, TpmInt, or any other Tpm thing are. If they are described earlier, such a description needs to be here instead or also. If they are not, they need to be defined (and not in the supplement).] We have added more details here: *'In contrast to TpmTrans and TpmInt, the most straightforward approach, TpmCen (Sec. S3.2), determines the grid cell interface temperature by calculating the mean of the two adjacent basal temperatures with respect to the pressure melting point at the grid cell centers (instead of applying the pressure melting point correction after the interpolation of the adjacent grid cell center temperatures)'*. However, we prefer to keep the in depth description including the equations in the supplement.

[3.2.4 - It's not my preference, but if you prefer to have the actual equations describing GSM in the supplement, I suppose that's fine. However we need at least a little bit of a qualitative description of what these different 'weights' imply. What is the context for understanding why these different choices should yield different surging behaviors?] This section refers to Q6. To clarify the purpose and context, we restructured and slightly adjusted this section to: *'Depending on the location of the adjacent minimum grid cell center basal temperature, either the ice flow (when the adjacent minimum basal temperature is downstream) or upstream propagation of the surge should be affected (decreasing basal interface temperature with increasing weight). For the large-scale surges, the adjacent minimum basal temperature is almost exclusively located upstream (e.g., video 02 of Hank [2023]). Changing the weight of the adjacent minimum basal temperature, therefore, affects the surge propagation rather than blocking parts of the ice flow.*

*Here we compare the effect of three different weights on the GSM event characteristics (Eq. (S5)): no consideration of adjacent minimum basal temperature ($W_{Tb,\min} = 0.0$), basal temperature at the interface depends to 50 % on the adjacent minimum basal temperature at the grid cell center (base setup, $W_{Tb,\min} = 0.5$), and basal temperature at the interface is equal to the adjacent minimum basal temperature at the grid cell center ($W_{Tb,\min} = 1.0$).'*

[Fig. 7 - I honestly can't figure out what this figure is trying to convey. Part of this is that I also can't figure out what the part of the text that references it is trying to convey either (L534–540). Please try to make this a little bit more clear.] The reasoning behind Fig. 7 is described in L523-533: *'We complement the above analysis by upscaling the 3.125 km base runs. For example, a 25x25 km grid cell contains a patch of 64 3.125x3.125 km grid cells. The scatter plot of the warm-based fraction (basal temperature with respect to the pressure melting point at 0 °C) and the mean basal temperature with respect to the pressure melting point of the patch can be used to estimate the parameters $T_{ramp}$ and $T_{exp}$ of the basal temperature ramp (Eq. (3)). [...] Consequently, this estimate for the basal temperature ramp should be a lower bound to the points in the scatter plot. [...]'*. The scatter plot described here is what is shown in Fig. 7. To make this clearer, Fig. 7 is now referenced right after the first 'scatter plot'.

[3.3 - Brinkerhoff and Johnson, 2015 suggest that the inclusion of membrane stresses leads to convergence under spatial grid refinement, whereas without them, the SIA does not lead to convergence. Can you place those results in the context of this section? Are the relatively weak convergence results here a result of GSM and PISM velocity solvers being insufficiently 'higher-order'?] Both the GSM and PISM use a velocity dependent switch between pure SIA and a membrane stress approximation. Analysis of GSM sensitivity to SIA/SSA switching rules will be added to the revised draft.

[Sec. 4 If you maintain these research questions as an organizing principle, I would like to see them revisited as they are resolved by the experiments rather than all at once at the

end.] We suspect that not every reader will be interested in every detail of the results section. The summary section provides an easy way to get the most important results and allows the reader to then jump to individual results for more details. Therefore, we would like to keep the summary section.

[L738–746 - I think that this section is kind of weird: none of the results presented in this work actually refute the resolution dependency conclusions of Hindmarsh (2009) or Brinkerhoff and Johnson (2015), yet the paragraph is written as if they did. As mentioned before, it seems just as reasonable to assert that those works saw more robust numerical convergence due to the use of more consistent membrane stress resolution schemes rather than because they fortuitously (or nefariously) made parameter choices that suppressed resolution effects.] This paragraph was added to provide possible explanations for the different conclusions, not necessarily to refute the conclusions of Hindmarsh [2009] or Brinkerhoff and Johnson [2015]. However, the fact that different parameter choices can yield very different results remains. To clarify this, we have updated *'This is in contrast to the findings of other studies examining thermally induced ice streaming [Hindmarsh, 2009, Brinkerhoff and Johnson, 2015]. However, both of these studies analyze just one parameter vector, and it is relatively easy to find a parameter vector for which, e.g., the GSM exhibits only a minor resolution dependence.'* to *'While other studies examining thermally induced ice streaming do not find a strong resolution dependence [Hindmarsh, 2009, Brinkerhoff and Johnson, 2015], these studies are not directly comparable. The different results are likely due to differences in the experimental design. For example, the hybrid SIA/SSA ice dynamics used in the GSM and PISM might lead to a stronger resolution dependence than the schemes used in Hindmarsh [2009], Brinkerhoff and Johnson [2015]. Additionally, both of these studies analyze just one parameter vector, and it is relatively easy to find a parameter vector for which, e.g., the GSM exhibits only a minor resolution dependence.'*

**References**

Taimaz Bahadory and Lev Tarasov. LCice 1.0-a generalized Ice Sheet System Model coupler for LOVECLIM version 1.3: Description, sensitivities, and validation with the Glacial Systems Model (GSM version D2017.aug17). *Geoscientific Model Development*, 11(9):3883–3902, 2018. ISSN 19919603. doi: 10.5194/gmd-11-3883-2018.

D. J. Brinkerhoff and J. V. Johnson. Dynamics of thermally induced ice streams simulated with a higher-order flow model. *Journal of Geophysical Research F: Earth Surface*, 120(9):1743–1770, 2015. ISSN 21699011. doi: 10.1002/2015JF003499.

Reinhard Calov and Ralf Greve. ISMIP HEINO. Ice Sheet Model Intercomparison Project - Heinrich Event INtercOmparison. pages 1–15, 2006. URL http://www.pik-potsdam.de/~calov/heino/he_setup_2006_11_02.pdf.

Reinhard Calov, Andrey Ganopolski, Vladimir Petoukhov, Martin Claussen, and Ralf Greve. Large-scale instabilities of the Laurentide ice sheet simulated in a fully coupled climate-system model. *Geophysical Research Letters*, 29(24):1–4, 2002. ISSN 19448007. doi: 10.1029/2002GL016078.

Reinhard Calov, Ralf Greve, Ayako Abe-Ouchi, Ed Bueler, Philippe Huybrechts, Jesse V. Johnson, Frank Pattyn, David Pollard, Catherine Ritz, Fuyuki Saito, and Lev Tarasov. Results from the Ice-Sheet Model Intercomparison Project-Heinrich Event INtercOmparison (ISMIP HEINO). *Journal of Glaciology*, 56(197):371–383, 2010. ISSN 00221430. doi: 10.3189/002214310792447789.

Johannes Feldmann and Anders Levermann. From cyclic ice streaming to Heinrich-like events : the grow-and-surge instability in the Parallel Ice Sheet Model. *The Cryosphere*, 11:1913–1932, 2017. doi: 10.5194/tc-11-1913-2017.

A. C. Fowler. Sub-temperate basal sliding. *Journal of Glaciology*, 32(110):3–5, 1986. doi: 10.3189/S0022143000006808.

Kevin Hank. Supplementary material for "Numerical issues in modeling ice sheet instabilities such as binge- purge type cyclic ice stream surging", February 2023. URL https://doi.org/10.5281/zenodo.7668490.

Richard C.A. Hindmarsh. Consistent generation of ice-streams via thermo-viscous instabilities modulated by membrane stresses. *Geophysical Research Letters*, 36(6):1–6, 2009. ISSN 00948276. doi: 10.1029/2008GL036877.

K.M. Cuffey and W.S.B. Paterson. *The Physics of Glaciers*. Butterworth-Heinemann/Elsevier, Burlington, MA, 4th edition, 2010. ISBN 9780123694614.

E. Mantelli, M. Haseloff, and C. Schoof. Ice sheet flow with thermally activated sliding. Part 1: the role of advection. *Proceedings of the Royal Society A: Mathematical, Physical and Engineering Sciences*, 475(2231), 2019. ISSN 14712946. doi: 10.1098/rspa.2019.0410.

Elisa Mantelli, Matteo Bernard Bertagni, and Luca Ridolfi. Stochastic ice stream dynamics. *Proceedings of the National Academy of Sciences*, 113(32):E4594–E4600, 2016. doi: 10.1073/pnas.1600362113. URL https://www.pnas.org/doi/abs/10.1073/pnas.1600362113.

Shawn J Marshall and Garry K C Clarke. A continuum mixture model of ice stream thermomechanics in the Laurentide Ice Sheet 2. Application to the Hudson Strait Ice Stream. *Journal of Geophysical Research: Solid Earth*, 102(B9):20615–20637, 1997. ISSN 2169-9356. doi: 10.1029/97jb01189.

Brian D. Papa, Lawrence A. Mysak, and Zhaomin Wang. Intermittent ice sheet discharge events in northeastern North America during the last glacial period. *Climate Dynamics*, 26(2-3):201–216, 2006. ISSN 09307575. doi: 10.1007/s00382-005-0078-4.

A. J. Payne. Limit cycles in the basal thermal regime of ice sheets, 1995.

A. J. Payne, P. Huybrechts, A. Abe-Ouchi, R. Calov, J. L. Fastook, R. Greve, S. J. Marshall, I. Marsiat, C. Ritz, L. Tarasov, and M. P.A. Thomassen. Results from the EISMINT model intercomparison: The effects of thermomechanical coupling. *Journal of Glaciology*, 46(153):227–238, 2000. ISSN 00221430. doi: 10.3189/172756500781832891.

D. Pollard and R. M. DeConto. Description of a hybrid ice sheet-shelf model, and application to Antarctica. *Geoscientific Model Development*, 5(5):1273–1295, 2012. ISSN 1991959X. doi: 10.5194/gmd-5-1273-2012.

A. A. Robel, E. Degiuli, C. Schoof, and E. Tziperman. Dynamics of ice stream temporal variability: Modes, scales, and hysteresis. *Journal of Geophysical Research: Earth Surface*, 118(2):925–936, 2013. ISSN 21699011. doi: 10.1002/jgrf.20072.

William H.G. Roberts, Antony J. Payne, and Paul J. Valdes. The role of basal hydrology in the surging of the Laurentide Ice Sheet. *Climate of the Past*, 12(8):1601–1617, 2016. ISSN 18149332. doi: 10.5194/cp-12-1601-2016.

Ondřej Souček and Zdenek Martinec. ISMIP-HEINO experiment revisited: Effect of higher-order approximation and sensitivity study. *Journal of Glaciology*, 57(206):1158–1170, 2011. ISSN 00221430. doi: 10.3189/002214311798843278.

H. C. Steen-Larsen and D. Dahl-Jensen. Modelling binge-purge oscillations of the Laurentide ice sheet using a plastic ice sheet. *Annals of Glaciology*, 48:177–182, 2008. ISSN 02603055. doi: 10.3189/172756408784700635.

Lev Tarasov, Arthur S. Dyke, Radford M. Neal, and W. R. Peltier. A data-calibrated distribution of deglacial chronologies for the North American ice complex from glaciological modeling. *Earth and Planetary Science Letters*, 315-316:30–40, 2012. ISSN 0012821X. doi: 10.1016/j.epsl.2011.09.010. URL http://dx.doi.org/10.1016/j.epsl.2011.09.010.

Lev Tarasov, Kevin Hank, and Benoit S. Lecavalier. Gsmv01.31.2023 code archive for lissq experiments, February 2023. URL https://doi.org/10.5281/zenodo.7668472.

Florian Andreas Ziemen, Marie Luise Kapsch, Marlene Klockmann, and Uwe Mikolajewicz. Heinrich events show two-stage climate response in transient glacial simulations. *Climate of the Past*, 15(1):153–168, 2019. ISSN 18149332. doi: 10.5194/cp-15-153-2019.

---

## Author Comment (AC3)

**Author's response to Anonymous Referee 2 Comment 1**

April 20, 2023

[1. Currently, the organization of the manuscript feels like a laundry list of simulations completed and a description of the results, without prioritizing the most important and revealing simulations and little discussion for what the results mean and how they relate to other studies. The study also seems to currently have about 10-15 areas of focus, making it challenging to see which end up actually being important. My recommendation would be to take all the description of simulations where numerical choices don't appear to have much influence on the results and move them into a supplement. They can be summarized in a few sentences perhaps at the end of the results section, but right now they add too much extra to the manuscript making it longer and hard to get through. This includes the sediment-hard bed transition, basal hydrology, basal hydrology instead of temperature ramp and the max time step.] Discussing numerical choices that do not significantly affect the results is equally important. However, we agree that the in-depth discussion can be moved to the supplement. Following the idea of the research questions, the results section will be restructured into the following three main themes: 1) strictly numerical aspects (Q1 and Q11), 2) numerical/modeling choices with a significant effect on the results (Q3, Q4, Q5, Q8), 3) numerical/modeling choices without a significant effect (Q2, Q6, Q9, Q10). As suggested, the third theme will only be a short summary with the details in the supplement.

[2. The way the results are currently described and presented also contributes to the challenge of reading through this manuscript. There are 15 tables, and it is difficult to understand what all the numbers in the tables mean. Figures 4 and 6 seem like a more intuitive way to present this information (though all the markers and line and shading in Figure 6 need explicit descriptions in the caption and perhaps a legend to be interpretable). There really shouldn't be more than a few tables in the main text of this manuscript, the rest should be relegated to a supplement.] Figures in the form of Figure 4 are not suitable to replace the tables because they only show the results of one parameter vector. The restructuring of the results section described above will remove two tables from the main manuscript, and, where possible, we will replace the tables with plots similar to Figure 6. Most information necessary to interpret Figure 6 was already in the caption. However, to make it even clearer, we slightly updated the caption to: *Percentage differences in event characteristics compared to the GSM base setup ($T_{ramp} = 0.0625$, $T_{exp} = 28$) for different basal temperature ramps at 3.125 km horizontal grid resolution (average of the 5 parameter vectors). The ramps are sorted from widest (first row) to sharpest (last row, see Fig. S25 for a visualization of all ramps). The different colors were added for visual alignment of the individual basal temperature ramps, and the horizontal bars represent the standard deviations. The shaded pink regions mark the numerical noise estimates (Tab. 5) and the black numbers in the title of each subplot represent the mean values of the base setup. No runs crashed and all runs had more than 1 surge event. The first 20 kyr of each run are treated as a spin-up interval and are not considered in the above. The x-axis is logarithmic. The exact values are given in Tab. S5.*

[3. The "such as" in the title of the paper is misleading. The only type of ice sheet instability discussed and tested in this paper is a thermal oscillatory instability (or B-P). A more accurate title would be simpler: "Numerical issues in modeling binge-purge type cyclic ice stream surging"] Changed to 'Numerical issues in modeling ice stream surge cycling'.

[4. Related to #3 and throughout - the term "binge-purge" oscillations has largely fallen out of favor in the ice sheet modeling community. These are more commonly called thermal oscillations or ice stream activation-stagnation cycles. It is OK to mention in the introduction that these have historically been referred to as binge-purge, but it isn't in keeping with the field to continue to refer to them as such throughout.] We are not aware of such a development [e.g., Roberts et al., 2016, Feldmann and Levermann, 2017, Ziemen et al., 2019, Schannwell et al., 2023]. Furthermore, the term *'thermal oscillations'* disregards the effect of basal hydrology and *ice stream activation-stagnation cycles* seems to be just a lengthy expression for *binge-purge cycles*. However, we will use *ice stream surge cycling* instead of *binge-purge cycling*.

[5. I am quite confused over how numerical noise is defined in this manuscript. It seems in section 3.1.3 that to quantify numerical noise that different solver choices are used. However, depending on the number of iterations occuring between the different tolerances (and the details of how the solver works) it would seem that this method could yield strongly different estimates for the "noise". Additionally, it is unclear why this is the correct bar for determining whether a change is "important", instead of a more physically meaningful quantity. Additionally, the rationale behind the set of simulations detailed in Table 6 is confusing. In a bitwise reproducible code, I don't see why the number of cores used in a simulation should have any influence on the simulations. This makes me concerned about the robustness of other simulations if the number of cores has such an important influence on the results. How reproducible are these results by other researchers? If the same setup is run on a different cluster architecture would the results be different?] Note that *These noise estimates set a minimum threshold for discerning physical significance of changes in surge characteristics due to physical model components*. We are unsure how a *physically meaningful quantity* could determine whether the modeled result is due to numerical aspects or a true physical phenomenon. However, we have tested the effect of different tolerances, and a short statement will be added to the revised draft. Additionally, we have added the following sentence to the description of the numerical noise: *By numerical noise, we refer to any non-physical differences in the model solution induced by numerical aspects such as rounding errors and convergence criteria of the numerical solver*.

When dealing with non-linear transitions such as surge onset and shutdown, small differences can accumulate and lead to a somewhat different result at the end of the model run. These small differences can be caused by, e.g., a different number of cores (see **online PETSc-FAQ link**). PETSc is used by PISM. So the results might be somewhat different on a different cluster. However, the whole point of introducing the numerical noise threshold is to identify what characteristics and relations are likely to be robust.

[6. In section 4 I see the summary of results, but very little discussion of what they mean (in some cases but not others). This seems to me to be the main missing piece of the manuscript to be useful to other researches, a discussion of how these results relate to the theory of thermal oscillations (from Schiavi, Mantelli, Robel, MacAyeal, etc.) and how they might relate to other ice sheet models.] It is not straightforward to compare our results to the existing theory because the theory in this context is not fully developed. However, a more in-depth discussion will be added to the revised draft.

**References**

Johannes Feldmann and Anders Levermann. From cyclic ice streaming to Heinrich-like events : the grow-and-surge instability in the Parallel Ice Sheet Model. *The Cryosphere*, 11:1913–1932, 2017. doi: 10.5194/tc-11-1913-2017.

William H.G. Roberts, Antony J. Payne, and Paul J. Valdes. The role of basal hydrology in the surging of the Laurentide Ice Sheet. *Climate of the Past*, 12(8):1601–1617, 2016. ISSN 18149332. doi: 10.5194/cp-12-1601-2016.

C. Schannwell, U. Mikolajewicz, F. Ziemen, and M.-L. Kapsch. Sensitivity of heinrich-type ice-sheet surge characteristics to boundary forcing perturbations. *Climate of the Past*, 19(1):179–198, 2023. doi: 10.5194/cp-19-179-2023. URL https://cp.copernicus.org/articles/19/179/2023/.

Florian Andreas Ziemen, Marie Luise Kapsch, Marlene Klockmann, and Uwe Mikolajewicz. Heinrich events show two-stage climate response in transient glacial simulations. *Climate of the Past*, 15(1):153–168, 2019. ISSN 18149332. doi: 10.5194/cp-15-153-2019.

---

## Author Response (AR1)

**Author's response to Anonymous Referee 1 Comment 1**

June 27, 2023

**1 General Comment**

[I think that this manuscript has the potential to be a useful contribution to the (slowly) growing body of literature on the relationship between thermodynamics and sliding. However, I think that this manuscript needs quite a bit of work to be publishable. My criticism is not generally aimed at the science, but rather at the presentation which reads more like a set of internal notes than a manuscript. In particular the organization does not proceed in a logical manner, and I struggle to find a narrative that relates adjacent sections. In addition, the manuscript includes difficult-to-remember jargon and sometimes dubious interpretations of referenced material. There are far too many tables; it is very difficult to make sense of tabular numbers appearing in columns that reference one of many difficult to differentiate experiments. Perhaps most critically, some of the most important figures - many of which are referenced frequently and are essential to the paper's conclusions - are relegated to supplemental material or appear in one of the authors' previous papers. I do not suggest any new experiments or analysis here. However, I do suggest that the authors revise this manuscript with a keen eye for the fact that it will, ultimately and hopefully, be read by others.]

We thank the referee for their constructive comments. We have addressed all the referee's comments and revised the text accordingly. A point-by-point reply is reported below, with referee comments in orange (line references refer to the old manuscript version) and our replies in black.

We agree that the readability of the paper benefits from a general restructuring and focusing on the key takeaways (Numerical Model Error Estimates (MNEEs) must be considered when numerically modeling ice stream surge cycles, surge characteristics are sensitive to the basal sliding activation criterion, resolution dependency can be reduced by incorporating a resolution-dependent basal sliding activation criterion). To streamline the paper, we grouped the 11 research questions into three main themes: 1) determine the 'MNEEs' as a metric to determine the significance of a change in model configuration (note that we use the MNEEs because we are not able to determine the model error; we provide more details on this issue in our replies below), 2) using the MNEEs, determine the sensitivity of the surge characteristics (number of surges, mean surge periodicity, mean surge duration, and mean ice volume change during a surge) to model aspects with a physical relevance (particular focus on the basal sliding activation criterion), and 3) perform a convergence study based on the results of 1) and 2). We also added a short paragraph describing the narrative of the sensitivity experiments in 2) (see our reply to comment *Sec 1.1* below).

Additionally, we replaced most of the tables in the results section with the two new figures 6 and 7. These figures summarise the sensitivities of the surge characteristics of all model components with a significant effect in a concise way. In terms of model components analyzed, we are now focusing only on model components that have significant effects on the surge characteristics in order to reduce the burden on the reader, as suggested by the reviewer. Additional model components that we showed to have no influence were moved to the supplement and are only briefly summarized in the main manuscript. Figure S18 showing the definition of the surge

characteristics (previously situated in the supplement) has been moved to the main manuscript (Figure 3).

Where possible, we changed the jargon to phrases that are easier to interpret and remember (e.g., 'numerical noise estimates' to 'Numerical Model Error Estimates (MNEEs)', 'base setup' to 'reference setup', 'event'/'surge event' to 'surge', 'event characteristics' to 'surge characteristics').

We reworked the summary and conclusions section to remove the dubious interpretations of referenced material and give a more extensive and accurate comparison to the existing literature (e.g., added comparison to the EISMINT temperature spokes).

**2 Detailed comments**

[Title - The title should be shortened to something like 'Numerical issues in modeling binge-purge behavior in ice streams.' No other instabilities are addressed and 'binge-purge', 'cyclic', and 'surging' are all redundant.] In order to streamline the title, we changed it from: *Numerical issues in modeling ice sheet instabilities such as binge-purge type cyclic ice stream surging*

to: *Numerical issues in modeling thermally and hydraulically driven ice stream surge cycling.*

We kept the term 'surge cycling' because 'surging' alone could also refer to just a single surge, and the analysis of multiple surges per run is a key aspect of this study.

[L1 - Delete 'as in any environmental system'.] Agree and done.

[revised manuscript text omitted]

The corresponding part in Sec. 2.3 states: *We compute the new 'Minimum Numerical Error Estimates' (MNEEs) metric by examining the model response to changes in the model configuration that are not part of the physical system. The MNEEs are defined as the percentage differences in surge characteristics when applying a stricter (than default) numerical convergence in the GSM and adjusting the matrix solver used in PISM (changing the number of processor cores used). They are then used as a threshold to determine if model sensitivities to changes in the model configuration that affect the physical system (e.g., the inclusion of a bed thermal model or sliding depence on effective pressure from basal hydrology) are above the level of background noise induced by iterative numerical solvers in the model. We refrain from drawing conclusions about the effects of a change in model configuration with physical relevance when the model sensitivities in question are smaller than the MNEEs. In these cases, the actual physical response of the model might be hidden within the numerics.*

*While the MNEEs are useful to our purpose, we wish to emphasize that they can not replace proper model verification and validation and are missing uncertainties due to, e.g., different approximations of the Stokes equations and other physical processes not included in the models. Nonetheless, they provide a minimum estimate of the numerical model error, which is still a significant improvement over ignoring this issue entirely.*

[L26 - What is 'numerical noise'? Something random? Pseudo-randomness in a chaotic system? Numerical error? This is a critical consideration but it's not really clear what it means in this paper here and elsewhere.] We agree that the concept of 'numerical noise' needed clarification. We changed the term 'numerical noise estimates' to 'Numerical Model Error Estimates (MNEEs)'. Refer to our answer to comment *L20 - I think it is strange to lead with validation as a means of motivating the current manuscript, when no validation occurs here. Validation is not the same as sensitivity testing.* for the details that have been added to the revised draft.

[L38 - Define 'meaningful'] 'Meaningful' in the sense of 'physically meaningful', i.e., that the modeled surges are not just due to numerical instabilities. However, the sentence works just fine without this phrase, and we, therefore, chose to remove it for clarity.

[L44 - The quote from Soucek and Martinec is relevant, but it misses the fact that there are a great many approximations that appear in, for example, the solution of the Stokes' equations. Why the emphasis on numerical rather than model error?] We agree that one is ultimately interested in the model rather than the numerical error. However, the analytical solution for the here used 3D thermo-mechanically coupled models with hybrid SIA/SSA ice dynamics is not fully developed, and it is not the goal of this study to progress the theory. Note that this is also the case for other ice sheet models with different approximations, e.g., BISICLES (sliding everywhere, minimal heat treatment at the bed (just a vertical flux, which in temperate regions

produces water)) or ELMER ice (too expensive for ensembles needed). Therefore, the model error cannot be determined in this context.

To provide at least some uncertainty estimates, we determine the MNEEs in the models by changing purely numerical model components (no physical relevance). Additionally, uncertainties associated with the numerical aspects of a model have received limited attention (compared to the effect of different approaches to the Stokes equations).

See also our answer to comment *L20 - I think it is strange to lead with validation as a means of motivating the current manuscript, when no validation occurs here. Validation is not the same as sensitivity testing.* for a revised version of this paragraph. Note that the quote from Souček and Martinec [2011] is now situated later in the text.

[L60–72 - While I recognize that this paper focuses on the ISMIP-HEINO setup, it would be worthwhile to try to contextualize this work with respect to the EISMINT-F experiments as well. There is a great deal of insight there regarding thermal sliding instabilities and the circumstances under which they appear.] The EISMINT-F experiment has an air temperature 15 K cooler than the reference experiment A but otherwise the same boundary conditions. Due to this colder air temperature, almost all examined models show cold ice spokes in the basal temperature fields extending into the melt zone. These spokes break the radial symmetry of the model results. While the effects are most pronounced in the temperature fields, the spoked pattern also exists in the ice velocity, flow factor, and ice thickness fields.

Our work is similar in that we use a comparable experimental design with respect to the boundary conditions, especially for the PISM experiments, but different because of the sediment distribution and the ice dynamics used in the models (SIA in EISMINT vs. hybrid SIA/SSA here). Bueler et al. [2007] show that when the derivative of the strain-heating term to the temperature field is horizontally smoothed, the spokes can be eliminated. They also discuss the possibility that the inclusion of membrane stresses provides a physical mechanism for this smoothing. Additionally, Bueler and Brown [2009] hypothesize that the spokes occurring in EISMINT experiment H (basal slip where the basal ice is at the melting point vs. no basal slip in Experiment A and F), are caused by a velocity discontinuity introduced by traditional SIA sliding laws.

While it is true that there is a lot that can be learned from the EISMINT-F and H experiments, we do not see basal temperature spokes in our experiments. Nevertheless, this is an interesting comparison, and we added the following paragraph to the conclusions: *Basal temperature spokes, such as the ones modeled in the EISMINT-F and H experiments [Payne et al., 2000], are not apparent in the PISM experiments. The GSM runs show some warm-based areas at the margins interspersed by colder regions, but this is likely due to a steep surface slope leading to a large driving stress, high velocity, and then consequently, a basal temperature increase. Therefore, neither the PISM nor GSM instabilities discussed here are comparable to the EISMINT temperature spokes. The absence of basal temperature spokes is likely due to the inclusion of membrane stresses in the ice dynamics of both models [Bueler et al., 2007, Bueler and Brown, 2009].*

[Sec 1.1 - I find the organization of the paper according to research question to be quite challenging to follow, perhaps mostly because there are so many (11) research questions. I think it would be better to group these into open questions rather than yes/no, and this might make for more comprehensible themes. For example, Q1,11 could be grouped as 'what aspects of simulated surges are due to numerical considerations?', while Q2–6,9 could be grouped as 'what modeling and solution choices influence surging?' and Q7,8,10 as 'what parameterizations of basal physics leads to the most robust conclusions?' or something similar.] We agree that the manuscript needed some restructuring to make it more accessible for the reader. To increase the readability, we grouped the individual research questions into the following main themes: 1) MNEEs (Q1), 2) sensitivity experiments with (previous Q2, Q3, Q4, Q5, Q7, Q8; now Q2, Q3, Q5, Q6, Q7, Q8) and without a significant effect on the results (previous Q6, Q9, Q10; now Q4, Q9, Q10), and 3) convergence study (Q11). As suggested by the second reviewer,

the sensitivity experiments without a significant effect will be only briefly summarised, with the details moved to the supplement. Since the results of the convergence study depend on previous experiments, this aspect is discussed last.

Within the above mentioned 3 main themes, and counter to the request of the reviewer, we decided to keep the specific research questions, but added the following paragraph to better link the different research questions within the sensitivity experiment section: *Here we aim to determine the significance of different model configurations on the surge characteristics. We are particularly interested in model configurations affecting the basal temperature and thus the surge behavior. Therefore, we first discuss the change in surge characteristics due to a bed thermal model ($Q_2$) and modeling choices affecting the basal temperature at the grid cell interface where the ice velocities are calculated ($Q_3$ and $Q_4$), including the basal sliding thermal activation criterion ($Q_5$). Previous studies examining the effects of ice stream behavior are often based on an idealized basal topography and sediment distribution and do not consider sub-glacial hydrology [e.g., Calov et al., 2010, Brinkerhoff and Johnson, 2015]. Therefore, we determine the change in surge characteristics due to these aspects in $Q_6$, $Q_7$, $Q_8$, and $Q_9$, respectively. Since thermally and hydraulically driven ice stream surges are not exclusive, we also investigate the differences between the two mechanisms when used as the primary smoothing mechanism at the warm/cold-based transition zone ($Q_{10}$).*

We believe that this organization allows the reader to jump right to the parts they are most interested in, now without detracting from the readability of the manuscript. Anyways, should the Referee find that our preferred paper organization is still challenging to follow, we are open to eliminating the research question structure altogether.

[L166 and elsewhere - I don't find it helpful to reference a manuscript that is 'in preparation' because such manuscripts are not readable and sometimes fail to ever get published. Is there some source code that could be referenced instead? An instruction manual? An older manuscript from which the ideas in the in prep manuscript are adapted?] We agree that referencing a manuscript 'in preparation' is not ideal. We hoped this manuscript would be available as a preprint by now, which is not the case. Therefore, we removed this reference.

The source code of the model version used in this manuscript can be found in the supplementary material [Tarasov et al., 2023] as stated in the Code and data availability section. Additionally, we have added this reference to the GSM description section. Older manuscripts on which the current GSM version is based on are also mentioned in this section [e.g., Pollard and DeConto, 2012, Tarasov et al., 2012, Bahadory and Tarasov, 2018].

[L198 - I am deeply skeptical of a model that 'activates' stress terms based on a heuristic that in turn depends on whether the bed is soft or hard (whatever that means). Does it not seem that such an obviously non-physical choice could lead to just as much variability in surging behavior as any of these other mechanisms? Validation is mentioned in the introduction, but what about verification? How does the reader know (especially given that there is no current reference to the model description) that this model converges to the true solution of some physically and mathematically justified system of equations under discretization refinement?] Thanks for raising this point. First of all, there was a mistake in the GSM description. The model version used within this paper does not differentiate between soft (100 % sediment cover) and hard (0 % sediment cover) beds for the SSA activation criteria. However, different SSA activation criteria are available in the GSM and we add a table showing the model sensitivity to this choice to the revised supplement, including a model setup with active SSA everywhere (Sec. S1.2).

The reference to the hybrid SIA/SSA ice dynamics of the GSM is [Pollard and DeConto, 2012]. See also our answer to comment *L20 - I think it is strange to lead with validation as a means of motivating the current manuscript, when no validation occurs here. Validation is not the same as sensitivity testing.* for a discussion about the difficulties regarding model verification and validation.

[L215 - I don't know what 'legacy' means here.] Changed to 'values used in previous GSM

modeling studies [e.g., Bahadory and Tarasov, 2018]'.

 We compare our definition of $F_{\mathrm{warm}}$ to the one used by Fowler [1986], Mantelli et al. [2019], so $F_{\mathrm{warm}}$ is correct here. To make this clearer, we changed: *A temperature ramp similar to the one suggested by Fowler [1986] and later Mantelli et al. [2019] [...]*

to: *For comparison, a temperature ramp similar to the one suggested by Fowler [1986] and later Mantelli et al. [2019] [...]*

We agree that the previous notation was confusing and replaced $F_{T_{ramp}}$ by $P_{T_{ramp}}$.

 For precision/accuracy and lack of alternatives, we prefer to stick to the phrase 'parameter vector'. Note that a parameter vector and ensemble member are not the same (e.g., multiple ensemble members can have the same parameter vector).

 Note that the benefits of an ensemble are explained earlier in the text: *'To partly address potential non-linear dependencies of surge cycling on model parameters, we use a high variance subset of 5 base GSM parameter vectors (each comprising 8 model input parameters) for our numerical experiments'.*

To make it clearer that each experiment is run with all parameter vectors, we updated this part to: *'In order to partly address potential non-linear dependencies of surge cycling on model parameters, we run each of our numerical experiments with a high variance ensemble of 5 GSM parameter vectors (each comprising 8 model input parameters) and 9 PISM parameter vectors (each comprising 6 model input parameters).'*

We have also added the following sentence as an introduction to section 2.1.2: *'Each GSM experiment is run with an ensemble based on 5 input parameter vectors.'*. Furthermore, we added *'Each PISM experiment is run with an ensemble based on these 9 input parameter vectors.'* to section 2.2.2.

 We agree that the word 'reference' is easier to interpret than 'base'. However, a 'reference simulation' and 'base setup' are not the same thing. In this study, there are 5 reference simulations (one for each parameter vector) but only one base setup. To avoid potential confusion, we are now using 'reference setup'.

 The forward referencing was meant to guide the reader and allow them to skip sections they are not interested in. However, this is better suited at the end of the introduction and was removed here.

 We agree that the climate forcing is an important detail and changed the text from: *'The GSM is run with an idealized downscaled North American geometry (Fig. 1, modified after the ISMIP-HEINO setup [Calov and Greve, 2006]) and simplified climate representation. The temperature forcing is defined by a domain wide surface temperature (rTnorth, Tab. 1) and a specified vertical temperature gradient (atmospheric lapse rate (lapsr in Tab. 1)). The surface temperature forcing is asymmetric in time (Fig. S1), enabling the analysis of the timing of cycling onset and termination under*

*different physical and numerical conditions (a comparison of ice stream ice volume evolution under constant and assymetric temperature forcing is shown in Fig. S2 for one parameter vector)'.*

to: *The GSM is run with an idealized down-scaled North American geometry (Fig. 1, modified after the ISMIP-HEINO setup [Calov and Greve, 2006]) and simplified climate representation. The surface temperature forcing in the GSM is given by*

$$T_{\text{surf}} = \text{rTsurf} + \text{lapsr} \cdot H + T_{\text{asym}}, \tag{1}$$

*where rTsurf and lapsr are input parameters for the domain-wide surface temperature constant and atmospheric lapse rate, respectively (Table 1), H the ice sheet thickness, and $T_{\text{asym}}$ the asymmetric (in time) temperature forcing (maximum difference of $10°C$, orange line in Fig. S1) calculated according to*

$$T_{\text{asym}} = \left| \left( \frac{t}{200 \text{ kyr}} \cdot 3 + 2 \right) - 1 \right| \cdot 5°\text{C}, \tag{2}$$

*where t is the model time ranging from $-200$ kyr to $0$ kyr (instead of $0$ kyr to $200$ kyr). The asymmetric temperature forcing enables the analysis of the timing of cycling onset and termination under different physical and numerical conditions (a comparison of ice stream ice volume evolution under constant and asymmetric temperature forcing is shown in Fig. S2 for one parameter vector).*

*The surface mass balance forcing is then determined by*

$$M_{\text{tot}} = M_{\text{acc}} - M_{\text{melt}}, \tag{3}$$

*where $M_{\text{acc}}$ and $M_{\text{melt}}$ are the surface accumulation and melt, respectively. The surface accumulation is defined by*

$$M_{\text{acc}} = \text{precRef} \cdot \exp \left( \text{hpre} \cdot T_{\text{surf}} \right), \tag{4}$$

*where precRef and hpre are the precipitation coefficient input parameters. Surface melt is calculated according to a Positive Degree Day (PDD) approach:*

$$M_{\text{melt}} = \text{rPDDmelt} \cdot \max \left( 0.0, \text{POSdays} \cdot (T_{\text{surf}} + 10.0°\text{C}) \right), \tag{5}$$

*where rPDDmelt is the input parameter for melt per PDD and the PDD constant POSdays is set to $100$ days $\text{yr}^{-1}$. Note that we set $T_{\text{surf}} = 0.1°C$ and $M_{\text{tot}} = -100$ m $\text{yr}^{-1}$ for ocean grid cells, and $T_{\text{surf}} = 0.1°C$ and $M_{\text{tot}} = -200$ m $\text{yr}^{-1}$ at the boundaries of the model domain.*

However, Fig. S1 only shows the asymmetric aspect of the temperature forcing (atmospheric lapse rate and parameter vector dependency are not considered). Due to the simplicity of the plot, we do not deem it important enough to be in the main manuscript.

Generally speaking, we moved information to the supplement when we suspected most readers would not be interested in reading about it. In all of these cases, we added a reference to the corresponding part in the supplement to ensure the reader is aware that additional information is available and where to find it. Where possible, we replaced the references to the supplement with references to the two new Fig. 6 and 7.

However, the information in the supplement is not essential to understanding the paper. Following this logic, we prefer to keep the remaining references to the supplement.

[2.2.1 - It's strange to imply that PISM is not also optimized for computational speed. It's the parallel ice sheet model, after all.] Thanks for bringing this up. It was not the intent of this statement to imply that PISM is not optimized. While both models are optimized, the optimizations are not for the same contexts. The idea behind using two different models is to minimize the possibility that drawn conclusions are solely a result of the used optimization schemes. Furthermore, the GSM is optimized for computational resource use to enable large ensembles over paleo timescales and therefore not parallelized (not the case for PISM).

To make this clearer, we have adjusted: *'The GSM is an ice sheet model developed specifically for glacial cycle ensemble modeling. The GSM is therefore numerically optimized for computational speed.'*

to: *'In contrast to the GSM, the Parallel Ice Sheet Model (PISM) is not specifically developed for glacial cycle ensemble modeling. Therefore, the two models use distinct sets of numerical optimizations for computational speed.'*.

 Agree and done.

Given the varying meanings of 'stable', we removed *'[...], indicating a stable solution of the numerical matrix solver even for runs with very high velocities.'*.

Yes, the corresponding part in K.M. Cuffey and W.S.B. Paterson. [2010] is: *'Speeds and displacements also vary widely. High velocities are about 100 m/day for short periods, and 5 km/yr maintained for one or two years. Low velocities are only several tens to a few hundred meters per year, values typical of many nonsurging glaciers'*. However, we now use the more appropriate comparison to Jakobshavn Isbræ: *For comparison, observed outlet glacier velocities at Jakobshavn Isbræ(Greenland) approach* 20 km yr$^{-1}$ *[Joughin et al., 2012, 2014]*.

The paragraph: *Excluding runs that show maximum sliding velocities* > 50 km yr$^{-1}$ *from the analysis yields similar results to the full* 10-member ensemble (Sec. S6 and Fig. S10), indicating a stable solution of the numerical matrix solver even for runs with very high velocities. In addition, the 50 km yr$^{-1}$ is exceeded no more than 7 times per 200 kyr run (100 yr output) and the maximum sliding velocities are generally within the observed range (Fig. S10). was removed from the revised draft because we now impose a maximum sliding velocity of 40 km/yr for all PISM runs (same as for the GSM). The analysis for higher velocities is, therefore, no longer needed.

 They are not interchangeably. The term HE should be exclusively used when referring (to at least some extent) to the ocean sediment records/IRD layers. The abbreviation 'HE' and most instances of the term 'Heinrich Event' were removed. The term 'event' was replaced by the more precise term 'surge'.

 Agree and done.

 Agreed. We changed: *'a large fraction of the pseudo-Hudson Strait area is ice-free when no surge occurs'*

to: *'a large fraction of the pseudo-Hudson Strait area is only ice-covered when a surge occurs'*

 This was indeed missing from the manuscript and will help the reader to better understand later sections. We added a plot and the following short description of the PISM surges to the revised draft (Sec. 3.1.1 and Fig. 5): *Surges in the PISM originate at the ice sheet margin in the soft-bedded pseudo-Hudson Strait (exact position varies between runs) and propagate towards the center of the pseudo-Hudson Bay (x* = 1300 km, *y* = 1500 km, *Fig. S8 and 5). The ice near the margin is already flowing downstream before the start of the surge (t* = 89.36 kyr*). However, the basal temperature is below the pressure melting point, and the ice velocities are low (*< 100 m yr$^{-1}$*). As the ice sheet upstream of the margin thickens, the warm-based area extends further downstream, particularly along the* 100 % *soft-bedded contour line (magenta line in Fig. 5). Once the warm-based area connects with the margin (t* = 89.42 kyr*), the ice velocities increase beyond* 100 m yr$^{-1}$*, activating the SSA (Sec. 2.2.1). Similar to the surges in the GSM, the sliding velocities then increase rapidly, quickly extending the warm-based area (t* = 89.43 kyr *and t* = 89.433 kyr*). The surge propagates upstream into the pseudo-Hudson Bay and the ice is transported along the pseudo-Hudson Strait into regions with increasingly negative surface mass balance rates (t* = 89.435 kyr *to t* = 89.45 kyr*, Fig. S7). The ice sheet*

*thins, the basal temperature at the margin falls below the pressure melting point, blocking parts of the upstream ice stream, and the surge ceases at $t = 89.47$ kyr ($\sim 100$ yr surge duration). The ice volume in the surge-affected area continues to decrease for, on average, another $2.5$ kyr due to the large amounts of ice in the negative surface mass balance regions. In contrast to the GSM, the PISM results remain symmetrical about $y = 1500$ km throughout the surge.*

[3.1.2 - I'm not sure it's reasonable to try to state a specific justification for why the time scales of this highly idealized and not-observationally-constrained experiment are dissimilar to geological records: many different factors may be in play here (some improving the fit, some to its detriment) and (as an example) saying that the period mismatch is just the result of domain size seems likes its missing a broader set of possibilities.] While it is true that several factors influence the period (e.g., bed thermal model, basal temperature ramp, basal hydrology, ...), the domain size seems to be the controlling one here. Previous experiments with a non-downscaled model domain (but otherwise identical experimental setup) yielded results within the limits of geological records. We changed this part from: *Due to the downscaled GSM domain, the mean modeled GSM period is shorter than the observed period of, on average, 7 kyr [K.M. Cuffey and W.S.B. Paterson., 2010].*

to :*'The mean modeled GSM period is shorter than the observed period of, on average, 7 kyr [K.M. Cuffey and W.S.B. Paterson., 2010]. However, exploratory GSM runs with a dimensionally accurate (not downscaled) model domain (but otherwise identical experimental setup) yielded periods within the range of geological inferences.'.*

[3.1.3 - I again struggle with the notion of 'numerical noise'; the paper would be well served by having a much more in-depth description of what is meant by this and where it comes from. With respect to the latter, in most modelling exercises, the numerical error is something that can be well quantified through comparison to exact solutions or by a theoretical analysis of the interpolation properties of numerical method. Yet here, what we're effectively measuring is the system's sensitive dependence to small perturbations. In that sense, it doesn't necessarily follow that stricter convergence criteria (in the case of GSM) would necessarily lead to any less 'noise'. Could an equivalent result be achieved by just adding white noise to the initial conditions?] To avoid repetition, please refer to our reply to comment *L20 - I think it is strange to lead with validation as a means of motivating the current manuscript, when no validation occurs here. Validation is not the same as sensitivity testing.* for a discussion about the difficulties regarding the determination of the model error.

The main idea behind using stricter convergence criteria was not to decrease the overall noise level, especially since we do not know the exact solution and, therefore, can not determine the noise level. Instead, we want to show (by changing purely numerical model components) that small differences in the surge characteristics between different model setups do not necessarily have a physical origin and might just be due to numerical errors. We added the following paragraph to the revised draft: *Note that the goal of these experiments is not necessarily to decrease the model error, especially since we do not know the exact solution and, therefore, can not determine the model error. Instead, we aim to show (by changing purely numerical aspects) what the minimum numerical errors are for each surge characteristic..* Furthermore, we have added *Surge cycling is sensitive to numerical aspects (e.g., numerical solver error).* as an introduction to the MNEE research question.

Model experiments with noise added to the surface temperature are shown in section '3.2.1 Adding surface temperature noise' (3.1.4 in the old manuscript). The differences in surge characteristics are generally smaller than for the experiments with a stricter numerical convergence criterion.

[3.1.5 - This section on implicit coupling is so vague as to be useless. What even is 'implicit coupling' in this context? Is this the same as implicit time stepping, i.e. Backward-Euler?] Implicit coupling (in contrast to the default explicit time step coupling) refers to the coupling between the thermodynamics and ice dynamics part of the model. To make this clearer in the text, we changed: *'The GSM has a default explicit time step coupling between the thermody-*

*namics and ice dynamics but also includes an optional implicit coupling scheme'*

to: *'As is standard for thermo-mechanically coupled glaciological ice sheet models, the GSM has a default explicit time step coupling between the thermodynamics and ice dynamics but also includes an optional implicit coupling scheme (c.f. Sec. 3.2.2).'*

and: *'we test the impact of implicit coupling (via an iterative implementation) between the thermodynamics and ice dynamics'*.

to: *'we test the impact of approximate implicit time step coupling via an iteration between the two calculations for each time step.'*.

[3.2.3 - I can't figure out what TpmTrans, TpmInt, or any other Tpm thing are. If they are described earlier, such a description needs to be here instead or also. If they are not, they need to be defined (and not in the supplement).] Since it is the most concise way, we embedded the equations describing the three approaches (previously in the supplement) into section 3.3.2 (3.2.3 in the old manuscript). Following this suggestion, we have also added the equations for the local basal hydrology to section 3.3.6 (3.2.6 in the old manuscript). Should the Referee find this too detailed, we are open to moving the equations back to the supplement and working on another solution.

[3.2.4 - It's not my preference, but if you prefer to have the actual equations describing GSM in the supplement, I suppose that's fine. However we need at least a little bit of a qualitative description of what these different 'weights' imply. What is the context for understanding why these different choices should yield different surging behaviors?] Due to the restructuring of the paper, this entire section has been moved to the supplement (no significant effect on the surge characteristics, now section S8.1). Although the equations and analysis are in the same document now, we restructured and slightly adjusted this section from: *Here we compare the event characteristics for three different weights when calculating the basal interface temperature in the GSM (Eq. (S5)): no consideration of adjacent minimum basal temperature ($W_{Tb,\min} = 0.0$), basal temperature at the interface depends to 50 % on the adjacent minimum basal temperature at the grid cell center (base setup, $W_{Tb,\min} = 0.5$), and basal temperature at the interface is equal to the adjacent minimum basal temperature at the grid cell center ($W_{Tb,\min} = 1.0$).*

*Depending on the location of the adjacent minimum grid cell center basal temperature, either the ice flow (when the adjacent minimum basal temperature is downstream) or upstream propagation of the surge should be affected. For the large-scale surges, the adjacent minimum basal temperature is almost exclusively located upstream (e.g., video 02 of Hank [2023]). Changing the weight, therefore, affects the surge propagation rather than blocking parts of the ice flow.*

to: *'Depending on the location of the adjacent minimum grid cell center basal temperature, either the ice flow (when the adjacent minimum basal temperature is downstream) or upstream propagation of the surge should be affected (decreasing basal interface temperature with increasing weight). For the large-scale surges, the adjacent minimum basal temperature is almost exclusively located upstream (e.g., video 02 of Hank [2023]). Changing the weight of the adjacent minimum basal temperature, therefore, affects the surge propagation rather than blocking parts of the ice flow.*

*Here we compare the effect of three different weights on the GSM event characteristics (Eq. (S1)): no consideration of adjacent minimum basal temperature ($W_{Tb,\min} = 0.0$), basal temperature at the interface depends to 50 % on the adjacent minimum basal temperature at the grid cell center (base setup, $W_{Tb,\min} = 0.5$), and basal temperature at the interface is equal to the adjacent minimum basal temperature at the grid cell center ($W_{Tb,\min} = 1.0$).'*

[Fig. 7 - I honestly can't figure out what this figure is trying to convey. Part of this is that I also can't figure out what the part of the text that references it is trying to convey either (L534–540). Please try to make this a little bit more clear.] Fig. 7 (Fig. 10 in the revised draft) shows the results of the upscaling experiment. The reasoning behind this figure is described in L523-530 (old manuscript): *'We complement the above analysis by upscaling the 3.125 km base runs. For example, a 25x25 km grid cell contains a patch of 64 3.125x3.125 km grid cells. The scatter plot of the warm-based fraction (basal temperature with respect to the pressure melting*

*point at 0 °C) and the mean basal temperature with respect to the pressure melting point of the patch can be used to estimate the parameters $T_{ramp}$ and $T_{exp}$ of the basal temperature ramp (Eq. (3)). [...] Consequently, this estimate for the basal temperature ramp should be a lower bound to the points in the scatter plot. [...]'.* The scatter plot described here is what is shown in Fig. 7. To make this clearer, Fig. 7 is now referenced right after the first 'scatter plot'.

Additionally, we have added: *For example, an upscaled* 25 km *patch (containing* 64 3.125 km *grid cells) with* 32 3.125 km *grid cells at the pressure melting point and* 32 3.125 km *grid cells at* $-1°C$ *with respect to the pressure melting point has a warm-based fraction of* 50 % *and a mean basal temperature of* $-0.5°C$. to the figure caption.

[3.3 - Brinkerhoff and Johnson, 2015 suggest that the inclusion of membrane stresses leads to convergence under spatial grid refinement, whereas without them, the SIA does not lead to convergence. Can you place those results in the context of this section? Are the relatively weak convergence results here a result of GSM and PISM velocity solvers being insufficiently 'higher-order'?] There is a possibility that the velocity-dependent switch between pure SIA and a membrane stress approximation used in both the GSM and PISM is insufficiently 'higher-order'. To test this, we ran additional GSM resolution scaling experiments with the SSA active everywhere. The differences in surge characteristics between the different resolutions are comparable to the results with the velocity-dependent SSA activation criteria. Therefore, we conclude that the velocity solvers are not the only reason for the strong resolution dependence. We added the following paragraph to the summary section: *Even though the studies are not directly comparable, the results of Brinkerhoff and Johnson [2015] offer some insight relevant to this study. For example, they suggest membrane stresses are necessary for convergence under horizontal grid refinement. The hybrid SIA/SSA ice dynamics used in the GSM and PISM might be insufficiently 'higher-order' and lead to a stronger resolution dependence than the schemes used in Hindmarsh [2009], Brinkerhoff and Johnson [2015]. However, GSM experiments with the SSA active everywhere show a resolution dependence comparable to the velocity-dependent SSA activation criteria (Table S24 and S18, respectively), indicating that the hybrid SIA/SSA ice dynamics are not the sole reason for the strong resolution dependence.*

Note that an analysis of GSM sensitivity to different SIA/SSA switching rules was added to the revised draft (Sec. S1.2).

[Sec. 4 If you maintain these research questions as an organizing principle, I would like to see them revisited as they are resolved by the experiments rather than all at once at the end.] We transformed this section into a 'Results Summary and Discussion' to make it a more standalone section. However, we suspect that not every reader will be interested in every detail of the results section. The new 'Results Summary and Discussion' section provides an easy way to get the most important information and allows the reader to then jump to individual results for more details. Therefore, we would like to answer the research questions in this revised section.

[L738–746 - I think that this section is kind of weird: none of the results presented in this work actually refute the resolution dependency conclusions of Hindmarsh (2009) or Brinkerhoff and Johnson (2015), yet the paragraph is written as if they did. As mentioned before, it seems just as reasonable to assert that those works saw more robust numerical convergence due to the use of more consistent membrane stress resolution schemes rather than because they fortuitously (or nefariously) made parameter choices that suppressed resolution effects.] We agree that this paragraph was confusing. We added it with the intention of providing possible explanations for the different conclusions, not to refute the conclusions of Hindmarsh [2009] or Brinkerhoff and Johnson [2015]. While the fact that different parameter choices can yield very different results remains, we want to clarify that we do not accuse the authors of making parameter choices that suppress resolution effects.

We have updated: *'Event characteristics in both the GSM and PISM show a strong resolution dependence for all sensitivity tests (Table 13+14 and 15). This is in contrast to the findings of other studies examining thermally induced ice streaming Hindmarsh [2009], Brinkerhoff and*

*Johnson [2015]. However, both of these studies analyze just one parameter vector, and it is relatively easy to find a parameter vector for which, e.g., the GSM exhibits only a minor resolution dependence. While Hindmarsh [2009] considers sub-temperate sliding, his model allows sliding far below the pressure melting point (order of $\delta = 1$ compared to $\delta = 0.01$ within this study, Eq. (5) and focuses on steady ice streams, not binge-purge-type surges. Over 200 kyr, even minor differences at the beginning of a run can slowly accumulate and yield overall different surge characteristics (e.g., Fig. S31). Furthermore, Brinkerhoff and Johnson [2015] examine ice stream statistics over the whole domain and not a specific soft-bedded region. Neither Hindmarsh [2009] nor Brinkerhoff and Johnson [2015] consider a bed thermal model.'*

to: *'Surge characteristics in both the GSM and PISM show a strong resolution dependence for all sensitivity tests (Table S18+S22 and S25). While other studies examining thermally induced ice streaming do not find a strong resolution dependence [Hindmarsh, 2009, Brinkerhoff and Johnson, 2015], these studies are not directly comparable. The different results are likely due to differences in the experimental design. For example, neither Hindmarsh [2009] nor Brinkerhoff and Johnson [2015] consider a bed thermal model. While Hindmarsh [2009] considers sub-temperate sliding, his model allows sliding far below the pressure melting point (order of $\delta = 1$ compared to $\delta = 0.01$ within this study, Eq. (10)) and focuses on steady ice streams, not ice stream surge cycling. Over 200 kyr, even minor differences at the beginning of a run can slowly accumulate and yield overall different surge characteristics (e.g., Fig. S30). Furthermore, Brinkerhoff and Johnson [2015] examine ice stream statistics over the whole domain and not a specific soft-bedded region. Additionally, both of these studies analyze just one parameter vector, and there are some parameter vectors for which, e.g., the GSM exhibits only a minor resolution dependence.'*

and the paragraph outlined in our response to comment *3.3 - Brinkerhoff and Johnson, 2015 suggest that the inclusion of membrane stresses leads to convergence under spatial grid refinement, whereas without them, the SIA does not lead to convergence. Can you place those results in the context of this section? Are the relatively weak convergence results here a result of GSM and PISM velocity solvers being insufficiently 'higher-order'?*


**Author's response to Anonymous Referee 2 Comment 1**

June 27, 2023

**1 General Comment**

[The goal of this study is a worthy one, and though the general topic of simulating thermal oscillations in ice sheet models has recieved considerable attention over three decades, a systematic study of the dependence of oscillation characteristics on choices in the numerical implementation has not been done. The main issue I see is that the manuscript in its current form is very challenging to read. It is organized more like a reference guide to a large number of simulations, rather than a discussion of the link between real physical processes, numerical choices and clear recommendations for how to remedy these issues in future simulations. In order to be publishable in a form that will be usable by other researchers, the manuscript needs substantial re-organization, reduction in length and re-writing in places. Below the major issues are listed in more detail (I will wait to comment on minor issues in a subsequent revision):]

We thank the referee for their constructive comments. We have addressed all the referee's comments and revised the text accordingly. A point-by-point reply is reported below, with referee comments in orange and our replies in black.

We agree that the readability of the paper benefits from a general re-organization and focusing on the key takeaways (Numerical Model Error Estimates (MNEEs) must be considered when numerically modeling ice stream surge cycles, surge characteristics are sensitive to the basal sliding activation criterion, resolution dependency can be reduced by incorporating a resolution-dependent basal sliding activation criterion). To streamline the paper, we grouped the 11 research questions into three main themes: 1) determine the 'MNEEs' as a metric to determine the significance of a change in model configuration (note that we use the MNEEs because we are not able to determine the model error; we provide more details on this issue in our replies below), 2) using the MNEEs, determine the sensitivity of the surge characteristics (number of surges, mean surge periodicity, mean surge duration, and mean ice volume change during a surge) to model aspects with a physical relevance (particular focus on the basal sliding activation criterion), and 3) perform a convergence study based on the results of 1) and 2).

We also reworked the 'Results Summary and Discussion' section to provide a more extensive and accurate comparison to the existing literature (e.g., added comparison to the EISMINT temperature spokes) and to discuss the link between real physical processes and numerical choices. We also added recommendations for what key model components should be used when modeling ice stream surge cycling.

**2 Detailed comments**

[1. Currently, the organization of the manuscript feels like a laundry list of simulations completed and a description of the results, without prioritizing the most important and revealing simulations and little discussion for what the results mean and how they relate to other studies. The study also seems to currently have about 10-15 areas of focus, making it challenging to see which end up actually being important. My recommendation would be to take all the description of simulations where numerical choices don't appear to have much influence on the results

and move them into a supplement. They can be summarized in a few sentences perhaps at the end of the results section, but right now they add too much extra to the manuscript making it longer and hard to get through. This includes the sediment-hard bed transition, basal hydrology, basal hydrology instead of temperature ramp and the max time step.] While discussing numerical choices that do not significantly affect the results is important, we agree that these details can be moved to the supplement to make the manuscript more accessible to the reader. Following this suggestion, we grouped the individual research questions into the following main themes: 1) MNEEs (Q1), 2) sensitivity experiments with (previous Q2, Q3, Q4, Q5, Q7, Q8; now Q2, Q3, Q5, Q6, Q7, Q8) and without a significant effect on the results (previous Q6, Q9, Q10; now Q4, Q9, Q10), and 3) convergence study (Q11). The sensitivity experiments without a significant effect will be only briefly summarised in the main manuscript. Since the results of the convergence study depend on previous experiments, this aspect is discussed last.

[2. The way the results are currently described and presented also contributes to the challenge of reading through this manuscript. There are 15 tables, and it is difficult to understand what all the numbers in the tables mean. Figures 8 and 6 seem like a more intuitive way to present this information (though all the markers and line and shading in Figure 6 need explicit descriptions in the caption and perhaps a legend to be interpretable). There really shouldn't be more than a few tables in the main text of this manuscript, the rest should be relegated to a supplement.] We agree that there were too many tables in the main manuscript. To increase the readability, we replaced most of the tables in the results section with the two new figures 6 and 7 (same plot idea as the previous figure 6). These figures summarise the sensitivities of the surge characteristics of all model components with a significant effect in a concise way. In terms of model components analyzed, we are now focusing only on model components that have significant effects on the surge characteristics in order to reduce the burden on the reader.

Most information necessary to interpret the previous figure 6 (figure 9 in the revised draft) was already in the caption. However, to make it even clearer, we slightly updated the caption from:

*Percentage differences in event characteristics compared to the GSM base setup ($T_{ramp} = 0.0625$, $T_{exp} = 28$) for different basal temperature ramps at 3.125 km horizontal grid resolution. The ramps are sorted from widest (first row) to sharpest (last row, see Fig. S25 for a visualization of all ramps). The shaded pink regions mark the numerical noise estimates (Tab. 5) and the black numbers in the title of each subplot represent the mean values of the base setup. No runs crashed and all runs had more than 1 surge event. The first 20 kyr of each run are treated as a spin-up interval and are not considered in the above. The x-axis is logarithmic. The exact values are given in Tab. S5.*

to

*Percentage differences in event characteristics compared to the GSM base setup ($T_{ramp} = 0.0625$, $T_{exp} = 28$) for different basal temperature ramps at 3.125 km horizontal grid resolution (average of the 5 parameter vectors). The ramps are sorted from widest (first row) to sharpest (last row, see Fig. S24 for a visualization of all ramps). Otherwise same as Fig. 6. No runs crashed and all runs had more than 1 surge event. The exact values are given in Tab. S10.,*

where the caption in figure 6 states

*[...] The different colors were added for visual alignment of the individual model setups, the stars are the ensemble mean percentage differences, and the horizontal bars represent the ensemble standard deviations. The shaded pink regions mark the MNEEs (Table 5) and the black numbers in the title of each subplot represent the mean values of the reference setup. The 3 small numbers between the first two columns represent the number of crashed runs (nC), the number of runs without a surge (nS0), and the number of runs with only one surge (nS1), respectively. The first 20 kyr of each run are treated as a spin-up interval and are not considered in the above. The x-axis is logarithmic. [...]*

Note that figures in the form of figure 8 (now figure 12) were not suitable to replace the tables because they only show the results of one parameter vector.

[3. The "such as" in the title of the paper is misleading. The only type of ice sheet instability discussed and tested in this paper is a thermal oscillatory instability (or B-P). A more accurate title would be simpler: "Numerical issues in modeling binge-purge type cyclic ice stream surging"] In order to give a more accurate description of the content of the manuscript, we changed the title from: *Numerical issues in modeling ice sheet instabilities such as binge-purge type cyclic ice stream surging*

to: *Numerical issues in modeling thermally and hydraulically driven ice stream surge cycling.*

[4. Related to #3 and throughout - the term "binge-purge" oscillations has largely fallen out of favor in the ice sheet modeling community. These are more commonly called thermal oscillations or ice stream activation-stagnation cycles. It is OK to mention in the introduction that these have historically been referred to as binge-purge, but it isn't in keeping with the field to continue to refer to them as such throughout.] Thanks for bringing this up. Except for historical contexts, we now use *ice stream surge cycling* instead of *binge-purge cycling.* Note that we were not aware of such a development [e.g., Roberts et al., 2016, Feldmann and Levermann, 2017, Ziemen et al., 2019, Schannwell et al., 2023, still use 'binge-purge']

[5. I am quite confused over how numerical noise is defined in this manuscript. It seems in section 3.1.3 that to quantify numerical noise that different solver choices are used. However, depending on the number of iterations occuring between the different tolerances (and the details of how the solver works) it would seem that this method could yield strongly different estimates for the "noise". Additionally, it is unclear why this is the correct bar for determining whether a change is "important", instead of a more physically meaningful quantity. Additionally, the rationale behind the set of simulations detailed in Table 6 is confusing. In a bitwise reproducible code, I don't see why the number of cores used in a simulation should have any influence on the simulations. This makes me concerned about the robustness of other simulations if the number of cores has such an important influence on the results. How reproducible are these results by other researchers? If the same setup is run on a different cluster architecture would the results be different?]

We agree that the concept of 'numerical noise' needed clarification. We changed the term 'numerical noise' to 'minimum numerical error estimates' and updated the first part of the introduction from: *The use of Ice Sheet Models (ISMs) has grown at least an order of magnitude over the last two decades. The relevance of such modeling studies to the actual physical system can be unclear without careful consideration and testing of numerical components and implementations. Model validation is particularly important when modeling highly non-linear ice sheet instabilities, for which it is hard to distinguish between numerical noise and physical phenomena. In addition, there are a number of numerical choices, such as for thermal activation of basal sliding, for which no model to date has documented sensitivities.*

to: *The use of Ice Sheet Models has grown at least an order of magnitude over the last two decades. The relevance of such modeling studies to the actual physical system can be unclear without careful consideration and testing of numerical aspects and implementations. This is especially true when modeling the highly non-linear ice sheet surge instability, which has significant implications not only for the ice sheet itself but also for the climate. In fact, it is often difficult to assess whether model results are physically significant (effects of physical system processes), a consequence of model-specific numerical choices, or a combination of both. This is especially important in the case of abrupt changes. Whether ice sheet instabilities observed in numerical simulations are the result of physical instabilities of the underlying continuum models or spurious effects of the discretization and numerical implementation of said models has long been debated [e.g., Payne et al., 2000, Hindmarsh, 2009] and is a consequential matter. The present study is concerned with characterizing the impact of model physics, numerical choices, and numerical errors on ice stream surge cycling.*

[...]

*As a result of the involved physics and expected behaviors, modeling of ice stream surge cycling is challenging. The challenges entail, among others, rapid surge onset, high ice velocities,*

*and non-linear (thermo-viscous, hydraulic, and thermo-frictional) feedbacks. In addition to the physical complexity, further challenges arise in the numerical modeling of ice stream surge cycling, whether in terms of model choices (e.g., choice of mechanical model, thermal modeling of the substrate, accounting for sub-glacial hydrology) and/or in terms of their numerical implementation (e.g., grid and time step size, convergence under grid refinement, etc.).*

*Our focus here is on the challenges arising from numerical modeling, both those related to the modeling choices and those related to the implementation. Numerical challenges have received limited attention in studies examining ice sheet surging. The few studies to date that do examine numerical aspects of surge cycling suggest strong sensitivities in model response to implementation choices such as grid size [e.g., Calov et al., 2010, Roberts et al., 2016, Ziemen et al., 2019]. However, the effects of different approximations of the Stokes equations have been previously addressed [e.g., Brinkerhoff and Johnson, 2015], and are therefore not discussed here.*

*[...]*

*In terms of different numerical choices, the impact on model results is usually determined by calculating the model error to the exact analytical solution. However, the theory behind the surge instability is not fully developed (no analytical solution exists), especially in the context of a spatially extended 3D system, thus precluding systematic benchmarking of numerical models.*

*To overcome this issue and provide at least a minimum estimate of the numerical model error, we first determine 'Minimum Numerical Error Estimates' (MNEEs). This is a new metric that aims to minimally resolve whether a change in surge characteristics due to changes in the model configuration is significant (see Sec. 2.3 for details).*

The corresponding part in Sec. 2.3 states: *We compute the new 'Minimum Numerical Error Estimates' (MNEEs) metric by examining the model response to changes in the model configuration that are not part of the physical system. The MNEEs are defined as the percentage differences in surge characteristics when applying a stricter (than default) numerical convergence in the GSM and adjusting the matrix solver used in PISM (changing the number of processor cores used). They are then used as a threshold to determine if model sensitivities to changes in the model configuration that affect the physical system (e.g., the inclusion of a bed thermal model or sliding depence on effective pressure from basal hydrology) are above the level of background noise induced by iterative numerical solvers in the model. We refrain from drawing conclusions about the effects of a change in model configuration with physical relevance when the model sensitivities in question are smaller than the MNEEs. In these cases, the actual physical response of the model might be hidden within the numerics.*

*While the MNEEs are useful to our purpose, we wish to emphasize that they can not replace proper model verification and validation and are missing uncertainties due to, e.g., different approximations of the Stokes equations and other physical processes not included in the models. Nonetheless, they provide a minimum estimate of the numerical model error, which is still a significant improvement over ignoring this issue entirely.*

The above description should clarify our intentions behind using the MNEEs instead of a *physically meaningful quantity* (we are unsure how a *physically meaningful quantity* could determine whether the modeled result is due to numerical errors or a true physical phenomenon). However, we have tested the effect of different tolerances and added the following short statement to the revised draft: *The differences in surge characteristics between different numbers of cores can be minimized by decreasing the relative Picard tolerance in the calculation of the vertically-averaged effective viscosity ($10^{-4}$ to $10^{-7}$) and the relative tolerance for the Krylov linear solver used at each Picard iteration ($10^{-7}$ to $10^{-12}$, Table S5 and Fig. S19). However, this leads to an unreasonable increase in model run time ($\sim 300\ \%$) that is not feasible for an ensemble-based approach (more than $50\ \%$ of all runs did not finish within the time limit of the computational cluster). Intermediate decreases in the relative tolerances still lead to significant differences in surge characteristics while increasing the model run time and are, therefore, not used in the PISM reference setup.*

When dealing with non-linear transitions such as surge onset and shutdown, small differences

can accumulate and lead to a somewhat different result at the end of the model run. These small differences can be caused by, e.g., a different number of cores (see **online PETSc-FAQ link**). PETSc is used by PISM. So the results might be somewhat different on a different cluster. However, the whole point of introducing the MNEE threshold is to identify what characteristics and relations are likely to be robust.

[6. In section 4 I see the summary of results, but very little discussion of what they mean (in some cases but not others). This seems to me to be the main missing piece of the manuscript to be useful to other researches, a discussion of how these results relate to the theory of thermal oscillations (from Schiavi, Mantelli, Robel, MacAyeal, etc.) and how they might relate to other ice sheet models.] Where possible, we added a more in-depth discussion to the revised draft (see below). However, a comparison to the existing theory is not straightforward because the theory in the context of 3D thermo-mechanically coupled models with hybrid SIA/SSA ice dynamics is not fully developed.

$Q_1$ - MNEEs: *We expect other ice sheet models with a comparable experimental design and ice dynamics to show similar levels of MNEEs. To minimize the possibility of interpreting numerical errors as a physical response to a change in model setup, it is crucial to determine MNEEs (or a comparable metric).*

$Q_3$ - interface temperature: *The additional heat transfer to the grid cell interface is comparable to spreading 50 % of the basal heating effect from sliding in a grid cell to the surrounding grid cells used in mPISM (latest version based on PISM v0.7.3) [e.g., Ziemen et al., 2014, 2019, Schannwell et al., 2023]. This spreading of basal heating warms the grid cells adjacent to an ice stream and was necessary to model Heinrich Event-like surges (Florian Ziemen, personal communication). While no additional heat transfer was added to PISM v2.0.2 used within this study, the till friction angles had to be reduced to model surges.*

$Q_5$ - basal temperature ramp: *To account for observational and experimental evidence of sub-temperate sliding [Barnes et al., 1971, Shreve, 1984, Echelmeyer and Zhongxiang, 1987, Cuffey et al., 1999, McCarthy et al., 2017], avoid an abrupt onset of sliding at the warm/cold-based transition that causes refreezing on the warm-based side [Mantelli et al., 2019], and minimize resolution dependencies, a basal temperature ramp (or similar mechanism) should be implemented in all ice sheet models for contexts where surge onset/termination are important.*

$Q_6$ - abrupt sediment transition: *Since the sediment cover can change within a few kilometers [e.g., Andrews and MacLean, 2003], we conclude that, despite the minor differences, an abrupt transition between soft and hard beds is a reasonable simplification, especially considering horizontal grid cell dimensions of 25 km or larger.*

$Q_7$ - bed topography: *In agreement with previous modeling studies [e.g., Winsborrow et al., 2010, and references within], the topography is a key aspect of ice stream modeling. When interested in a comparison with observational data or proxy reconstructions, a more realistic topography (in contrast to the idealized flat topography) should be used.*

$Q_{11}$ - convergence study: *Even though the studies are not directly comparable, the results of Brinkerhoff and Johnson [2015] offer some insight relevant to this study. For example, they suggest membrane stresses are necessary for convergence under horizontal grid refinement. The hybrid SIA/SSA ice dynamics used in the GSM and PISM might be insufficiently 'higher-order' and lead to a stronger resolution dependence than the schemes used in Hindmarsh [2009], Brinkerhoff and Johnson [2015]. However, GSM experiments with the SSA active everywhere show a resolution dependence comparable to the velocity-dependent SSA activation criteria (Table S24 and S18, respectively), indicating that the hybrid SIA/SSA ice dynamics are not the sole reason for the strong resolution dependence.*

---

## Author Response (AR2)

**Author's response to Anonymous Referee 2 Comment 2**

August 8, 2023

**1 General Comment**

[This is an updated version of the manuscript with new title "Numerical issues in modeling thermally and hydraulically driven ice stream surge cycling" by Hank et al. It is greatly improved based on revisions responding to prior reviews. I think it has more logical flow and clearer messages about the importance of various modeling choices in simulating ice stream surging.

In some ways I think the revision does not quite go far enough in paring down the level of detail in the manuscript and the amount of repetition in describing the results (mainly in the amount of recapping of results in section 4 right after they are described in a similar level of detail in section 3). However, I also recognize that stylistic suggestions regarding the structure of papers are ultimately up to the authors. As someone who has worked on this problem and is predisposed to being very interested in the topic, I found it hard to get through the 40+ pages of painstaking detail on this suite of modeling experiments as a linear reading experience. It may be that most readers will use this paper as a reference guide, without reading it in its entirety. Even in such a form, it will be a useful contribution to the field. Ultimately it is up to the authors to decide whether it is worth substantially reducing the level of detail in sections 2 and 4. In section 2, there are places where modeling detail not pertinent to the questions defined in this study could be moved to a supplement or even just cited out to a previous paper. In section 4, there are many descriptions of results that are also given in section 3 which could be removed. Section 1.3 could be greatly shortened by just stating the questions you set out to answer. The first 20 lines of section 5 recap results right after a section that is largely recapping of results. Overall, I think it is possible for this manuscript to be 25 % shorter without losing much of the important detail of interest to readers.

One other major concern that remains from the first review is that it does not seem like a parameterization should be explicitly made to be resolution-dependent. It is unclear that the "optimal" temperature ramp parameters for coarse resolution would be the same between GSM and other models, and its unclear what is gained by designing the parameterization this way since it will undoutedly affect many others metrics beyond those which are the focus of this study. Additionally, it appears that the result of "compensation" of temperature ramp parameters for resolution difference are not even definitive, so why not just exclude this aspect of the discussion?]

We thank the referee for their constructive comments. We have addressed all the referee's comments and revised the text accordingly. A point-by-point reply is reported below, with referee comments in orange and our replies in black.

To reduce the length of the paper, we removed the repetitive parts in Section 4. Furthermore, we removed all details in Section 2 that are not directly related to one of the experiments in this manuscript. The maximum time step size experiment was removed entirely, as it did not add anything new to the literature. Additionally, we removed the first part of the conclusions (Section 5), as this information is also given in Section 4. We refrain from just stating the research questions in Section 1.3 because we prefer to have at least some context of why these

questions are important and why we are trying to answer them. However, we shortened the description to the essential information. Overall, a reduction of 25 % (or 10 pages) is not possible without removing essential information or entire experiments, especially since the reduction was partly offset by some of the requested revisions (e.g., replacement of Figure 12). Nevertheless, we were able to reduce the length of the paper by 4 pages.

The arguments regarding the necessity for a basal temperature ramp and its resolution dependency are clearly stated in the text and would apply to any discretized glaciological ice sheet model. However, we agree that, e.g., different ice dynamics might lead to slightly different "optimal" basal temperature ramp parameters. We make this clear now with *However, given potential dependencies on the particular ice sheet model, we recommend resolution testing to determine the optimal basal temperature ramp.*. Since this is discussed in Section 4, we removed it from the conclusions.

**2  Detailed comments**

Responses to comments requiring a detailed response are listed below. Otherwise we have fully implemented the specific suggestions of the reviewer and do not list them below.

[Throughout: the term parameter "vector" is more typically called a parameter "set"] Since "parameter vector" is also common usage and technically accurate, we favor this.

[Ln 8: high → fine] We replaced all instances of *high horizontal grid resolutions* with *fine horizontal grid resolutions*.

[Ln 17: delete sentence one] and [Ln 17: The relevance of ice sheet modeling...of numerical implementations.] The first sentence was included to emphasis that despite the frequent use of ice sheet models within the glaciological community, the numerical aspects are often a *black box* for their users. To keep this emphasis, we decided to stick to the current version: *The use of ice sheet models has grown at least an order of magnitude over the last two decades. The relevance of such modeling studies to the actual physical system can be unclear without careful consideration and testing of numerical aspects and implementations.*.

[Ln 21: significant → realistic] While it is difficult to assess whether model results are physically realistic (especially in a paleo context), this is not the point we want to raise here. Furthermore, the results can be within a physically realistic range but still be a consequence of model-specific numerical choices. Therefore, we keep *physically significant*.

[Ln 31: work → heat] and [Ln 34: work → heating] Ln 31 is correct as stated *heat from geothermal and deformation work sources*. For Ln 34, we now state *increase of heat from deformation work*.

[Ln 34: delete "or to"] Changed: *can warm the surrounding ice close to or to the pressure melting point*

to: *can warm the surrounding ice (close) to the pressure melting point*.

[Ln 44: delete "quasi"] Since the periodicity is often irregular, we prefer to keep *quasi*. However, to clarify this, we updated the reference from [Robel et al., 2013] to [Souček and Martinec, 2011].

[Ln 84: model equations are not a numerical choice, they are a component of the system this is being modeled.] We agree that the wording was too vague. Normally, anything required just because of discretization (in time and space) is not a model equation. To make this clearer, we updated: *In this study we seek to disentangle the effects of numerical choices (both in terms of model components and in terms of their implementation) on ice sheet surges.*

to: *Herein, we disentangle the effects of numerical choices (e.g., grid size) and physical system processes (e.g., sub-temperate basal sliding) on ice sheet surges via numerical experiments.*

[Ln 92: delete "high variance"] The "high variance" is a key aspect of our experimental design, as we try to *partly address potential non-linear dependencies of surge cycling on model parameters*. Therefore, we prefer to keep this phrase.

[Ln 213: shelf-shallow ice] Changed to: *hybrid shallow shelf/ice physics*.

 To avoid the apparent confusion, we've change *one-at-a-time* to *one-factor-at-a-time*.

 The effect of different SSA activation thresholds was tested and is briefly decribed in Sec. 2.1.1: *The hybrid SIA/SSA ice dynamics are activated for grid cells with a SIA velocity exceeding* 30 m yr$^{-1}$. *Changing these activation velocities (*20 m yr$^{-1}$ *and* 40 m yr$^{-1}$*) has no significant effect on the surge characteristics (Table S1). Activating the SSA everywhere leads to more, shorter, and weaker surges because no threshold velocity needs to be overcome to initiate basal sliding (Sec. S1.2). Note that we set an upper limit of* 40 km yr$^{-1}$ *for the SSA velocity to ensure that sliding velocities stay within a physically reasonable range.*

The exact results can be found in Sec. S1.2 of the supplement.

 Thanks for bringing this up. We have updated: *Since the GSM setup and climate forcing are symmetric about the horizontal axis in the middle of the pseudo-Hudson Strait (y =* 250 km *in Fig. 1), we interpret the induced asymmetry as a numerical induced bifurcation.*

to: *Since the GSM setup and climate forcing are symmetric about the horizontal axis in the middle of the pseudo-Hudson Strait (y =* 250 km *in Fig. 1), we interpret the induced asymmetry as 'spontaneous symmetry breaking' similar to the results described in Sayag and Tziperman [2011].*

 To avoid debate over what *comparison* means/entails, we've changed the original sentences from: *As the two model setups and physics are somewhat different (see Table 2 for details), we do not intend to compare model results directly. Instead, our aim is to increase confidence in model results by showing that the same conclusions can be drawn from two different models.*

to: *As the two model setups and physics are somewhat different (see Table 2 for details), this permits more confident conclusions that are not model specific.*

 Agreed. We have added the following sentence to the revised draft: *These minor differences can be caused by, for example, a different order of floating point arithmetic operations and the processor-number-dependent preconditioner used in PISM [PISM 2.0.6 documentation, 2023].*

 As mentioned in your comment above, many readers will expect the same or at least similar results for different numbers of cores. We decided to use the number of cores for two reasons. 1) To emphasize that in a highly non-linear system such as the one examined here, even the smallest differences can lead to substantial differences in surge characteristics. 2) To show the potential numerical sensitivity of the default PISM setup, likely blindly used by many ice sheet modellers, to prompt the community to pay more attention to numerical issues.

 It is noise at every climatic time step. We updated: *Adding low levels of uniformly distributed surface temperature noise (±0.1°C and ±0.5°C) to the climate forcing does not significantly affect the surge characteristics for the GSM (Table S3).*

to: *Adding low levels of uniformly distributed surface temperature noise (maximum amplitude of ±0.1°C and ±0.5°C) to the climate forcing (updated every* 100 yr*) does not significantly affect the surge characteristics for the GSM (Table S3).*

[Ln 527: but is the implicit coupling scheme more accurate (for example with respect to grid

spacing)] A priori, the implicit scheme will be more accurate. But given the highly increased computational cost (increase in the run time of $\sim 265$ %), a repeat of the horizontal grid resolution convergence study with the implicit coupling scheme is beyond the bounds of this paper.

[Ln 634-651: the discussion here is confusing and could be shortened and clarified] To make this part less confusing, we changed it from: *We complement the above analysis by upscaling the 3.125 km reference runs. For example, a 25x25 km grid cell contains a patch of 64 3.125x3.125 km grid cells. The scatter plot (e.g., Fig. 10) of the warm-based fraction (basal temperature with respect to the pressure melting point at $0\,°C$) and the mean basal temperature with respect to the pressure melting point of the patch can be used to estimate the parameters $T_{ramp}$ and $T_{exp}$ of the basal temperature ramp (Eq. (8)). However, this does not account for the connectivity between the faces of, e.g., a 25 km grid cell. Without a continuous warm-based channel from one grid cell interface to another, there should be effectively no basal sliding across the grid cell, even when the average basal temperature is close to the pressure melting point. Consequently, this estimate for the basal temperature ramp should be a lower bound to the points in the scatter plot. Furthermore, the upscaling results depend on the bed properties (soft sediment vs. hard bedrock) and the specific scenario (surge vs. quiescent phase). As such, the upscaling statistics only consider grid cells within the pseudo-Hudson Strait area during surges. Due to the limited storage capacity for the 10 yr output fields, only the first 10 kyr after the first surge are used for the upscaling experiments.*

to: *A more physically-based approach to determining an appropriate scale-compensating temperature ramp stems from our motivation for research question Q5 above. We bundle all 3.125x3.125 km grid cells of our reference runs into patches of, e.g., 64 grid cells. Each patch represents a coarser, e.g., 25x25 km grid cell. We then determine the warm-based fraction (basal temperature at the pressure melting point) and the mean basal temperature with respect to the pressure melting point of each patch. We can then estimate the parameters $T_{ramp}$ and $T_{exp}$ of the basal temperature ramp (Eq. (8)) by plotting the warm-based fraction against the mean basal temperature for all patches (e.g., Fig. 10) and fitting a basal temperature ramp with the preliminary assumption that a corresponding coarse grid cell should have an ice streaming fraction proportional to the sub-grid warm-based area.*

*However, this upscaling analysis does not account for the connectivity between the faces of, e.g., a 25 km grid cell. Without a continuous warm-based channel from one grid cell interface to another, there should be effectively no basal sliding across the grid cell, even when the average basal temperature is close to the pressure melting point. Consequently, the best estimate for the two parameters of the basal temperature ramp should be a lower bound to the points in the scatter plot.*

*Furthermore, the upscaling results depend on the bed properties (soft sediment vs. hard bedrock) and the specific scenario (surge vs. quiescent phase). Therefore, we only consider patches within the pseudo-Hudson Strait area during surges. Due to the limited storage capacity for the 10 yr output fields, only the first 10 kyr after the first surge are used for the upscaling experiments.*

However, due to this clarification, we were unable to shorten this part.

[Ln 741: the correct way to describe these results is that the solution exhibits "convergence", but is not "converged"] We changed it from: *In general, both models show convergence, but the discrepancies between different horizontal grid resolutions are significant.*

to *In general, both models exhibit convergence under systematic horizontal grid refinement for the overall ice volume (mean bias, Table S19+S23 and S26), but the solution is not fully converged at the finest resolutions tested.*

[Figure 12: It would be clearer to plot the surge characteristics/metrics with respect to grid spacing. This is the typical plot made in convergence studies.] Agreed. We moved Figure 12 into the supplement and replaced it in the main text with a plot similar to Figures 6 and 7.

[Section 3.4.2: it is difficult to take anything away from this convergence study since half

of the simulations did not finish. Would make more sense to just delete this section an add a sentence at end of previous section saying that PISM is too computationally intensive to conduct a robust convergence study.] We agree that the limited number of runs can somewhat skew the statistics, especially when comparing the results of the convergence study to those of other PISM experiments. However, the remaining runs still provide useful insight. Furthermore, the GSM convergence study (and all other experiments) is only based on one additional run (5 GSM vs. 4 PISM runs). Considering that runs crashed for some GSM experiments, excluding the PISM convergence study based on these grounds would also mean excluding some GSM experiments. Since this section is already quite short, we prefer to keep it its current form.

[Ln 788-794: I struggled with this paragraph because it is not clear that is possible to compare the magnitude of sensitivity to very different kinds of changes to parameters and other numerical choices. It would be better to just whether changes were greater or less than MNEEs. Isn't that the point of defining these?] Agreed. We removed this part from the revised draft.

[Ln 811-812: why do you expect this? Seems to be quite a big claim.] Upon rereading, our intended message (without checking the actual numerical sensitivity, one should not assume that MNEEs can be ignored) was lost. We have updated this paragraph from: *We expect other ice sheet models with a comparable experimental design and ice dynamics to show similar levels of MNEEs. To minimize the possibility of interpreting numerical errors as a physical response to a change in model setup, it is crucial to determine MNEEs (or a comparable metric).*

to: *Given the nonlinearities in the SSA (or higher approximation) ice sheet system, there is no a priori reason to confidently assume other ice sheet models will have ignorable MNEEs for unstable contexts such as surge cycling and grounding line response. Therefore, it is crucial to determine MNEEs (or a comparable metric) to minimize the possibility of interpreting numerical errors as a physical response to a change in model setup or forcing.*

[Ln 880-881: I'm not sure that this can be generalized since this hydrological model is quite simple and doesn't simulate many things other subglacial hydrology models do, like the increase in effective pressure with the development of channelized hydrology.] Thanks for bringing this up. While we have not tested different hydrology models for the GSM in this study, the simple hydrology model has been shown to be adequate for this context given the large parametric uncertainties in more complete basal hydrology models [Drew and Tarasov, 2022, under review]. We have added: *In general, this also holds for subglacial hydrology models with higher complexity [Drew and Tarasov, 2022, under review].*

[Ln 964: given the difference in configurations should we even expect to get temperature spokes in these simulations? fast flow is confined to a narrow strip which is different from the EISMINT simulations.] We agree that we do not necessarily expect temperature spokes in our simulations. This comparison was added by request of the first reviewer. Since it does not add key information to the understanding of this manuscript, we removed it from the revised draft.

**References**

M. Drew and L. Tarasov. Surging of a hudson strait scale ice stream: Subglacial hydrology matters but the process details don't. *The Cryosphere Discussions*, 2022:1–41, 2022. doi: 10.5194/tc-2022-226. URL https://tc.copernicus.org/preprints/tc-2022-226/.

PISM 2.0.6 documentation. Petsc options for pism users, August 2023. URL https://www.pism.io/docs/manual/practical-usage/petsc-options.html.

A. A. Robel, E. Degiuli, C. Schoof, and E. Tziperman. Dynamics of ice stream temporal variability: Modes, scales, and hysteresis. *Journal of Geophysical Research: Earth Surface*, 118(2):925–936, 2013. ISSN 21699011. doi: 10.1002/jgrf.20072.

Roiy Sayag and Eli Tziperman. Interaction and variability of ice streams under a triple-valued sliding law and non-Newtonian rheology. *Journal of Geophysical Research: Earth Surface*, 116(1), 2011. ISSN 21699011. doi: 10.1029/2010JF001839.

Ondřej Souček and Zdenek Martinec. ISMIP-HEINO experiment revisited: Effect of higher-order approximation and sensitivity study. *Journal of Glaciology*, 57(206):1158–1170, 2011. ISSN 00221430. doi: 10.3189/002214311798843278.

---

## Author Response (AR3)

**Author's response to Editors Comment 1**

August 24, 2023

**1 General Comment**

[Thank you for submitting a revised version of your article draft. Although you addressed most of the concerns and suggestions raised by the reviewers and myself, I feel another and hopefully final round of minor modifications would be needed to have the manuscript in good shape for publication. You'll find in the attached file a version of your latest manuscript highlighted including notes for various issues one still would need to address prior to publication.

The minor revision should focus on: - Removing research question list style (Q1-11) and rather use the other grouping already implemented into 3 or 4 subsections, discussing these points in a logical and systematic fashion. This would add clarity and allow to remove some repetition. - Maybe be more explicit that all this stud arises from using existing software that creates limitations where there may not be any if another approach was employed. For example, the fact that the PISM results significantly depend on the number of cores used for the simulation is something that should not happen. I feel some of the issues arise also because of the various parametrisation of physical processes used in the target software. Reflecting a this in, e.g., the conclusion would be valuable. - Some additional work is needed on polishing the technical notification and equations. Only variables should be in italic, and ideally one could define concise variable for some of the long terms that show up in the equations.]
We thank the editor for their constructive comments. We have addressed all the editor's comments and revised the text accordingly. A point-by-point reply is reported below, with referee comments in orange and our replies in black.

As all general comments are also addressed by detailed comments, please refer to the detailed comments sections for our replies.

**2 Detailed comments**

Responses to comments requiring a detailed response are listed below. Otherwise we have fully implemented the specific suggestions of the editor and do not list them below.

[Title: change "issues" to "challenges"] To better describe the content of this manuscript and include all of our experiments, we updated the title from: *Numerical issues when modeling thermally and hydraulically driven ice stream surge cycling*

to: *Modeling sensitivities of thermally and hydraulically driven ice stream surge cycling*

[Ln 19: could also be called "spontaneous localisation".] The sentence *This is especially important in the case of abrupt changes.* was removed entirely, as it is a repetition of *This is especially true when modeling the highly non-linear ice sheet surge instability, which has significant implications not only for the ice sheet itself but also for the climate.*

[Ln 80: This needs more details. The terminology here is already specific to the models and gives no information to the reader. What are you doing, running ensembles, running more nonlinear solver iterations, else?] We do not fully understand this comment, as there is no terminology specific to the models in this sentence and *high variance ensemble* explicitly states that we are running ensembles. However, we slightly updated this sentence from: *In order*

*to partly address potential non-linear dependencies of surge cycling on model parameters, we run each of our numerical experiments with a high variance ensemble of 5 GSM and 9 PISM parameter vectors.*

to: *To partly address potential non-linear dependencies of surge cycling on model parameters, we run each numerical experiment with a high variance ensemble of 5 GSM and 9 PISM parameter vectors instead of just a single run.*

Furthermore, we updated the first sentence of this paragraph from *In terms of ice flow models, we primarily use the 3D glacial systems [GSM, Tarasov et al., 2023].*

to: *In terms of ice flow models, we primarily use the 3D glacial systems model with hybrid shallow shelf/ice physics [GSM, Tarasov et al., 2023].*

[Ln 181: To me this seems to be among the most meaningful metric one could use in a study like this one and do not fully get how it is different from them MNEEs you are proposing] As stated in the cited line, the difference between results for different resolution (or different configurations) is the metric. The MNEEs are the minimal threshold for determining if these differences are significant. We've noted the text was confusing in this regard, and have now replaced the word *metric* with *threshold* when associated with the MNEEs: *This is a new metric that aims to minimally resolve whether a change in surge characteristics due to changes in the model configuration is significant (see Sec. 2.3 for details).*

to: *This is a minimal threshold to resolve whether a change in surge characteristics due to changes in the model configuration is significant (see Sec. 2.3 for details).,*

*We compute the new 'Minimum Numerical Error Estimates' (MNEEs) metric by examining the model response to changes in the model configuration that are not part of the physical system.*

to: *We compute the new 'Minimum Numerical Error Estimates' (MNEEs) threshold by examining the model response to changes in the model configuration that are not part of the physical system.,* and

*Therefore, it is crucial to determine MNEEs (or a comparable metric) to minimize the possibility of interpreting numerical errors as a physical response to a change in model setup.*

to: *Therefore, it is crucial to determine MNEEs (or a comparable threshold) to minimize the possibility of interpreting numerical errors as a physical response to a change in model setup.*

[Ln 369: How would changing the number of cores used result in a different matrix based solver? How does this reflect to the stricter numerical convergence criteria used in GSM, i.e., can't one do both for both codes?] To avoid confusion, we removed the part about the matrix based solver. The sentence now reads *The MNEEs are defined as the percentage differences in surge characteristics when applying a stricter (than default) numerical convergence in the GSM and changing the number of processor cores used in PISM.* and we moved up one sentence from the results section: *The differences between PISM runs with different numbers of processor cores can be caused by, for example, a different order of floating point arithmetic operations and the processor-number-dependent preconditioner used in PISM [PISM 2.0.6 documentation, 2023].*

While we tested stricter numerical convergence criteria for PISM (Sec. 3.2), they led to an unreasonable increase in the model run time beyond the run-time limit of the computational cluster. Furthermore, as mentioned in our last response to Referee 2, *many readers will expect the same or at least similar results for different numbers of cores. We decided to use the number of cores for two reasons. 1) To emphasize that in a highly non-linear system such as the one examined here, even the smallest differences can lead to substantial differences in surge characteristics. 2) To show the potential numerical sensitivity of the default PISM setup, likely blindly used by many ice sheet modellers, to prompt the community to pay more attention to numerical issues.*

[Ln 440: what does this mean? Is the outer loop the nonlinear solver? ideally, you monitor linear and nonlinear resduals, and make sure they all converge.] We updated this sentence from: *In a second experiment, we additionally increase the maximum iterations from 2 to 3 for the outer Picard loop (ice dynamics) and from 2 to 4 when solving the non-linear elliptic SSA*

*equation.*

to: *In a second experiment, we additionally increase the maximum iterations from 2 to 3 for the outer Picard loop solving for the ice thickness and from 2 to 4 when solving the non-linear elliptic SSA equation for horizontal ice velocities.*

[Ln 654: what is Pa?] Pa is the unit Pascal, as indicated in the text: *We enforce that $N_{\text{eff}}$ never falls below 10 kPa (denominator in Eq. (21), similar results for $N_{\text{eff,min}} = 5$ kPa).*

[Results Summary and Discussion: I would remove the Q-based layout and use the same 3 group you proposed in the introduction. Also, part of the discussion could be slightly srteamlined further in order to avoid repetition.] We removed parts of the MNEEs and basal temperature ramp paragraphs in the discussion section to further streamline the summary. However, the manuscript is already structured according to the three groups (minimum numerical error estimates (MNEEs), sensitivity experiments, and convergence study). The research questions mainly split the otherwise 10 pages long sensitivity experiment results section into further subgroups, providing additional guidance for the reader. Especially given the number of research questions, we feel strongly about retaining this structure (and wish it was more common when there are more than a few research questions in one paper).

[Conclusions: Maybe good to highlight here that agood part of the issues you encountered are actually due to models themselves, and that less parametrised models with correct multicore parallelisation should by construction avoid most of the issues and challenges you are reporting here.] We agree that some of these issues would be irrelevant in an ideal world (e.g., models with 50 m grid resolution). However, such high grid resolutions are currently unfeasible and will likely remain unfeasible for the foreseeable future. Therefore, all state-of-the-art ice sheet models, including the GSM and PISM, require some level of parameterization.

The processor-number-dependent preconditioner used by PETSc's KSP is critical to its parallel scalability [PISM 2.0.6 documentation, 2023]. Furthermore, the GSM avoids any parallelization issue (only single core usage) yet still shows numerical challenges.

As increasing the awareness of numerical challenges that must be considered when modeling surges is one of the key takeaways, we prefer not to include the proposed statement and avoid giving the reader a false sense of security.

**References**

PISM 2.0.6 documentation. Petsc options for pism users, August 2023. URL `https://www.pism.io/docs/manual/practical-usage/petsc-options.html`.

Lev Tarasov, Kevin Hank, and Benoit S. Lecavalier. Gsmv01.31.2023 code archive for lissq experiments, February 2023. URL `https://doi.org/10.5281/zenodo.7668472`.

---

## Author Response (AR4)

**Author's response to Editors Comment 2**

August 30, 2023

[Thank you for addressing partly my latest suggestions. I will accept the manuscript for publication. Ideally, I would appreciate if you could still remove the Q-based formulation.]

We previously tried other organizing principles as well as grouping the current 11 research questions into fewer and broader questions, but we still think the current approach is the best. The topics we address are too different to group them into more general themes. For example, we tried to group all the thermal aspects (bed thermal model, grid cell interface temperature calculation, weight of minimum adjacent temperature, basal temperature ramp) into one question. However, each of these aspects is a complex topic that needs to be explained and addressed individually. Overall, reducing the number of research questions does not increase the manuscript's readability, nor does it reduce the length of the manuscript.

Furthermore, we do not see the benefit of removing the research questions entirely. The description of the research questions needs to remain in the paper even without the research questions themselves, as it provides key information to understand the context of this manuscript. A reformulated text without the research questions would likely increase the length of the manuscript while decreasing the structure and readability. Therefore, we decided to keep the research questions.